# The role of oceanic heat flux in reducing thermodynamic ice growth in Nares Strait and promoting earlier collapse of the ice bridge

Sergei Kirillov[1], Igor Dmitrenko[1], David G. Babb[1], Jens K. Ehn[1], Nikolay Koldunov[2], Søren Rysgaard[1,3,4], David Jensen[1], and David G. Barber[1,†]

[1]Centre for Earth Observation Science, University of Manitoba, Winnipeg, Manitoba, Canada
[2]Alfred Wegener Institute, Bremerhaven, Germany
[3]Arctic Research Centre, Aarhus University, Aarhus, Denmark
[4]Greenland Institute of Natural Resources, Nuuk, Greenland
[†]deceased, 15 April 2022

*Correspondence to*: Sergei Kirillov (sergei.kirillov@umanitoba.ca)

**Abstract.** The ice bridge in Nares Strait is a well-known phenomenon that affects the liquid and solid freshwater flux from the Arctic Ocean through the strait and controls the downstream North Water polynya in northern Baffin Bay. Recently, the ice bridge has been in a state of decline, either breaking up earlier in the year or not forming at all, and thereby increasing the sea ice export out of the Arctic Ocean. The decline of the ice bridge has been ascribed to thinner and therefore weaker ice from the Arctic Ocean entering Nares Strait, however local forcing also affects the state of the ice bridge and thereby influences when it breaks up. Using a variety of remotely sensed data we examine the spatial patterns of sea ice thickness within the ice bridge, highlighting the presence of negative ice thickness anomalies on both the eastern and western sides of the Strait, and identifying a recurrent sensible heat polynya that forms within the ice bridge near Cape Jackson in northwestern Greenland. Using the sea ice-ocean model FESOM2, we then attribute these ice thickness anomalies to the heat from warmer subsurface waters of Pacific and Atlantic origin that reduces thermodynamic ice growth throughout winter on the western and eastern sides, respectively. The consequently weaker and thinner areas within the ice bridge are then suggested to promote instability and earlier break up. This work provides new insight into the structure of the Nares Strait ice bridge, and highlights that warming of the modified Atlantic and/or Pacific Waters that enter the Strait may contribute to its further decline.

## 1. Introduction

Nares Strait is a narrow waterway between Ellesmere Island and Northwestern Greenland that connects the Arctic Ocean and northern Baffin Bay (Fig. 1). This strait represents one of the major gates through which cold and fresh Arctic seawater and sea ice discharge into the north Atlantic (Kwok, 2005; Beszczynska-Möller et al., 2011; Münchow, 2016). The southward ice and water flow is maintained by persistent orographically channelled winds (Ito, 1982; Samelson et al., 2006) and a sea level gradient between the Lincoln Sea and Baffin Bay (Münchow and Melling, 2008; Shokr et al., 2020). The situation changes dramatically with the formation of the ice bridge[1] which blocks southward ice drift during winter. The presence of the ice bridge significantly reduces the annual sea ice transport through Nares Strait. An average annual ice export of ~141 km$^3$ during years when the ice bridge exists is about half of that exported during bridgeless years (Kwok et al., 2010). In particular, the ice bridge prevents the loss of thick, old ice from the Last Ice Area, located north of Ellesmere Island and Greenland (Moore et al., 2019), by hindering its transport south, where it in turn may affect navigation and the maritime industry (Barber et al., 2018). Generally, the composition

---

[1] In the absence of established consensus on the terminology, hereafter we prefer to use the term "bridge" for landfast ice blocking Nares Strait instead of "arch" which is used to describe the characteristic dome-like shape of the bridge's leeward (southern) edge.

of the ice bridge can be described as a mix of first year ice (FYI), multiyear ice (MYI) and a minor contribution of icebergs. Kwok (2005) reported that MYI comprises between 18% and 75% of the total ice area drifting through Nares Strait and accounts for most of the total ice volume flux (Kwok et al., 2010). Under-ice sonar measurements in Kennedy Channel in the northern part of the strait showed the modal peak of ice drafts of 2.0–2.1 m throughout winter, including the periods of ice growth and melt (Ryan and Münchow, 2017). Based on a limited high-resolution satellite altimetry dataset (ICESat-2, no-snow assumption) for Nares Strait in January 2020, Kirillov et al. (2021) demonstrated that the ice bridge mainly consisted of relatively thick ice with a pronounced peak in the fraction of sea ice with a draft between 2.6–2.8 m.

The period when the strait is impenetrable for ice transport depends on the timing of ice bridge formation and break-up. The ice bridge in Kane Basin has typically formed between late-October and early-April and broken up in June-July (Kwok, 2010; Vincent, 2019). Kirillov et al. (2021) demonstrated that drifting ice has a higher likelihood to consolidate under a specific combination of favorable atmospheric and oceanic conditions. Analysis of the conditions during 16 bridge formations between 2001-2021 revealed that the bridge forms under cold air temperatures (less than -15°C), around neap tide, and during a cessation or even reversal in the prevailing north-northeasterly winds in the strait. However, the bridge in Kane Basin may have failed to form even under atmospheric and oceanic conditions that are favourable for consolidation, indicating the importance of additional factors such as ice thickness (Kirillov et al., 2021). Based on AVHRR satellite data from 1979 to 2019, Vincent (2019) reported on a recent trend, though not confident, towards later formation and earlier breakup of the ice bridge. Additionally, Vincent (2019) found that the ice bridge failed to form only two times during the first two decades of observational records (in 1993 and 1995; Vincent, 2019) and six times during last two decades (in 2007, 2009, 2010, 2017, 2019 and 2022), underscoring a general shortening in the duration of the bridge and pointing to changes in the environmental conditions.

The formation of the Nares Strait ice bridge also contributes to the formation and maintenance of the North Open Water (NOW) polynya. Being the largest recurring latent heat (wind-driven) polynya in the Canadian Arctic (Tamura and Ohshima, 2011; Preußer et al., 2019), the NOW has significant impacts on oceanographic, atmospheric and biological conditions and processes in northern Baffin Bay (Dumont et al., 2010). The high biological productivity supported by NOW is of critical importance to Inuit that hunt and fish in the area (Hastrup et al., 2018; QIA 2020). Beyond the NOW and other latent heat polynyas, there are several polynyas that form within the landfast ice cover of the Canadian Arctic and are at least partly attributed to the sensible heat flux from the ocean (Topham et al., 1983; Smith et al., 1990). Most of these polynyas are formed in the narrows where strong tidal and mean currents cause vigorous vertical mixing and facilitate upward heat transfer from the warm subsurface water at depth (Topham et al., 1983). In addition to these sensible heat polynyas, which are visible at the surface, there are numerous so-called "invisible polynyas" in the Canadian Arctic where the ocean heat flux limits ice growth and leads to an ice cover that is appreciably thinner than the surrounding ice (Melling et al., 2015). Covered with thinner ice, these areas are prone to breaking up earlier and therefore may also facilitate the collapse of landfast ice during summer. So far, the consideration of polynyas in Nares Strait has been mainly limited to the NOW polynya and, to a lesser extent, to the Lincoln Sea polynya, which is located north of the strait (Barber and Massom, 2007). However, it is also known that several sensible heat polynyas form within the stable landfast ice cover within Nares Strait, though they are relatively small and less studied. On the western side of Nares Strait, sensible heat polynyas form near the Bache Peninsula along the eastern side of Ellesmere Island (Schledermann, 1980; Hannah et al., 2009) and seem to be highly biologically productive spots given the presence of prehistoric settlements in the area extending back 2500 to 3000 years (Schledermann, 1978). On the eastern side of Nares Strait there is a sensible heat polynya that forms at Cape Jackson in the central part of the bridge (at ~80° N, Fig. 1). To our best knowledge, this is the northernmost sensible heat polynya on our planet. Although some historical materials and papers mention this polynya (Kane, 1856; Vibe, 1950), its existence has remained hidden from a wide scientific community and there have been no special studies conducted on.

The first written account of the Cape Jackson polynya was found in the report of the Second Grinnell Expedition led by Dr. Elisha Kane, which was looking for Sir John Franklin's expedition. From the words of Mr. Morton and Hans Hendric whose dog sled party reached this area in June 1853, Dr. Kane made the followed record:

"*June 21, Wednesday. They stood to the north at 11.30 p.m., and made for what Morton thought a cape [Cape Jackson], seeing a vacancy between it and the West Land. The ice was good, even, and free from bergs, only two or three being in sight… They*

*reached the opening seen to the westward of the cape by Thursday, 7 a.m.*" (Kane, 1856; p. 284).

Followed by a description of substantially deteriorated ice in the vicinity of the discovered open water:

"*The ice was weak and rotten, and the dogs began to tremble. Proceeding at a brisk rate, they had got upon unsafe ice before they were aware of it. … The only way to induce the terrified, obstinate brutes to get on was for Hans to go to a white-looking spot where the ice was thicker, the soft stuff looking dark; then, calling the dogs coaxingly by name, they would crawl to him on their*

*bellies. So they retreated from place to place, until they reached the firm ice they had quitted. A half-mile brought them to comparatively safe ice, a mile more to good ice again.*".

Several years later, in the middle of May 1861, the expedition of Hayes approached Kennedy Channel along the Ellesmere coast. Although he likely failed to reach the entrance to the channel, the following observation was made close to Cape Frazer:

"*To the north-east the sky was dark and cloudy, and gave evidence of water; and Jensen was not slow to direct my attention to the*

*"water-sky"* (Hayes, 1867; p. 395).

On the other hand, Knud Rasmussen, who stepped over Morton's and Hendric's footprints near Cape Jackson earlier in the season (end of April) of 1917, did not report any open water there (Rasmussen, 1921; p. 61-65).

Although the ocean-derived character of the polynya at Cape Jackson is a matter of fact, the question about heat source remains unanswered because so little is known about the ocean state beneath the ice bridge between its formation and break up. The

consolidation of sea ice modifies the structure of water flow through the strait leading to a shift of southward geostrophic flow from the middle to the western side of the strait (Rabe et al. 2010; Rabe et al. 2012; Shroyer et al. 2015). However, these observations were limited mainly to Kennedy Channel and may not apply to the entire strait, limiting their use in identifying the processes maintaining ice-free conditions at Cape Jackson at times during winter. This study intends to partially fill these gaps by examining the ice-ocean interactions that occur under the bridge during winter, and subsequently examine their influence on the

formation of the polynya at Cape Jackson and other yet unknown invisible polynyas in Nares Strait.

The overarching goal of this paper is to demonstrate an impact of ocean heat on the sea ice in Nares Strait and discuss its possible role in breaking up the ice bridge. The paper is organized as follows. In Section 2, we describe datasets used in this study. The results obtained from the remote sensing data on sea ice in Kane Basin are presented in Sections 3.1 and 3.2. In the next section, we show the results of 1-D simulations of ice growth in Peabody Bay (see location in Fig. 1b). The description of ocean circulation

and water mass structure in Kane Basin obtained with the Finite volumE Sea ice Ocean Model version 2 (FESOM-2) numerical model is given in Section 3.4. The findings are summarized and discussed in Section 4, followed by main conclusions in Section 5.

## 2. Data and methods

### 2.1 Satellite imagery

In this study, we used remote sensing data from different satellites to demonstrate the presence of thinner ice and ice-free polynyas in Nares Strait. First, the true color imagery from the Moderate Resolution Imaging Spectroradiometer (MODIS, Level 1B) and Sentinel-2 (Level-1C) were used to identify the ice-free areas in Nares Strait. MODIS was first launched in 1999 on board the

Terra satellite, with the second sensor launched in 2002 aboard the Aqua satellite. In addition to the daily global composites of the true color band composition (Bands 1, 3 and 4; spatial resolution 250 m), the MODIS brightness temperature ($T_b$, band 31 mid-infrared; spatial resolution 1 km) was also used. $T_b$ is calculated from the top-of-the-atmosphere radiances and shows the relative temperature difference between open water, and thick and thin sea ice. The Sentinel-2A and 2B satellites with high-resolution multispectral optical imager (spatial resolution 10 m) onboard were launched in 2015 and 2017, respectively, as part of the European Space Agency's Copernicus mission. The usage of both optical and infrared products is considerably limited by the presence of clouds. In this research, we used the images obtained during clear-sky conditions.

The daily averaged sea ice brightness temperatures from the 89V GHz channel and snow depths obtained by Advanced Microwave Scanning Radiometer (AMSR2) not interfering with clouds were also used. AMSR2 is carried by the Japan Aerospace Exploration Agency (JAXA) Global Change Observation Mission – Water 1 (GCOM-W1) satellite. The Level-3AMSR2 data is provided by the National Snow & Ice Data Center (NSIDC) at 6.25 km spatial resolution for temperatures (Cavalieri et al, 2014; Meier, et al, 2018) and 12.5 km for snow depths (Tedesco & Jeyaratnam, 2019). Note neither MODIS nor AMSR2 brightness temperatures are indicative of surface temperature alone, but measure the radiance of microwave and mid-infrared radiation that is expressed in units of temperature (K) of an equivalent blackbody. Therefore, brightness temperatures are influenced by a combination of surface temperature, emissivity, and reflectance of the surface. In this study, we used $T_b$ to highlight a temperature contrast between adjacent regions, but did not interpret it as absolute temperatures of the ice/snow surface.

## 2.2. Ice surface elevation data

In addition to satellite imagery, surface elevation data from the recently launched Ice, Cloud and Land Elevation Satellite (ICESat-2, launched in October 2018) were used to characterize the ice surface elevations in Nares Strait. The elevations are acquired by the Advanced Topographic Laser Altimeter System (ATLAS) instrument that counts individual photons in short segments along three pairs of strong and weak beams. For this study we used ATL07/L3A v.3 product that contains the relative elevations for sea ice/snow and open water (Kwok et al, 2020a). The along-track resolution of ATL07 data is variable according to the number of photons returned, but the typical lengths of segments along the three strong and three weak beams are 15 and 60 m, respectively. At a close distance from the polynya at Cape Jackson (up to 20 km), the ice-free polynya area was used as a reference level for an estimation of absolute elevations of sea ice/snow surface heights. The absolute elevations over narrow bridges may also be determined via linear interpolation of sea surface heights measured at the opposite sides of an ice bridge (Babb et al., 2022), however, this approach requires ICESat-2 tracks cross a bridge from edge to edge which does not work for the long and narrow bridges that form in Nares Strait. Therefore, considering the lack of leads and sea surface height references within the bridge, we introduced a new experemental method based on finding the ATL07 along-track anomalies of elevations ($\tilde{h}$). The anomalies are calculated relative to the mean elevation of any ascending or descending track crossing the bridge between 55-76°W and 78.25-82.5°N, and then averaged onto a 1.5x1.5 km grid. This resolution was chosen to ensure that neighboring strong beams, which are separated by ~3 km, were not projected into the same grid cell. Although such resolution increases the noise in the spatial distribution of $\tilde{h}$ (especially in the areas with mobile ice), it highlights the areas with strong elevation gradients that are spatially consistent throughout winter. Even though the accuracy of individual ICESat-2 readings is relatively high (less than 5 cm, Brunt et al., 2019), the accuracy of the averaged anomalies calculated with this method is estimated to not exceed a few millimeters.

Although ATL07 data are adjusted for geoidal/tidal variations and inverted barometer effects, they may still contain unknown uncertainties related to the regional synoptic variability of sea level associated with strong winds (Samelson and Barbour, 2010) and ocean dynamical effects . However, the presence of immobilized ice and relatively short along-track coast to coast distances within Kane Basin suggests that spatial variations of these uncertainties are relatively small at least compared to the calculated

anomalies (more details are given in Section 3.2). Another type of uncertainty related to estimating sea ice thickness from $\tilde{h}$ is connected to the presence of snow and, what is more important, to the generally unknown spatial and temporal variability of its accumulation rate. The existing climatology or estimates of snow depths in the Arctic either do not cover the study area (Warren et al., 1999; Kwok et al., 2020b) or are too coarse to provide a good spatial coverage in Nares Strait (Rostosky et al., 2018; Glissenaar et al., 2021). Using DMSP SSM/I-SSMIS brightness temperatures, Landy et al. (2017) reported >0.3 m mean snow depth in the central Kane Basin by the end of winter. However, as we will show later, this height seems to be overestimated and the more modest snow depth of 19±2 cm in Peabody Bay obtained from mean March-April AMSR2 data (Tedesco & Jeyaratnam, 2019) is thought to be a more reliable estimate of the snow thickness in this area. The effect of the spatial variability of snow depth on the elevation anomalies remains unknown and is not accounted for in this study.

## 2.3. Modelling

The numerical model of opportunity (FESOM2, Danilov et al., 2017; Scholz et al., 2019) was used for deriving high-resolution water dynamics in Nares Strait. The model's sea ice component is discretized on the same unstructured grid as the ocean component. It can faithfully reproduce sea ice observations, including sea ice linear kinematic features when using high model resolution (Koldunov et al., 2019; Wang et al., 2016). We used the same model setup as Wang et al. (2020) with horizontal resolution of 30 km in the global ocean, except for the Arctic Ocean, where the resolution is 1 km. In the vertical, 70 z levels are used. The model was driven by the atmospheric reanalysis fields from JRA55-do v.1.4 (Tsujino et al., 2018). It was initialized from PHC3 ocean state climatology starting from the year 2000 and has only been run for 10 years. Here we analyse the last five model years from 2006 to 2010. Despite using a low resolution JRA55 product that is probably unable to reproduce orographic strengthening of wind within the relatively narrow Nares Strait (Moore and Våge, 2018; Moore, 2021), the structure of flow in Kennedy Channel and in Smith Sound obtained with FESOM2 qualitatively coincides (not shown) with that obtained by Shroyer et al. (2015) who used the Polar MM5 regional atmospheric model, which has a finer horizontal resolution of 6-km. Although the maximum speed of the southward jet in Kennedy Channel in FESOM2 simulations is about 30% lower compared to Shroyer's results, the speeds in Smith Sound are almost the same.

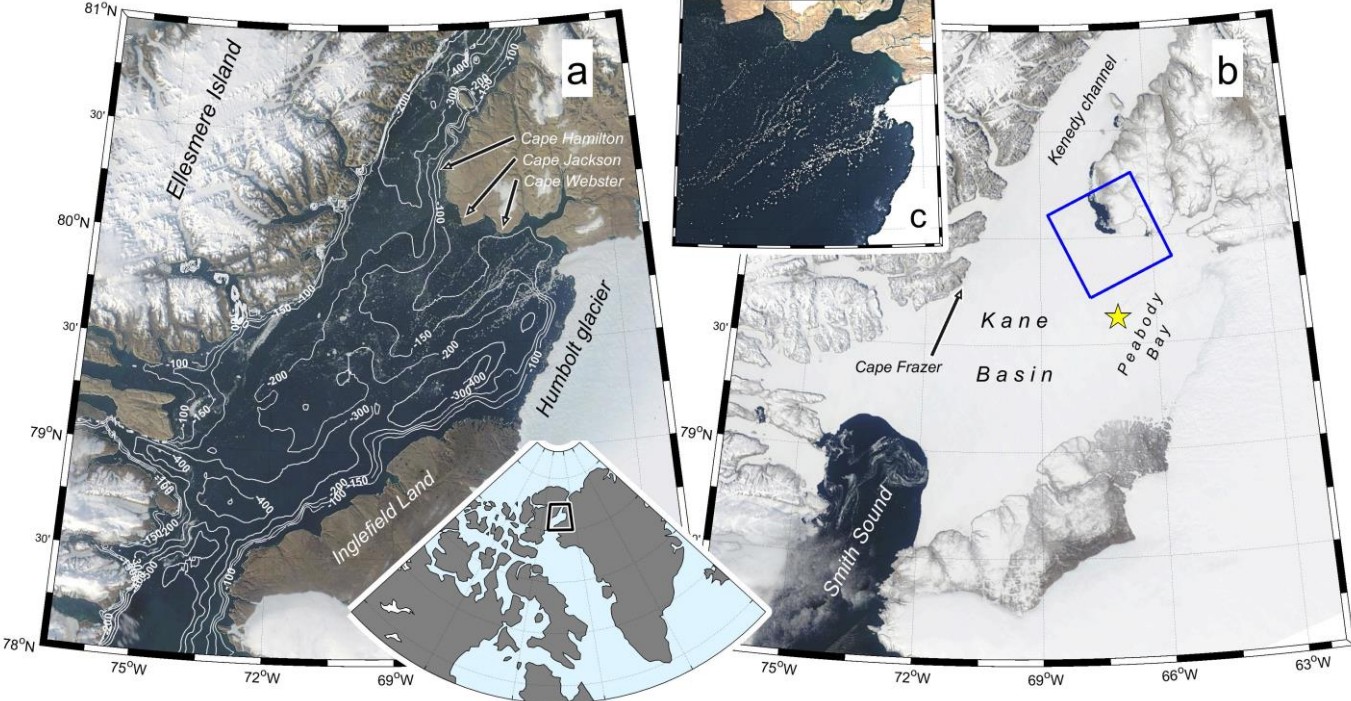

**Figure 1: (a) The bathymetry of the central part of Nares Strait with contours corresponding to 100, 150, 200, 250 and 300 m isobaths. (b) The true color image of the sensible polynya at Cape Jackson on 6 June 2021. The blue square borders a region shown in Figure 2 and the yellow star shows the position of ERA-5 reanalysis data used in the ice growth model. (c) The high-resolution Sentinel-2 image showing the chains of grounded icebergs in the northern Peabody Bay on 8 September 2019.**

To assess the quality of the model in this region, we also compared the thermohaline structure and along-channel current speeds in the Kennedy Channel where an intensive mooring observation program was conducted in 2003-2006 and 2007-2009 (Münchow and Melling, 2008; Münchow, 2016). It was found that both vertical and cross-channel distributions of temperature/salinity and current velocities in FESOM2 simulations generally have good agreement with the mooring records in this region (Münchow & Melling, 2008; Rabe et al., 2010; Münchow, 2016). For instance, the model performs well in reproducing the shift of the southward jet towards the Ellesmere coast (at ~1/4 of the channel width) and also the existence of a countercurrent on the Greenlandic side, although with lower velocities. The mean modelled temperatures and salinities both demonstrate the presence of cross-channel gradients towards Greenland that become stronger at depth, which is in good accordance with observations. Similar to the results reported in Rabe et al. (2010), the model reproduces a fresher surface layer on the western side of the strait. In addition, the model fairly reproduces the uplifting of isohalines and isotherms over the western slope in winter. However, despite a good qualitative correspondence of model results and observations in the Kennedy Channel, a dearth of bathymetry data in Kane Basin (and especially its eastern part) adds some uncertainty to the model results here. A large number of floating and grounded icebergs that originate from the Humboldt glacier may also affect the quality of model simulations in this area.

Although there is no overlap between the ICESat-2/AMSR data (2019-2021) and the available output of the FESOM2 model (2006-2010), we assume that there is very little interannual variability in the circulation under the ice-bridge given that the two major factors controlling water dynamics in this region (the along-strait sea-level gradient and wind) do not vary much interannually. The mean difference in sea level between Alert (data from the Canadian Tides and Water Levels Data Archive) and Thule (University of Hawaii Sea Level Center) from December to June was 1.56±0.04 m in 2006-2010 and 1.59±0.05 m in 2019-2021. The mean difference of sea level pressure between Alert and Thule, that may be used as a fair proxy of wind speed in Nares

Strait (Samelson and Barbour, 2008), also revealed no significant changes from 2006-2010 (3.26±0.22 hPa) to 2019-2021 (3.33±0.23 hPa).

## 2.4. Atmospheric forcing and 1-D ice growth model

The 6-hourly records of 2 m air temperature, wind speed and humidity in Kane Basin were taken from the ERA-5 global reanalysis database (Copernicus Climate Change Service (C3S), Hersbach et al, 2020) and used to run a 1-D sea ice growth model. We used data taken from the central part of Peabody Bay (66.75W, 79.625N, the yellow star in Fig.1) where orographic effects are less pronounced than in the main channel of the strait (Moore, 2021). The sea ice model allows for ocean heat flux and snow accumulation, and may also effectively reproduce flooding, which is associated with a large snow load and leads to the formation

of a snow-ice layer (Kirillov et al., 2015). The model was run with different snow accumulation rates and ocean heat fluxes from 17 December 2019 to 30 April 2020. Two principal scenarios of sea ice growth were simulated: (1) under different but invariable ocean heat fluxes and snow accumulation rates to simulate the ice growth in the central Peabody Bay, and (2) without snow cover directly at Cape Jackson. To avoid computational issues, the snow was added in the model discretely every 15 days instead of a continual accumulation. Although the spatial resolution of ERA5 reanalysis data is lower than what would be ideal for resolving

orographic effects in the narrow and steeply sided Nares Strait (Moore and Våge, 2018; Moore, 2021), the key goal of the ice growth modelling was not obtaining the absolute ice thicknesses, but reproducing the spatial variations of combined ice and snow surface heights in the vicinity of the polynya. From this perspective, even though the modelled absolute ice thicknesses calculated using ERA-5 data may be meteorologically biased, the accuracy of meteorological data seems to have a considerably smaller effect on the investigated spatial differences of ice thickness between the polynya and surrounding landfast ice compared to unknown

snow accumulation rate and possible spatial variations of ocean heat flux.

Air temperatures were additionally used to quantify the annual number of freezing-degree days during winter that is calculated as the sum of the daily degrees below the freezing point of seawater (-1.8 °C). The freezing-degree days (FDD) were used to roughly estimate the thermodynamic growth of sea ice through the empirical relationship connecting FDD and ice thickness. The most often used parameterization of ice growth under average snow conditions in the Arctic was introduced by Lebedev (1938) in a

225 form of $h(m)=0.0133 \times FDD^{0.58}$, although other possible parameterizations also exist (Bilello, 1961) and were considered in Section 3.3.

## 3. Results

### 3.1. Identifying the polynya at Cape Jackson

MODIS imagery confirms that every winter when the ice bridge is formed in the strait since the observations began in 2000, a

230 polynya appears at Cape Jackson late in the season. The snapshot true color images in Fig. 2 show the state of ice cover in vicinity of the cape in May when the air temperatures are still negative with a climatic mean varying form -12 °C (2 May) to -7 °C (20 May; ERA5). However, open water is often evident in MODIS images starting from early March (not shown) when the mean climatic air temperatures are below -25 °C but downwelling shortwave radiation is increasing with the coming of spring. The appearance of open water near Cape Jackson in March is also evident in a few high-resolution Sentinel-2 images obtained in 2020

and 2021 (Fig. 3). Although the configuration of dark spots corresponding to open water or thin sea ice in Fig. 2 varies from year to year, the high persistence in their positioning suggests the existance of a highly persistent ocean heat flux that exceeds the intense loss of heat to the atmosphere. The presence of a sensible heat polynya here is also evident in brightness temperatures in January-

February (MODIS $T_b$, not shown), although polynya signatures are less clear in those images because of their lower spatial resolution. The higher values of $T_b$ in vicinity of Cape Jackson may indicate an ice-free surface, locally thinner ice, locally thinner snow or both of the latter above an invisible polynya.

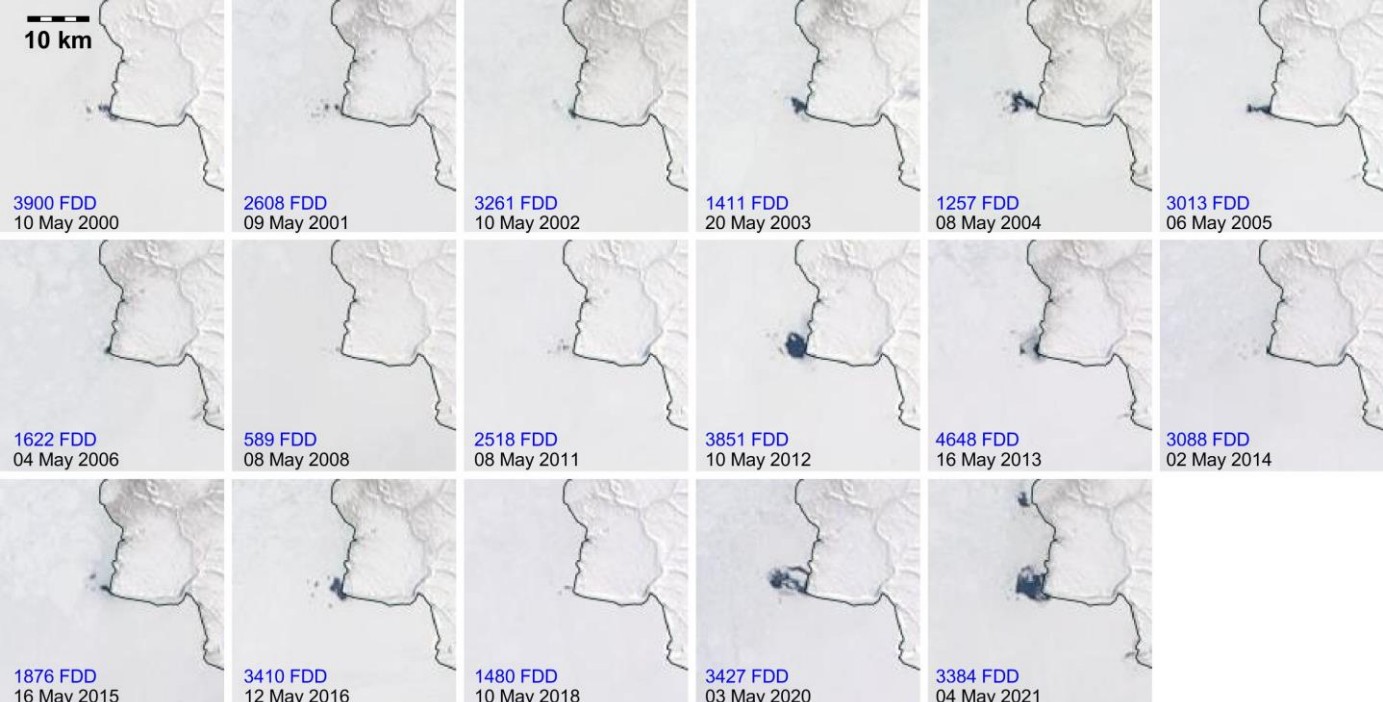

**Figure 2: The presence of sensible heat polynya at Cape Jackson during winters when ice bridge in Nares Strait formed between 2000 and 2021 in MODIS imagery. Blue numbers correspond to the sum of freezing degree days (FDD) from a corresponding date of bridge formation (see Kirillov et al., 2021) to 30 April. Each panel is 37.5 x 37.5 km.**

ICESat-2 transects confirm the presence of thinner ice in the vicinity of the polynya. Fig. 3 shows the elevations adjusted to the sea level in the polynya along selected tracks crossing the polynya area in March-May 2020 and in March-April 2021. Despite the presence of some short-scale variance, all tracks begin to show a decrease in elevation 5 to 15 km away from the polynya. The probability distribution of absolute elevations from the selected scenes shows a modal height of 0.26 m, which corresponds to an estimated modal thickness of 2.42 m (for $\rho_{ice}$ = 915 kg m$^{-3}$, $\rho_{water}$ = 1025 kg m$^{-3}$) under snow-free conditions. This modal thickness, however, may be very sensitive to the accuracy of ICESat-2 measurements and more specifically to the offsets applied to adjust each individual track in Fig. 3 to sea level. An error of 1 cm in offsetting would alter the ice thickness estimate by about 9 cm. Additionally the basic no-snow assumption introduces a much greater error in the estimated ice thickness. If 50% of the 0.26 m surface elevation is attributed to a snow layer with density of 300 kg m$^{-3}$, the resulting ice thickness decreases to 1.56 m, and to 1.14 m if the snow-to-ice ratio above the sea level is 75-25%.

Beyond the central part of the polynya, there is also evidence of patches of thinner ice and open water surrounding that area (e.g. Fig. 3d), suggesting that there is some spatial variability of the ocean heat flux that maintains the polynya. The ICESat-2 tracks also demonstrate irregular spikes and elevated segments, which likely indicate the presence of thicker MYI ice, ridges or icebergs in this region.

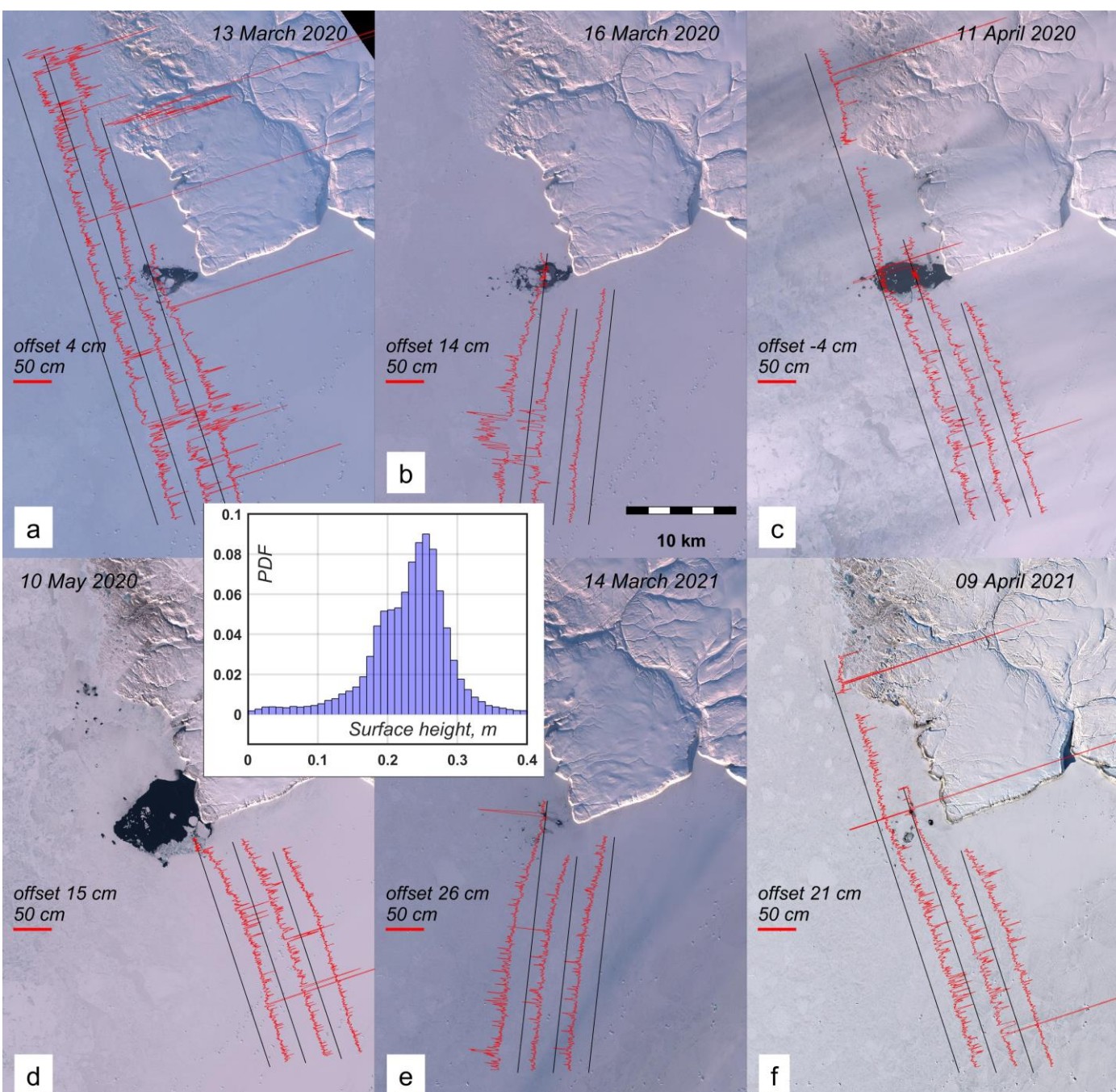

**Figure 3: The ATL07 elevations (red lines) from 3 strong beams overlaid on Sentinel-2 images of the polynya in March-May 2020 and in March-April 2021. The elevations are adjusted to the sea level in the polynya and plotted as absolute deviations from the ground tracks (black lines) in direction normal to the tracks. The open water was used as a reference level to calculate the corresponding offsets for each subset. The red scale bar in each panel corresponds to 50 cm elevation. The inset histogram demonstrates the probability distribution of all heights shown in the panels a-f.**

### 3.2. Basin-wide anomalies of sea ice thicknesses

Expanding from the area around Cape Jackson, Fig. 4 represents the seasonally (January-April) averaged $\tilde{h}$ throughout Nares Strait in 2019, 2020 and 2021. In 2019, the bridge did not form, leaving the main channel of Nares Strait covered with mobile ice throughout winter, though landfast ice still formed across Peabody Bay in the eastern part of Kane Basin (Fig. 4a). The landfast

ice edge represents a pronounced boundary dividing these two areas with different $\tilde{h}$ patterns. In the main channel, the anomalies are highly irregular and form a speckled pattern, whereas the anomalies in Peabody Bay form a consistent pattern with positive anomalies in the southeast and negative anomalies to the northwest. The anomalies in Peabody Bay vary mainly between -5 and +5 cm, although elevated negative anomalies were observed along the northern coast of Peabody Bay during all 3 winters. These elevated anomalies are mainly constrained between Cape Jackson and Cape Webster, although they can be traced further along the coast in both northwestern and eastern directions. The values of $\tilde{h}$ of -15 to -10 cm in this coastal zone are about two times smaller than anomalies found along the individual ICESat-2 tracks at Cape Jackson in March-May (Fig. 3). This discrepancy is partly attributed to the averaging of data from both ascending and descending tracks (ICESat-2 repetition cycle is 90 days) that were used to compute the mean $\tilde{h}$ in each 1.5x1.5 km cell, but it is a combination of both these tracks with different along-track regional means together that seems to result in some smoothing of anomalies presented in Fig. 4. Note, however, that both the patterns and the magnitudes of anomalies are very similar when calculated from only descending (~along the strait) or ascending (~across the strait) tracks (not shown). Another interesting feature observed during winters with a bridge is the presence of persistent negative anomalies of $\tilde{h}$ along the western coast of Nares Strait (Fig. 4b, c). These anomalies are not visually associated with any ice-free polynyas at least until late summer.

A relatively short off-shore extension (about 10 km) of both coastal zones (along the western coast of Nares Strait and along the northern coast of Peabody Bay) eliminates the regional variations of sea level as a factor contributing considerably to the anomalies observed there. For instance, Samelson and Barbour (2008) reported the relatively small spatial gradient of sea-level pressure over the full width of Kane Basin corresponds to about 2 cm of sea level difference with higher levels along the Greenlandic side of the strait. A geostrophic adjustment requires less than 2 cm sea-level drop from Ellesmere Island to Greenland in Kennedy Channel (Münchow et al., 2006). Also, using the tidal gauge records at Alert and at the opposite sides of Smith Sound, Münchow & Melling (2008) reported the across- and along-channel sea-level differences varying in Nares Strait from a few centimetres to about 10 cm, respectively. However, these relatively large differences could be associated with the local dynamical effects as all bottom pressure sensors were deployed in shallow bays not far from the areas covered with mobile ice at Smith Sound and at Alert. The actual sea level gradients below the ice bridge in Nares Strait and their input to the observed ICESat-2 anomalies remain unknown, but are thought to be small compared to the gradients associated with the anomalies observed along the western coast of Nares Strait and at the northern coast of Peabody Bay.

The difference in elevation anomalies between the southeastern and northwestern parts of Peabody Bay is correlated with a similar difference in the observations of $T_b$. The mean January-March AMSR2 brightness temperatures for all three years are shown in Fig. 5 and highlight the presence of warmer (thinner) ice in the northwest compared to colder (thicker) ice in the southeast. April was excluded from analysis because the spatial contrasts in $T_b$ become reduced due to the effect of increasing solar radiation. Additionally, AMSR2 data did not reveal temperature anomalies in the area where negative $\tilde{h}$ anomalies are observed along the coast in western Nares Strait (Fig. 5).

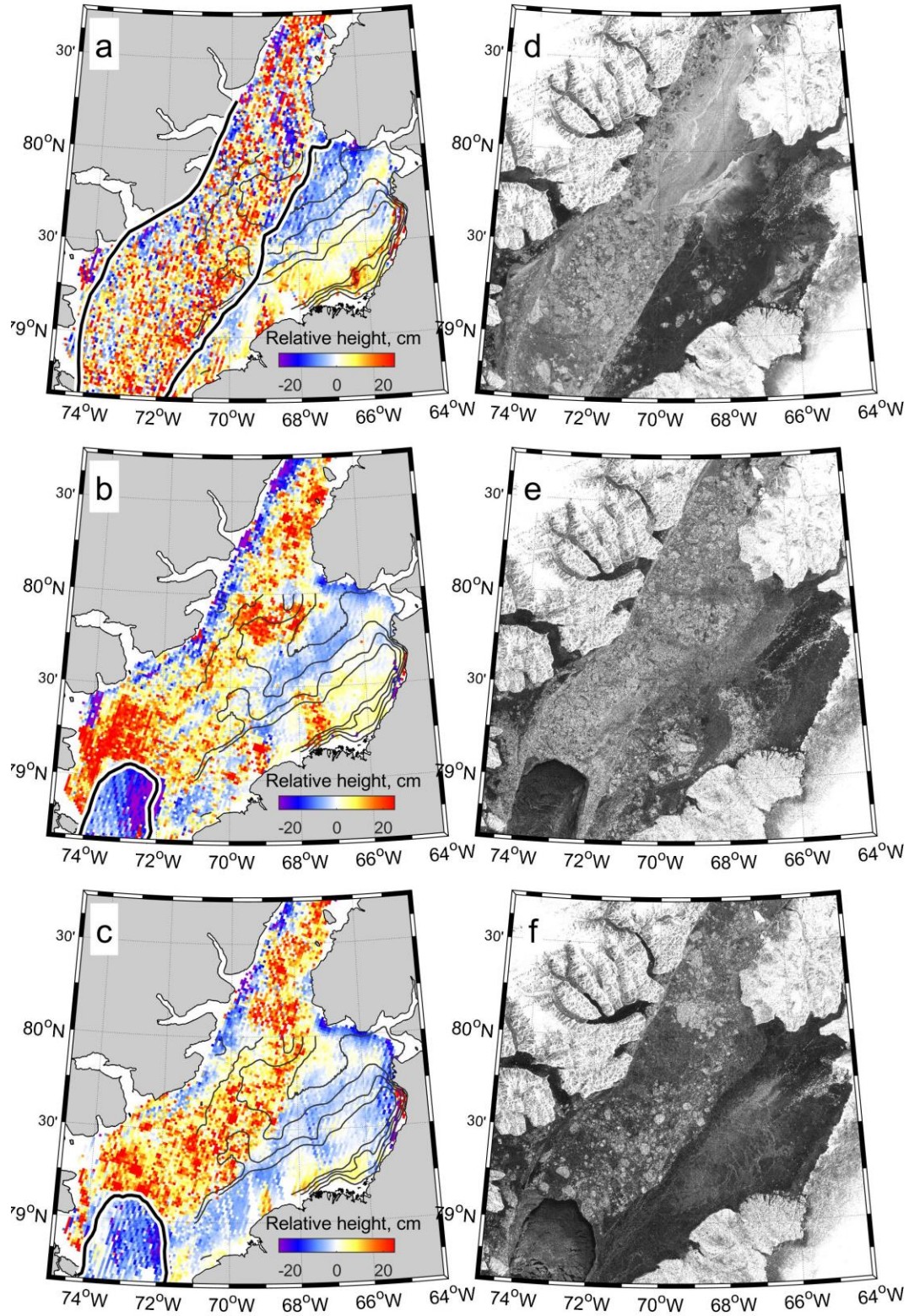

**Figure 4: (a-c)** The along-track anomalies of ATL07 elevations averaged over 1.5x1.5 km cells in January-April 2019 (a), 2020 (b) and 2021 (c). The thick black lines correspond to approximate positions of landfast ice edge during these winters. **(d-e)** Sentinel-1 SAR images from January 16 in the corresponding years.

The microwave Synthetic Aperture Radar (SAR) Sentinel-1 images presented in Fig. 4d-f help to characterize the composition of sea ice presented in Kane Basin and specifically in Peabody Bay during these three winters. It is clear that the sea ice in the bay consists mainly of smooth ice (black areas) with inclusions of ridges characterized by high backscatter and seen as white patches. The spatial distribution of ridges generally corresponds to the regions where elevated values of $\tilde{h}$ were found (e.g. Fig. 4b, e).

Besides the irregular patches corresponding to ridges, the numerous grounded or ice-locked icebergs are also seen in the northeastern as linear white filaments in the vicinity of Humboldt glacier terminus (e.g. Fig. 4f and also Fig. 1c). The generally smooth character of the ice surface suggests that sea ice in the bay is predominantly first year ice that has formed locally through thermodynamics. Contrary to this, the sea ice in the main channel is represented by a mix of relatively thin FYI (corresponding to dark areas) and thicker individual MYI floes (light spots) regardless of the presence of the ice bridge.

The results presented in Sections 3.1 and 3.2 generally support the hypothesis that it is the ocean heat flux that is responsible for formation of ice thickness anomalies in Nares Strait including polynya at Cape Jackson. In the next section, we will roughly estimate the intensity of this flux in northwestern Peabody Bay by calculating the ice growth through numerical simulation with 1-D thermodynamic model.

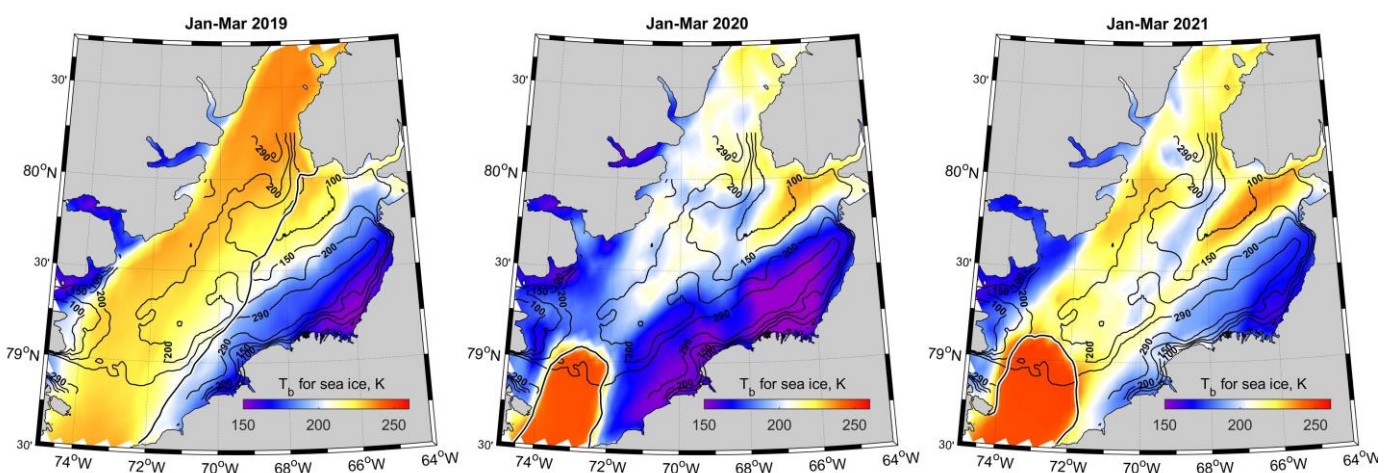

**Figure 5: The mean AMSR2 brightness temperatures in January-March 2019, 2020 and 2021. The 100, 150, 200 and 250 m isobaths are shown in background, whereas the black lines in the right panel corresponds approximate positions of landfast ice edges.**

### 3.3. Modeling of the thermodynamic ice growth in northwestern Peabody Bay

The timing of landfast ice formation in Peabody Bay varies from year to year, but generally occurs in October according to satellite data. The stable landfast ice first forms in the southeastern bay separated from the northwestern part by hundreds of grounded icebergs from Humboldt glacier that form several well-separated chains roughly aligned along isobaths. Most of these icebergs are grounded at the eastern slope of the mid-basin ridge (Fig. 1c) and represent a "glue" that helps new thin ice bind together and stabilize. Although air temperatures in Kane Basin fall below zero in early September, the northwestern part of Peabody Bay, near Cape Jackson, can remain unfrozen until November or December. In winter 2019/20, for instance, this area was ice-free or covered with thin mobile ice until 17 December when the bridge formed. The amount of FDD from that date to the end of April estimated from ERA-5 air temperatures in the central part of Peabody Bay reached 3427, giving an estimated ice thickness of 1.49 m under average snow conditions according to parameterization of Lebedev (1938). However, less snow may result in faster ice growth and greater resulting thicknesses. Using empirical relationship describing the ice growth under negligible amount of snow near

Churchill (Canada) would give 1.86 m of ice at the end of winter (Graystone's formula in Bilello, 1961). Another formula proposed in Bilello (1961) for snow-free ice surface yields ice thickness of 2.00 m which is still smaller than the 2.42 m thickness estimated from ICESat-2 data in the vicinity of Cape Jackson under the no-snow condition. The wide range of these ice thickness estimates underlines the importance of snow depth data for analysis of ICESat-2 surface heights and/or in-situ ice draft measurements. In addition, these empirical relationships do not take into account the ocean heat flux that may play a critical role in ice growth and conditions the existence of sensible heat polynyas.

To investigate the possible joint effect of snow and ocean heat flux on the observed spatial variations of surface heights in the vicinity of Cape Jackson, we applied the 1-D thermodynamic ice growth model with the same parameters as in Kirillov et al. (2015). To simulate the transition of absolute elevation from the ice-free polynya to the distant ice that had a modal surface height of 0.26 m (Fig. 3), we modeled the ice growth at some distance from the polynya towards the central Peabody Bay (Fig. 6b) and at Cape Jackson where the polynya is observed (Fig. 6c). Away from the polynya, we used 4 cm mo$^{-1}$ as the snow accumulation rate to reach a modest snow thickness of 14 cm at the end of winter, which is reasonably close to the 19±2 cm obtained with AMSR2 data for Peabody Bay (not shown). Based on this accumulation rate and no additional heat flux, we estimate the end of winter ice thickness to be 1.3 m (Fig. 6b). Adding in the heat flux from below lowers this thickness and also shifts the timing of maximum ice thickness. For instance, if the flux exceeded ~40 W m$^{-2}$ the maximum ice thickness occurred in February and snow-ice formed after the surface flooded. Under the snow-free conditions and no additional heat flux, the ice thickness at the end of winter can reach about 1.7 m (Fig. 6c). Within the polynya, in order to have open water by end of April the heat flux would need to reach 70 W m$^{-2}$, while a heat flux above 200 W m$^{-2}$ is required to form an ice-free polynya in early March (Fig. 6c).

To investigate a sensitivity of model results to the uncertainties in forcing parameters and/or possible biases of ERA5 data in Peabody Bay, we conducted several model runs with biased air temperature, wind speed, snowfall rate and ocean heat flux for the experiment presented in Fig. 6b. It was found that increasing wind speeds by 1 m s$^{-1}$ increased ice thickness at the end of the season by 4.02 cm, whereas increasing air temperature, snow accumulation rate and ocean heat flux by 1° C, 1 cm mo$^{-1}$ and 1 W m$^{-2}$ reduced ice thickness by 2.45, 7.41 and 3.36 cm, respectively. Using these numbers, we can estimate a relative input of each parameter to the final results. Using 1 m/s, 2 °C, 2 cm/mo and 5 W m$^{-2}$ as the possible biases or uncertainties of the model forcing parameters and taking the impact of ocean heat flux as 100%, we may estimate the relative input of other factors as 88% (snow), 29% (air temp) and 24% (wind speed) that underlines a larger contributing effect of snow and sensible heat flux on the ice growth. The most important part of Fig. 6 is the last panel showing the combined surface height of ice and snow at some distance from the polynya under different combinations of snowfall rate and ocean heat flux. Under the assumption that the additional, presumably sensible, heat flux in the polynya is large enough for keeping it ice-free or covered with very thin ice with a negligibly small freeboard during the entire winter, these heights can be compared to the 0.26 m mode obtained from ICESat-2 data (Fig. 3). A set of model experiments showed that maximum combined surface height at the end of winter of 0.36 m corresponds to 0 W m$^{-2}$ heat flux and a snow accumulation rate of 12 cm mo$^{-1}$ (Fig. 6d, black dash-dotted line). An insulating effect of snow at the higher accumulation rates does not allow ice to grow thick and, therefore, limits its freeboard considerably. However, smaller snowfall rates and modest heat fluxes may give flatter shape of surface heights in the end of growing season that is in a better agreement with the stable pattern of surface heights from March to May shown in Fig. 3. For instance, at 4 cm mo$^{-1}$ and 10 W m$^{-2}$ (solid red line) the modeled heights are around 0.20 m in April.

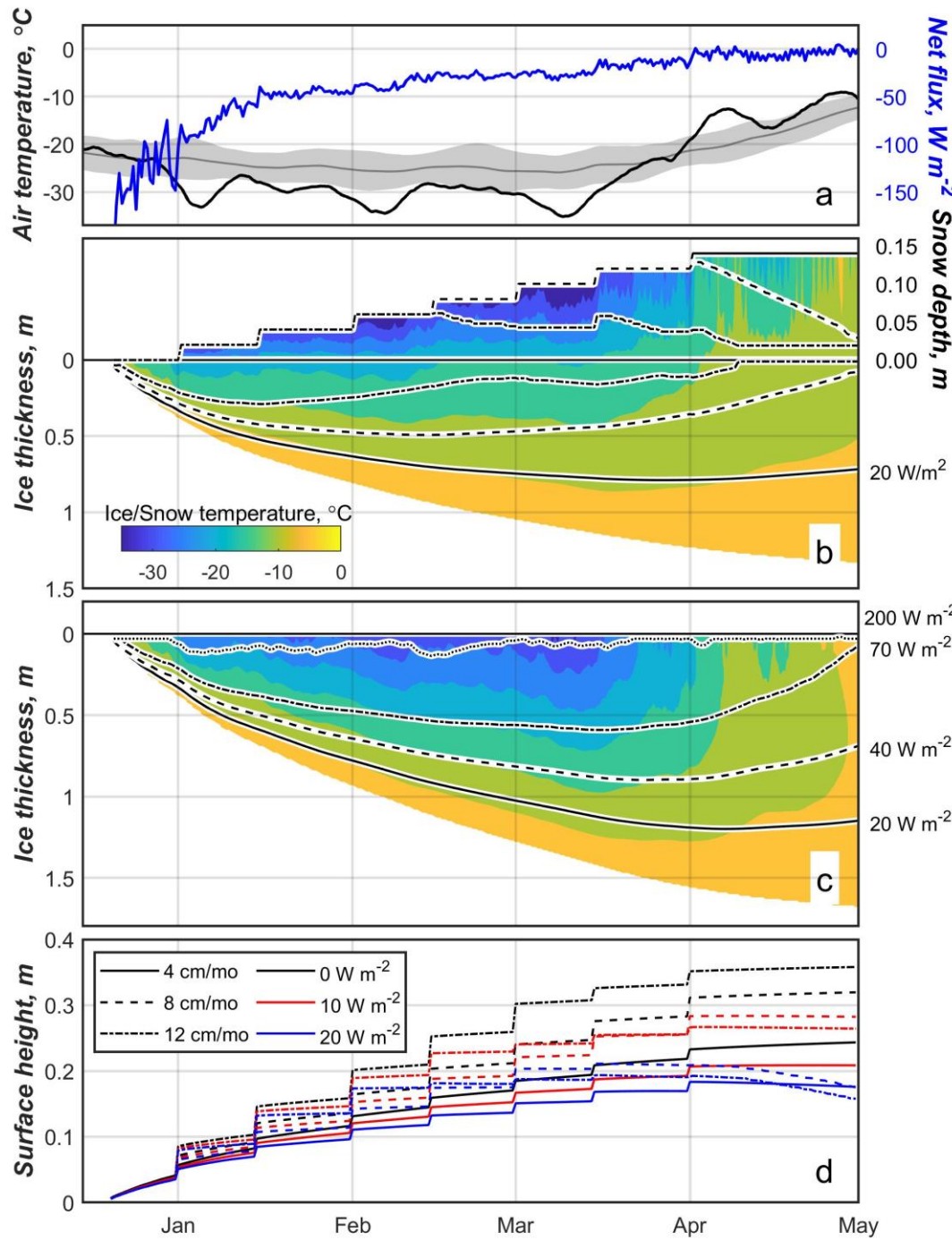

**Figure 6: (a) The 2 m air temperature (black) and the net surface heat flux (blue). Grey line and shading indicate the mean 1979-2020 temperature and standard deviation. (b) The modeled evolution of ice/snow temperature and thickness in the vicinity of polynya under a constant snowfall rate of 4 cm/month (corresponding to 14 cm of snow pack at the end of March). (c) The same, but under no-snow conditions – a proxy of the polynya at Cape Jackson. Black lines in panel (b) and (c) show the evolution of ice and snow thicknesses if an additional heat flux of 20 (solid), 40 (dashed), 70 W m⁻² (dash-doted) or 200 W m⁻² (doted) from below is applied. (d) The calculated difference between the sea surface in ice-free polynya and the combined surface heights of ice and snow within surrounding ice for different snow accumulation rates and ocean heat fluxes.**

## 3.4. Ocean circulation and thermohaline structure in Kane Basin

To understand the physical mechanisms responsible for ice thinning in northwestern Peabody Bay and formation of the polynya at Cape Jackson, we took advantage of the hi-resolution modelling of Nares Strait circulation and thermohaline structure conducted using the FESOM-2 model. According to the model results, the ocean circulation in Nares Strait consists of three principle patterns

(Fig. 7a, b). The first one is the southward flow from the Lincoln Sea through the Robeson and Kennedy Channels and then along

the western coast of Kane Basin (blue streamlines in Fig. 7a-b). This flow occupies the entire water column and consists of 3

distinctive layers: i) cold brackish mixed water within the upper 50-60 m, ii) the relatively warm upper halocline that is observed

at 70-110 m and (iii) the warm underlying layer that is mainly associated with Atlantic Water (AW) which originated in the North

Atlantic and was transported a long way from Fram Strait into the Arctic Ocean and to Northern Greenland (e.g. Melling et al.,

2001). The first two layers are mainly associated with Pacific Water (PW) which comprises more than 50% of the upper 100-150

395    m in Nares Strait (Jones and Eert, 2006; Münchow et al., 2007). The second pattern is associated with the northward-flowing

subsurface current at the eastern side of Smith Sound (red streamlines in Fig. 7b). A branch of this flow recirculates in the southern

Kane Basin and merges with the southward jet, but another branch heads north along the Greenland slope toward Peabody Bay

(Fig. 7b). This current is also associated with a penetration of relatively warm AW, but from northern Baffin Bay. The third

allocated pattern is associated with a cyclonic gyre in Peabody Bay (grey streamlines in Fig. 7a, b).

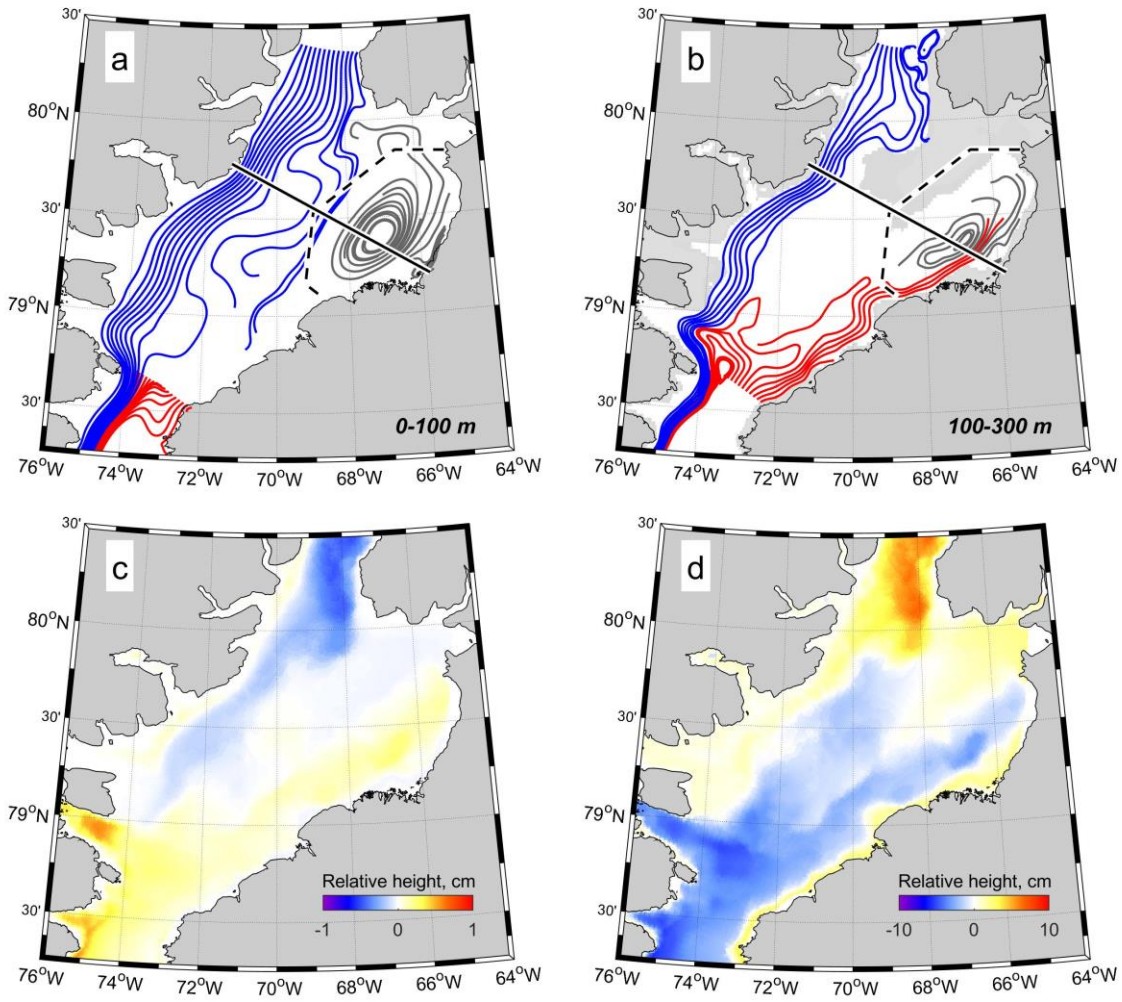

**Figure 7: (a-b) The trajectories of water parcels calculated from the mean 2006-2010 model current velocities averaged within 0-100 m and 100-300 m, respectively. The blue, red and grey lines correspond to three main circulation patterns in the region (see more details in the text). The black dashed line envelopes the region used for vorticity calculation (Figure 8) and the solid line shows the position of transect presented in Figure 9. The shaded area in panel (b) corresponds to the regions shallower than 150 m. The spatial distribution of the mean winter (November-June) thermo- and halosteric sea-level anomalies is shown in (c) and (d), respectively.**

The model does not directly incorporate the formation of landfast ice, but it implicitly reproduces bridge formation via a cessation

of ice motion through the strait. The average speed of ice drift in the main channel becomes less than 2 cm s$^{-1}$ from December to

May (Fig. 8a). In Peabody Bay, drift speeds are generally weaker compared to the main channel and an analog of landfast ice is observed there between November-December and June (Fig. 8a). Although the average duration of the observed ice bridges in Nares Strait (about 5 months, between 20 January and 28 June; Vincent, 2019) is shorter than that predicted with the model (>6 months), we may use December-June model data to describe the ocean circulation and thermohaline structure in the water column beneath the ice bridge. Furthermore, the model fails to reproduce no-bridge situations for winters 2006/07, 2008/09 and 2009/10. This discrepancy could be attributed to a critical impact of rheology and physical characteristics of the ice on bridge formation and the inability of the model to predict ice consolidation from a simple set of favourable atmospheric and oceanic conditions (Kirillov et al., 2021). However, considering the main goal of this study, we can disregard this failure and use the model-derived parameters during winter to describe the state of the ocean beneath the ice bridge.

The cessation of drift in winter leads to a considerable modification of the ocean circulation in Kane Basin. The virtual presence of the ice bridge causes a shift of the southward jet core from the surface to approximately 100 m (Fig. 9a, b). Such behaviour is in agreement with observations reported in Rabe et al. (2012) and results of other models (Shroyer et al., 2015). In Peabody Bay, the seasonal change of circulation is more pronounced. The cyclonic gyre occupying the entire water column within this region in summer weakens significantly and even reverses to anticyclonic circulation at the surface during winter (Fig. 9a, b). Although the modeled sea ice in the bay becomes immobilized in November, the summer gyre remains strong until December (Fig. 8b).

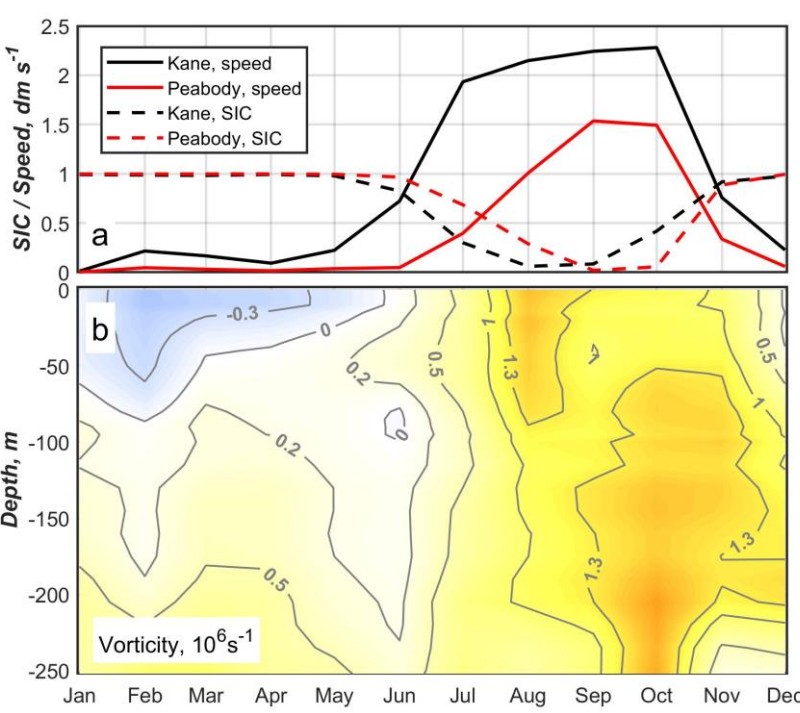

**Figure 8: (a) The mean monthly SIC (dashed lines) and drift speeds (solid lines) within the main channel of Kane Basin (black) and in Peabody Bay (red). (b) The seasonal evolution of mean 2006-2010 vorticity within Peabody Bay. Both datasets are calculated from FESOM data.**

The seasonal changes in temperature and salinity in Kane Basin are mainly observed at shallow depths above 100 m. Below this depth, both parameters can be considered stable with no significant seasonal changes (Fig. 9c, d). The AW penetrating to the basin from the north is approximately 0.2-0.3 °C warmer compared to the southern branch from Baffin Bay (+0.12 vs -0.15 °C). This difference is also evident from historic observational data, but the confidence of those measurements is not very high because of a high spatio-temporal irregularity of observational data. All temperature profiles in Peabody Bay were obtained during summer and

only a few measurements were made concurrently with observations in the main channel. Since a key intention of this research was to bring up the phenomenon of the sensible heat polynya in the middle of the ice bridge in Nares Strait and to discuss potential driving mechanisms, we limited the analysis of thermohaline state of water in the strait by model outputs that agree reasonably well with the observational data. A more detailed description of Kane Basin hydrography based on in-situ measurements can be found in Moynihan (1972), Sadler (1976), Bourke et al. (1989), Jones and Eert (2004), Münchow et al. (2007), Rabe et al. (2010), and Münchow et al. (2011). The most up-to-date status of temperature measurements in Kane Basin can be found in Rignot et al. (2020) where the authors used historical temperature profiles to discuss the oceanic influence on the retreat of the Humboldt glacier.

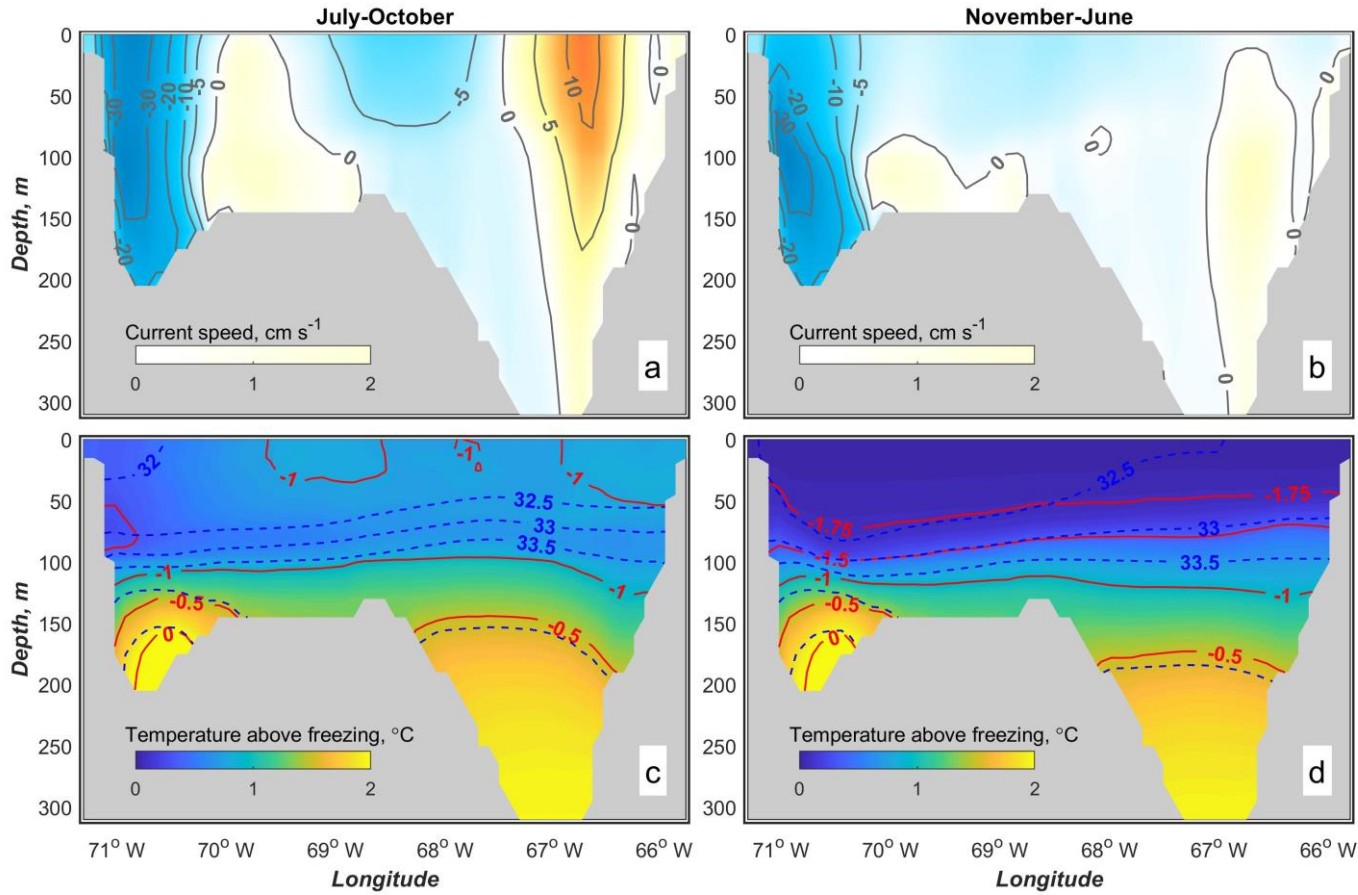

**Figure 9: The modeled mean 2006-2010 summer (mobile ice) and winter (ice bridge) cross-sectional current speed (a, b) and water temperatures and salinities (c, d) at the transect across the central Kane Basin. Positive speeds correspond to northward flow.**

While analysing ATL07 data, it is also important to know if horizontal variations of temperature and salinity could be responsible for spatial pattern of the observed elevation anomalies via steric effects that are not taken into account in ICESat-2 products. The spatial anomalies of thermosteric and halosteric sea level heights in Nares Strait were calculated from FESOM-2 temperature and salinity data similarly to the approach used in Volkov et al. (2013). It was found that temperature has predictably low impact on sea level in this area. The thermosteric anomalies caused by the horizontal variations of water temperature in Nares Strait do not exceed ±1 cm, but in Peabody Bay they are even smaller (Fig. 7c). The halosteric anomalies are generally an order of magnitude higher, but the difference between northwestern and southeastern Peabody Bay is also small and accounts for 3-4 cm only (Fig. 7d).

## 4. Discussion

### 4.1. The formation of ice thickness anomalies in Peabody Bay and polynya at Cape Jackson

The sea ice in Peabody Bay during winter is mainly represented by a smooth thermodynamically grown landfast ice cover with sporadic inclusions of ridges in the southern part of the bay (Fig. 4d-f). The landfast ice typically starts to form in the bay in October, but in the northwestern bay, it remains unstable until ice bridge formation in December or even later dates in some years. As a result, at the start of winter the area in the vicinity of Cape Jackson is either covered with a thin mobile ice or remains ice-free. This anomaly further develops into a visible, or 'invisible' (Melling et al, 2015), polynya after the formation of the ice bridge in the southern Kane Basin and the immobilization of ice in the main channel.

According to altimetry data along the individual ICESat-2 tracks crossing the polynya in 2020 and 2021, the modal height of snow-covered sea ice surrounding the polynya was 0.26 m above sea level (Fig. 3). The analysis of the spatial anomalies of ICESat-2 elevations ($\tilde{h}$) showed that maximal negative anomalies of 0.10-0.15 m are observed along the entire northern coast of Peabody Bay including the polynya area (Fig. 4a-c). Over the rest of the bay, $\tilde{h}$ varies within a relatively modest range between -0.05 and 0.05 m, but it forms the apparent pattern with mainly negative anomalies in the northwestern and positive anomalies in the southeastern parts of the bay. This pattern was highly pronounced in 2019 and 2020, but less evident in 2021 (Fig. 4a-c) which could be associated with an altered ocean heat flux limiting ice growth in certain regions. The MODIS brightness temperatures, $T_b$, shown in Fig. 10 generally support the idea that the thermal state of the surface water in Nares Strait may vary considerably. In December 2019 (Fig. 10b), for example, the high $T_b$ conditioned the ice-free (or covered with thin ice) area in the northwestern part of Peabody Bay and at the eastern side of Kennedy Channel. The pattern of higher temperatures in this area resembles a plume that extends from the landfast ice edge towards the entrance of Kennedy Channel. Although the signatures of surface water with different temperatures in Kennedy Channel can also be traced in theleads within the mobile sea ice in December 2018 and 2020, $T_b$ was observed to generally be lower and may indicate reduced ocean heat transport towards the surface from below during those years. Of course, the spatial difference in temperatures (within leads) presented in these early winter snapshot images cannot explain the seasonally averaged elevation anomalies shown in Fig. 4. However, these differences demonstrate that there may be significant variability in the sensible heat flux at the ocean surface around the time when sea ice begins to form.

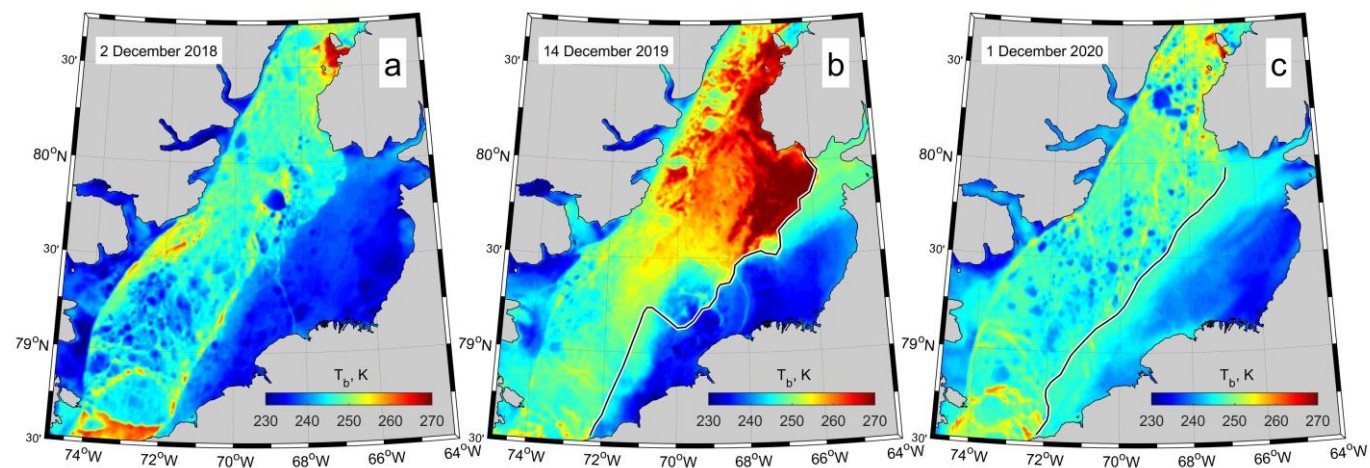

**Figure 10: MODIS clear-sky brightness temperatures on (a) 02 December 2018, (b) 14 December 2019 and (c) 01 December 2020. Images (b) and (c) show $T_b$ several days prior to ice bridge formations in both years. The black lines indicate approximate positions of the landfast ice edge at the date of each imagery.**

The steric anomalies in Peabody Bay are several times smaller than the observed anomalies of elevations (2-3 cm vs ~10 cm; Fig. 4a-c and Fig. 7d, respectively) and, therefore, seem to have a small effect on $\tilde{h}$. In fact, extraction of mean steric anomalies from $\tilde{h}$ would even increase the contrast of $\tilde{h}$ between the northwestern and southeastern bay because of their generally opposite signs in these two regions. Although it strengthens the confidence of the observed pattern in $\tilde{h}$, it is possible that an unknown spatial distribution of snow may considerably affect the magnitude of anomalies and our suggestions about the ocean heat impact on ice thicknesses at least away from Cape Jackson and the northern coast of Peabody Bay where stronger much higher anomalies are found (Fig. 4). Strikingly, to our knowledge, the only published values on in-situ measurement of snow and ice thickness at the end of winter was found in Dr. Kane's report and related to the southeastern part of Peabody Bay. The snow and ice thicknesses measured during Dr. Kane's expedition were "*knee deep*" and "*seven feet five inches* [2.25 m]." (Kane, 1856; p.281). All available contemporary satellite-derived snow depth products suffer from missing data, low spatial resolution and large uncertainties limiting their practical usage for regional studies.

To investigate the effect of uncertain snow depths and ocean heat flux on $\tilde{h}$ in the vicinity of the polynya we used a 1-D sea ice growth model. Several simulations with different snow accumulation rates and ocean heat fluxes were run to find an optimal combination of these parameters to match the observed modal surface height of 0.26 m near the Cape Jackson polynya (Fig. 3). These simulations were made under the consideration that the polynya is ice-free. It was found that keeping the polynya open in early March requires a relatively large (>200 W m$^{-2}$) additional, presumably sensible, ocean heat flux to compensate the heat loss to the atmosphere. This amount was obtained from the model forced by atmospheric conditions in the central part of Peabody Bay. The polynya at Cape Jackson is usually ice-covered (invisible) during most of winter, starts opening in early March and reaches approximately 3-4 km in size by the end of April. Considering that the model covers the period from December to April, we may suggest that a shift in radiation balance caused by the presence of moister air over open water within the polynya mainly occurs at the end of winter and using ERA-5 data from the central Peabody Bay results in a relatively small overestimation of the sensible heat flux needed to keep this polynya nearly ice-free. From our point of view, the more considerable effect may be the mechanical removal of frazil and nilas ice at the end of winter when the presence of open water results in more intensive formation of ice crystals that may be carried away by advection (e.g. Smith et al., 1990). The full loss of heat to the atmosphere in this case is not entirely compensated by the sensible heat flux from below, but partly explained by the latent heat of ice formation. Although it is difficult to estimate an impact of frazil ice removal and shift of atmosphere boundary layer characteristics on the ice balance without specialized in-situ measurements, we suppose that the reported 200 W m$^{-2}$ as well as 70 W m$^{-2}$ heat fluxes may not be entirely associated with the sensible heat and, therefore, overestimated. Note also that another possible mechanism of ice loss through breakage and removal of relatively thick new ice floes associated with strong tidal currents and winds (Topham et al., 1983) is not thought to play a considerable role in the polynya at Cape Jackson. The analysis of daily Sentinel-2 images in April-May 2020 (accessible through Sentinel Playground hub operated by Sinergise Laboratory for geographical information systems, www.sinergise.com/en) revealed that the individual ice floes idle in the center of the polynya most of the time, instead of clustering near edges as they usually do in the Dundas polynya studied by Topham et al. (1983). It implies that tidal currents at Cape Jackson are not thought to be strong enough to initiate breakage and removal of new ice from the polynya here.

The different combinations of the snow accumulation rates and ocean heat fluxes give a wide range of modelled heights of combined ice and snow away of the polynya (Fig. 6d). However, the best approximations of the target surface height of 0.26 m were obtained for the heat fluxes between 10-20 W m$^{-2}$ and modest snow accumulation rate of 4-8 cm mo$^{-1}$ that corresponds to 14-28 cm of snow accumulated by the end of winter. Although there is no confident regional data on snow accumulation rate in

Peabody Bay, this amount looks feasible as it envelops 14-24 cm snow depths (5-95% percentile range) in March-April obtained
from daily AMSR2 data (not shown). The larger snow accumulation rates and heat fluxes lead to eventual flooding and drop of
elevations at the end of winter, whereas smaller rates and fluxes resulted in heights increasing with time that are not evident in the
altimetry data (Fig. 3). Even though the model could reproduce the observed difference of surface height within the landfast ice
around the polynya reasonably well, the obtained results should be considered only as an approximation because steady snow
accumulation rates and heat fluxes were used, which is likely not representative of their true change in time. A temporal variability
of heat transport into the surface layer near Cape Jackson, for instance, can be expected based on the observed interannual variations
in brightness temperatures (Fig. 10) and in the changes of cyclonic circulation within Peabody Bay throughout winter (Fig. 8).
Even though the obtained ocean heat flux exceeding 200 W m$^{-2}$ or even 70 W m$^{-2}$ within the polynya at Cape Jackson seems to be
overestimated, it is clear that ocean sensible heat plays a principal role in the maintenance of this polynya and in limiting the ice
growth along the northern coast of Peabody Bay. In winter, this sensible heat flux is associated with the warm AW penetrating
Kane Basin from northern Baffin Bay and spreading northwards along the eastern coast (Fig. 7 and 9). According to FESOM-2
simulations, water temperatures in Peabody Bay start to increase from about 60 m and reach near-bottom maximum of -0.15 °C or
~1.75 °C above freezing (Fig. 9d). Although there is no available data on the fractional composition of water masses in Peabody
Bay, we suggest that the upper thermocline may contain a fraction of PW transported southward through Nares Strait. In the
southern Kennedy Channel, the fraction of PW exceeds 50% above 110 m (Jones and Eert, 2006; Münchow et al., 2007), while
further south, this water may be partly mixed at depth with the northward coastal flow and transported to northern Peabody Bay as
part of the cyclonic gyre (Fig. 7a). Regardless of its origin, the presence of water with temperatures above the freezing point below
60 m is thought to be the source of the sensible heat that limits sea ice growth along the northern coast of the bay. This heat is
brought to the surface by upwelling that is associated with either tidal flow or the mean circulation. Although the recurrent tidally
driven upwelling and vigorous vertical mixing is a well-known process leading to the formation of visible and invisible polynyas
in the numerous narrows of the Canadian Arctic Archipelago (Topham et al., 1983; Smith et al., 1990; Hannah et al., 2009), the
analysis of daily Sentinel-2 images supports the suggestion that tidal currents at Cape Jackson seem to be considerably lower than
in other polynyas. Tidal currents may still contribute to uplifting the upper thermocline water closer to the surface and cause an
enhanced vertical heat transfer towards the surface near prominent headlands such as Cape Jackson or Cape Webster, but the
presence of the 50-km long coastal zone of thinner ice implies some contribution of upwelling associated with a horizontal
convergence of the mean flow. Although the dearth of bathymetry data and limitations of FESOM-2 make a confident evaluation
of this mechanism difficult, the regional circulation generally supports the existence of a convergence zone in the northern part of
Peabody Bay where northward flow turns westward (Fig. 7a). An additional factor facilitating vertical heat exchange in the
northwestern Bay may be associated with the presence of hundreds of icebergs (originating from Humboldt glacier) grounded at
the eastern flank of the mid-basin ridge in Peabody Bay. In satellite images, these icebergs are seen to form well-separated parallel
chains oriented from NE to SE (Fig. 1c) that may act as a steering grid for a thorough water flow. Although the mean water
dynamics in the bay during winter is relatively weak compared to the main channel (Fig. 9b), the strong semidiurnal tidal current
may provide a necessary kinetic energy for generating shear instabilities near hundreds of grounded icebergs.

**4.2. The formation of ice thickness anomalies along the western coast of Nares Strait**

The along-track elevation anomalies in Nares Strait shown in Fig. 4b and 4c also revealed a narrow (about 10 km wide) wedge of
negative anomalies spreading over 200 km along the western side of Nares Strait from Cape Frazer at the south to Hall Basin at

the north. At first glance, these observations contradict Ryan and Münchow (2017) who reported on the presence of thicker ice near Ellesmere Island (mean draft of 1.33 m) compared to the Greenlandic side (0.88 m) in Kenney Channel, based on upward-looking under-ice sonar data. Although Ryan and Münchow's results are consistent with an Ekman-layer response of the surface ocean and sea ice to local winds from the north, the authors considered only the periods of mobile sea ice when the potential impact of ocean heat on individual ice floes is relatively small due to their fast passage through the channel. However, a persistently enhanced ocean heat flux (even if small) beneath the ice bridge may play a great role in restricting ice growth near Ellesmere Island during winter.

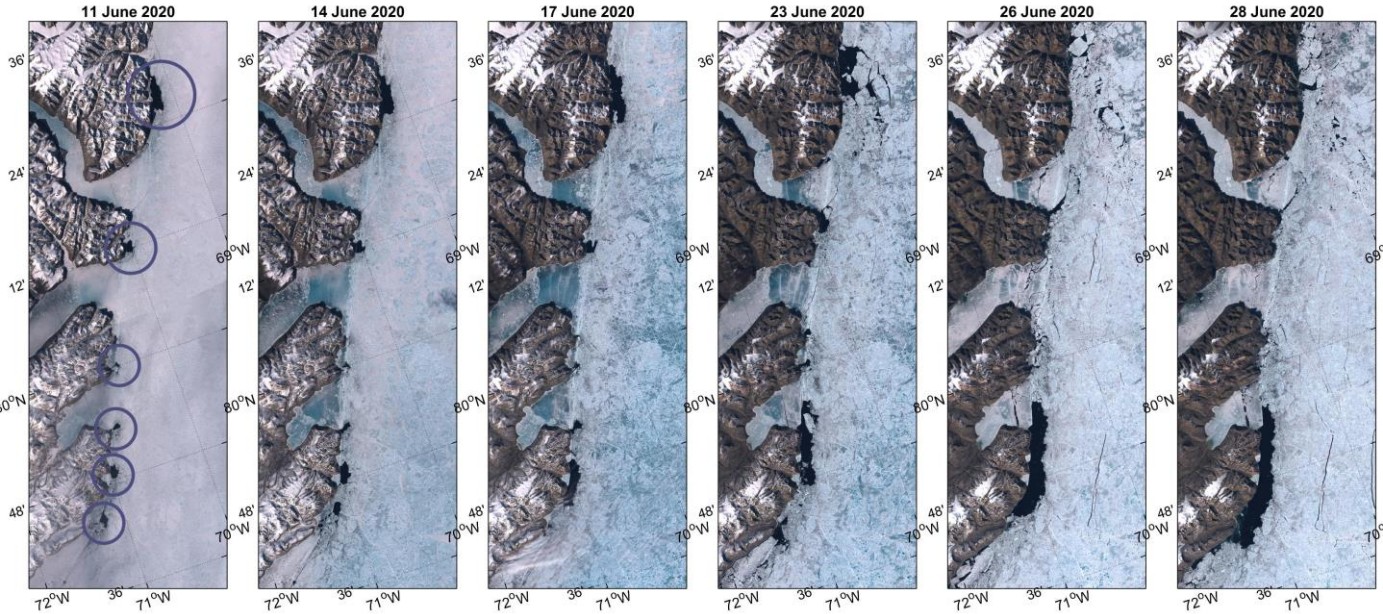

**Figure 11: The series of Sentinel-2 images showing the evolution of polynyas along western coast of Kane Basin from the first appearance until the start of bridge collapse in June 2020. The circles in the left panel indicates the regions where open water polynyas first appear.**

Although the ice elevation anomalies along Ellesmere Island are comparable or even exceed those observed along the northern coast of Peabody Bay (Fig. 4b, c), they are not accompanied by higher brightness temperatures (Fig. 5b, c). The absence of the temperature contrast can be attributed to a generally thicker ice cover that contains a large portion of MYI, whereas the ice in Peabody Bay mainly consists of FYI, or the ice may be covered with less snow (or a combination of these two factors). The suggestion of a stronger ocean heat flux along the western side of Nares Strait is also supported by the fact that chains of sensible heat polynyas form at the end of spring, shortly before the ice bridge collapses (Fig. 11). In 2020, for example, the western polynyas started to open in early June, whereas the landfast ice in Kane Basin remained solid (except the polynya at Cape Jackson) until the bridge broke up at the very end of June.

Similar to Peabody Bay, the oceanic heat flux needed for limiting sea ice growth along the western side of Nares Strait is associated with warm water at depths below 70-80 m (Fig. 9d). The upper thermocline layer in the main channel is mainly comprised of relatively cold PW, whereas the considerably warmer AW prevails at depths below 150 m (Jones et al., 2003; Jones and Eert, 2006; Münchow et al., 2007). The baroclinic adjustment of the ocean to the intensification of the southward current in winter induces upwelling above the core of southward undercurrent that shifts the upper thermocline closer to the surface (Rabe et al., 2012; Shroyer et al., 2017). As a result, water above the freezing point can be found starting from 30-40 m depth near the Ellesmere coast (Fig. 9d) forming favourable conditions for a larger heat transport to the bottom of sea ice here. Unfortunately, the lack of in-situ measurements does not allow us to quantify the vertical oceanic heat fluxes into the western Nares Strait polynyas, but it is likely

that the barotropic semidiurnal tide, with magnitudes comparable to the speed of the mean southward flow (Münchow, 2016; Davis et al., 2019), greatly affects their intensity. Transformation of these currents over steep topography generates a baroclinic semidiurnal tidal wave that may considerably enhance vertical mixing in the water column through benthic stresses and shear instabilities (Davis et al., 2019). From this perspective, the fact that most of the western polynyas first appear near prominent headlands (Fig. 11) generally support the idea that the enhanced heat fluxes along the Ellesmere coast are attributed to the topographically controlled instabilities associated with the mean current and reversible tidal flow. Another mechanism that may enhance the heat flux in the area is associated with the sub-ice turbulence generated by interaction of very rough under-ice topography in the channel and tidal flow (Ryan and Münchow, 2017) that is expected to be accelerated around headlands. In combination with upwelling of the upper thermocline water along the western coast in winter (Shroyer et al., 2017), this mechanism may be considered as a key factor resulting in an enhanced vertical heat flux towards the bottom of sea ice along the Ellesmere coast.

Despite the extensive evidence from true-color satellite imagery, a more detailed analysis of polynyas in western Nares Strait remains difficult. A relatively short cross-shore extension of the polynyas and the short period of their existence prevents us from using the ICESat-2 tracks in a similar way to what was done for the polynya at Cape Jackson (Fig. 3). The western polynyas are simply bypassed by the relatively sparse along-track ATL07 data with their low repeat rate. Without a reference level, we were furthermore not able to use the ice growth model for estimation of absolute ice/snow surface elevations within the wedge of negative anomalies along Ellesmere Island similarly to how it was done in Section 3.3.

### 4.3. The inferred role of thinner ice in Kane Basin in ice bridge break-up

The break-up of the ice bridge in Nares Strait is a complex process that may be determined by a number of factors (Vincent & Marsden, 2001; Kirillov et al., 2021). A certain combination of wind forcing, spring-neap tidal cycle and ice type composition is believed to determine the timing of ice bridge break-up, similarly to the bridge formation process (Kirillov et al., 2021). Based on the results presented here, we suggest that thinner coastal ice, formed under conditions of enhanced oceanic heat flux, weakens the cohesion of landfast ice against the shoreline in Kane Basin. We can further speculate that such weakening may facilitate an earlier ice bridge break-up (comparing to a supposed no-polynyas situation) as it leads to formation of patches of mobile ice in the middle of the ice bridge in Kane Basin. While shifting around, this ice may gain some kinetic energy from wind and tide and eventually result in additional dynamical load on the parts of the bridge that still remain in place but are thermally deteriorated by this time of year. More important, however, is that the edge of these patches does not provide a necessary structural support (as the arch in Smith Sound does) and that mechanical stresses cause an inward collapse of the ice bridge. This process can be seen as a propagation of new fractures outwards from the polynya area. Similar fracturing, which results in a decohesion of the landfast ice in the channel from the coast, was reported for simulations of the ice bridge collapse in Plante et al. (2020). For example, in 2020 the polynya was generally well-constrained during March-May (Fig. 12). The polynya started to expand in late May, reached its maximum size in mid-June, and the surrounding ice cover broke up around 22 June (Fig. 12). This partial break-up of the landfast ice surrounding the polynya appeared to initiate further fracturing of the ice cover in the main channel (Fig. 13). On 26 June, the still consolidated landfast ice in the main channel lost its cohesion with the western coastline and began to shift slowly southward. This movement began while the bridge was still in place, but large pieces of the ice bridge began breaking apart along the southern arch. A complete collapse of the ice bridge occurred a few days later around 1 July 2020.

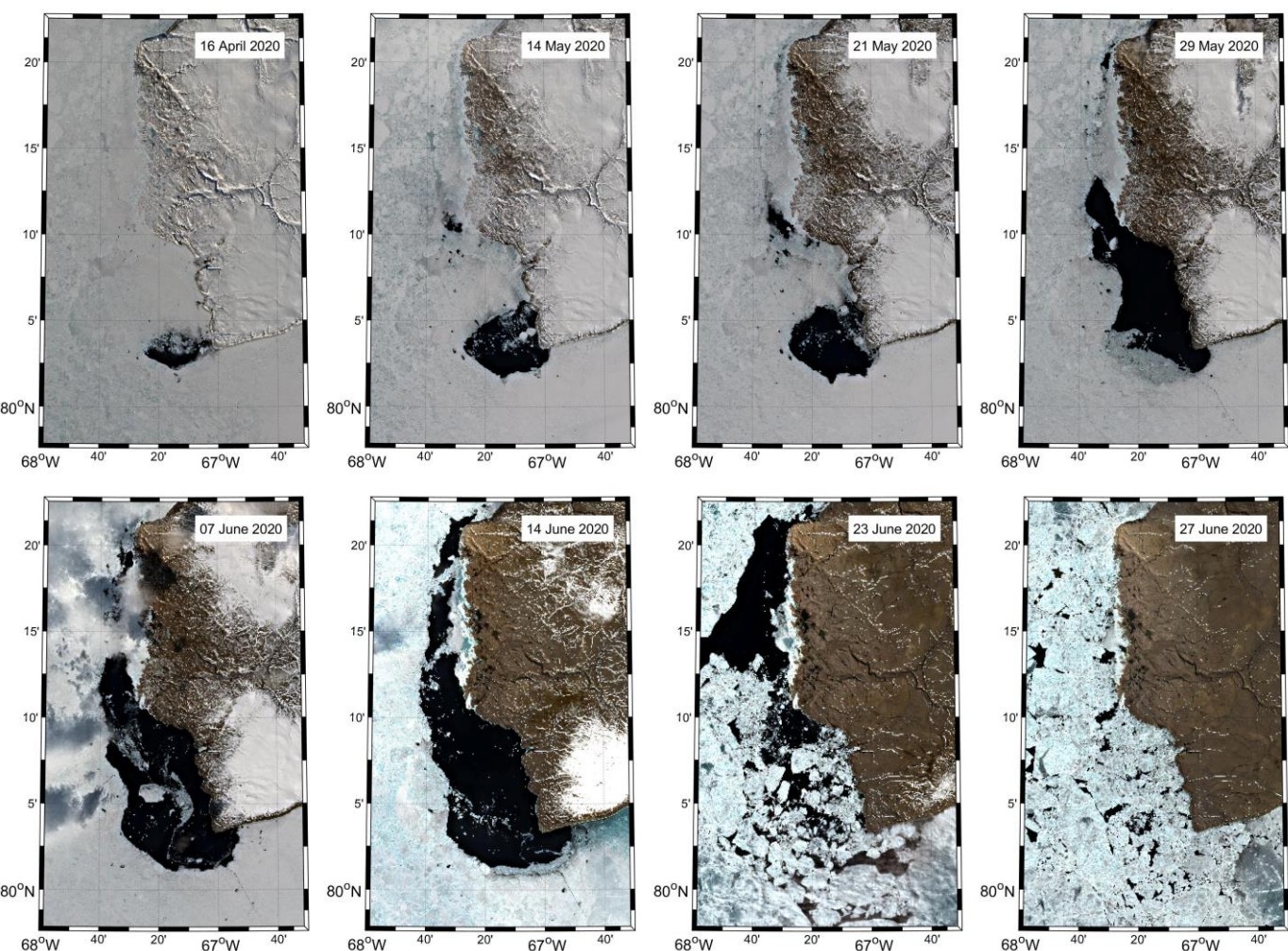

**Figure 12: The evolution of polynya at Cape Jackson from April to June 2020 from Sentinel-2 data.**

Although the satellite images presented in Fig. 13 and results of the numerical modelling (Plante et al., 2020) generally support our suggestion that visible and invisible (i.e., thin ice areas) polynyas may facilitate ice bridge break-up in Nares Strait, this suggestion remains speculative without more detailed research. However, we would like to emphasize the observations that the first movements of the immobilized ice cover occurred in areas with negative ice thickness anomalies during winter and where polynyas are observed. The critical role of the ocean heat flux in formation of these areas is underlined by the existence of the ice-

free polynya in the center of the ice bridge from as early as March. The most obvious source of oceanic heat available to maintain these polynyas along the eastern and western sides of Nares Strait are warm subsurface water of Atlantic origin and relatively colder upper thermocline water of Pacific origin, respectively. In terms of Peabody Bay, AW in front of the nearby Humboldt glacier has warmed by 0.9 °C since 1961 (Rignot et al., 2021), although this trend is based on fairly sparse summer data in the area and is therefore highly uncertain. However, based on a more consistent, albeit much shorter time series of mooring data obtained

in Kennedy Channel in 2007-2009, Münchow et al. (2011) reported a statistically significant warming in the southward branch of AW of 0.027 °C/year. There is also evidence that AW temperatures increased in the Lincoln Sea between 2003-2011 and the 1990s (de Steur et al., 2013). Through the mechanisms we have described here, warming AW in Nares Strait exerts a greater influence on ice growth and therefore the stability of the ice bridge. In fact, coincident to the warming of AW in and around Nares Strait has

been a recent tendency for the duration of the ice bridge to become shorter, or for the bridge to not form at all, an event that has now happened during six of the last 15 years (i.e., 2007, 2009, 2010, 2017, 2019 and 2022). Continued warming of AW as a result of climate change will continue to affect the formation and stability of the ice bridge, and may lead to even more years when an ice bridge fails to form in Nares Strait. However, a quantitative estimate of the changing ocean thermal impact on bridge stability can be only accomplished with coupled ice-ocean models incorporating a comprehensive ice rheology or sea ice dynamics model (e.g. Plante et al., 2020; West et al., 2021), which is beyond the scope of this study.

**Figure 13: The Sentinel-2 images showing the fracturing propagation in the central part of the ice bridge at the end of June 2020.**

## 5. Summary

This paper present insights into the ice-ocean interaction beneath the ice bridge in Nares Strait. The research was motivated by the fact that a sensible heat polynya has been observed via remote sensing in the center of the ice bridge (at Cape Jackson) every winter. In fact, our research is standing on the shoulders of Dr. Elisha Kane who discovered open water at Cape Jackson and expressed an opinion that "… *ocean-currents may exert on the temperature of these far-northern regions* [and keep water ice-free]" (Kane, 1856; p. 309). To examine Dr. Kane's enlightened suggestion, we used a combination of remote sensing data, and results from hi-resolution FESOM2 oceanic circulation model and 1-D ice growth model. The ICESat-2 altimetry demonstrates that the polynya at Cape Jackson is a part of a much broader area in the northwestern Peabody Bay forming negative ice thickness anomalies during both ice-bridge and no-bridge winters. These anomalies are likely associated with the upward heat flux from the relatively warm Atlantic Water penetrating to Kane Basin through Smith Sound and circulating cyclonically at depth within Peabody Bay. Using the observed mode of absolute surface heights around the polynya (0.26 m) as a target elevation, we ran a

series of simulations of ice growth with different combination of the snow accumulation rates and ocean heat fluxes. In simulations of anomalous ice thicknesses in the northwestern Peabody Bay, the best match was obtained using a relatively modest heat flux of 10-20 W/m$^2$ and snow accumulation rates of 4-8 cm/mo, although there might be a large uncertainty associated with possible changeability in both variables during winter. We suggested that such heat flux could be associated with the presence of dense chains of icebergs, calved from Humboldt glacier, that have become grounded on the eastern slope of the mid-basin ridge in Kane Basin. The interaction of these icebergs with the mean and tidal currents may function as an efficient stirring machine that enhances vertical heat exchange between AW and surface layer. However, the maintenance of open water at Cape Jackson requires much stronger localized heat flux. We found that at least 200 W m$^{-2}$ of sensible heat is required to keep the polynya ice-free throughout winter and about 70 W m$^{-2}$ to open it in the end of April, although not considering the altered atmospheric boundary layer characteristics over the polynya and contribution of ice advection within the polynya could result in an overestimation of these fluxes. We suggest that the enhanced heat transfer at Cape Jackson is controlled by tidally driven upwelling and mixing – the mechanism that is responsible for the formation of other sensible heat polynyas within the landfast ice cover in the Canadian Arctic Archipelago (Topham et al., 1983; Hannah et al., 2008). However, the existence of the large negative ice thickness anomalies along the entire northern coast of Peabody Bay should require another mechanism of uplifting AW and heat transfer to the surface layer. We attribute this mechanism to coastal upwelling associated with a horizontal convergence of the mean currents in the northern part of the bay, but more detailed research is needed to examine this suggestion further.

Another prominent zone with negative surface height anomalies was revealed along the western side of Kane Basin and Kennedy Channel. This narrow (~10 km) zone spreads over 200 km along the coast of Ellesmere Island from Cape Frazer in the south to the Hall Basin in the north and turns into the system of coastal polynyas that form several weeks prior to the ice bridge break-up. The ice thickness anomalies along the western coast were considered to be associated with sensible heat released either from the upper thermocline water of Pacific origin or, less likely, from the underlying Atlantic Water that both carry relatively warm water southward from Lincoln Sea. The jet of this flow follows along the western slope of the strait during winter and, in combination with tidal currents, might enhance shear instabilities over a steep topography and thereby enhance vertical mixing in the water column. Another mechanism that may enhance vertical mixing under the ice bridge is associated with the sub-ice turbulence generated by the interaction of mean and tidal flow with rough under-ice topography. In combination with upwelling of the upper thermocline water along the coast in winter (Shroyer et al., 2017), this mechanism is considered to play a key role in a considerable increase of vertical heat flux towards the bottom of sea ice along Ellesmere Island. However, quantitative estimating of its intensity is not possible without in-situ measurements.

As a result of the ocean heat flux, there are extended areas of considerably thinner ice along the northern and western coasts of Kane Basin at the end of winter. Analysing satellite images showing the stages of the Nares Strait ice bridge collapse in 2020, we demonstrated that a partial break-up of landfast ice around the polynya at Cape Jackson initiated further spreading of fractures and, therefore, could promote an earlier collapse of the bridge compared to a no-polynya situation. Moreover, the prospect of further amplification of ocean heat fluxes, due to increases of the temperature of inflowing Atlantic and, probably, Pacific Waters linked to climate warming, is expected to further impact the stability of ice bridge in Nares Strait or its formation altogether.

**Data availability**

All satellite data used in this research were obtained from open sources. ICESat-2 ATL07 data is available from National Snow & Ice Data Center (https://nsidc.org/data/atl07) as well as AMSR sea ice brightness temperatures and snow depths (https://nsidc.org/data/AU_SI6/versions/1). The MODIS true color imagery and band-31 sea surface brightness temperature were

obtained through NASA's Worldview application (https://worldview.earthdata.nasa.gov), part of NASA's Earth Observing System Data and Information System (EOSDIS). The optical Sentinel-2 images are available from the Copernicus Open Access Hub (https://scihub.copernicus.eu/dhus). The FESOM-2 model source code and configuration files are available from https://github.com/FESOM/fesom2. The portion of 1km-resolution FESOM-2 data used for this research can be found at https://doi.org/10.5281/zenodo.6360063.

## Author contributions

SK designed this research, generated all figures and performed the majority of data analysis. ID, DBabb, JE, SR, DJ and DBarber contributed to data analysis and developing of the research concept. The original manuscript was drafted by SK and further revised and edited by ID, DBabb, JE, SR, DJ, NK and DBarber. NK provided the results of high-resolution FESOM2 model for Nares Strait region. All authors have read and agreed to the published version of the manuscript.

## Competing interests

The authors declare that they have no conflict of interest.

## Acknowledgements

We dedicate this paper to our colleague, Prof. Dr. David G. Barber (1960–2022), who passed away after the consequences of heart attack in April 2022. Dr. Barber was a distinguished scientist whose invaluable input to the Arctic studies is difficult to overestimate. Having been a great person, colleague and leader, David dedicated all his life and passion to expand our knowledge on the sea ice and its role in the Arctic climate system. We appreciate Humfrey Melling from the Institute of Ocean Science (Canada) and another anonymous reviewer for their useful comments and suggestions that helped improve the paper considerably. The authors thank National Snow and Ice Data Center for the ICESat-2 data on sea ice heights (https://nsidc.org/data/ATL07/versions/3). We acknowledge the use of MODIS data and imagery available from NASA's Worldview application (https://worldview.earthdata.nasa.gov), part of NASA's Earth Observing System Data and Information System (EOSDIS). This work is a contribution to the Arctic Science Partnership (ASP) and ArcticNet.

## Financial support

Funding for this work was mainly provided by the Canada Excellence Research Chair (CERC) program (D. Dahl-Jensen, PI). D. Babb, J. Ehn, S. Rysgaard and D. Barber are supported by the Natural Sciences and Engineering Research Council (NSERC) of Canada. D. Babb is additionally supported by the Canadian Meteorological and Ocenographic Society (CMOS).

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
