# Peer review of "The role of oceanic heat flux in reducing thermodynamic ice growth in Nares Strait and promoting earlier collapse of the ice bridge"

_Ocean Science, 2022_

## Referee Comment (RC2)

**The role of oceanic heat flux in reducing thermodynamic ice growth in Nares Strait and promoting earlier collapse of the ice bridge**

Sergei Kirillov, Igor Dmitrenko, David G. Babb, Jens K. Ehn, Nikolay Koldunov, Søren Rysgaard,

David Jensen and David G. Barber

**Overview**

Nares Strait is an important oceanic connection between the Arctic Ocean and the Atlantic. It carries a sizeable fraction of the total outflows of Arctic surface and Pacific waters and provides a quick exit for thick old ice leaving the "last ice area". The rates of outflow of both seawater and ice are reduced when the strait is covered by fast ice, a condition of changeable duration that occurs in many but not all winters.

This paper explores factors that may reduce the viability of shore-to-shore fast ice in Nares Strait during winter. The discussion is based upon ice-cover observations acquired solely by satellite-based remote sensing instruments. These include Sentinel-2 SAR imagery (50 m), MODIS "true colour (250 m)", MODIS "mid-infrared brightness temperature (1000 m)", Sentinel 2 "high resolution optical imager (10 m)", AMSR "89-GHz brightness temperature (6250 m)" and IceSat's Advanced Topographic Laser Altimeter System "ice-plus-snow elevation (60 m)". No contemporary "ground-truth" data on the ice cover or the ocean were collected. A 1D sea-ice thermodynamic model was used in an attempt to distinguish the separate contributions of ice and snow to measured elevation. Oceanographic insight was garnered from a run of the Finite volumE Sea ice Ocean Model v2 for the 5-year interval, 2006-2010. The authors provide no indication that the viability of this model has been evaluated in Nares Strait.

The authors draw attention to a small polynya that forms off Cape Jackson on Greenland's coast at the southern end of Kennedy Channel during late winter of years when fast ice covers Nares Strait. They argue that this feature is indicative of a localized upward flux of sensible heat from the ocean to the ice. Using lidar data, they map a larger negative anomaly in surface elevation (ice plus snow) around the polynya. They use the measured elevations to constrain a 1D thermodynamic model of ice-plus-snow, driven by surface air temperature, to explore the complementary influences of ocean heat flux and snow depth in reducing ice-thickness. The "best match" corresponded to snow accumulation at 4-8 cm/mo and a 10-20 W/m2 heat flux from the ocean. Atlanticderived water found below 150-m depth was proposed as the source of this heat, delivered directly to the underside of sea ice via upwelling. The paper also documents a band of relatively low ice-plus-snow surface elevation along the eastern coast of Ellesmere Island during two years with fast ice, but this was not observed during the one studied year (2019) when ice was mobile throughout the winter. Upwelled Atlantic-derived water was proposed as the source of this anomaly also. In closing remarks, the authors speculate that the zones of thermally weakened fast ice that they have identified on both sides of Kennedy Channel and Kane Basin in late winter weaken the stability of fast ice along the full 550-km length of Nares Strait and promote its earlier collapse in summer.

**Assessment overall**

The authors make a useful contribution in drawing attention to the influence of oceanic heat flux on the fast ice cover of Nares Strait. Oceanic heat flux has been shown to have a noticeably impact on the fast ice cover of the Canadian polar shelf, particularly in shallow waters with strong tidal currents where small polynyas form (Topham et al., 1983 [JGR 88(C5);]; Melling et al., 1984 [CSR 3(3)]; Melling, 2002 [JGR, 107]; Hannah et al., 2009; Melling et al., 2015). It might indeed be surprising if such features were not found along Nares Strait.

They make ingenious use of information from a variety satellite remote sensors to document ice-cover characteristics and state of motion, to detect polynyas within fast ice, measure ice-plus-snow elevation and map surface temperature. However no in situ data are brought into play. This is regrettable. In consequence, the accuracy, precision and possible bias of elevation and snow-surface temperature measurements for example were not determined, so that confidence in these data and the results derived from them is eroded. Moreover, the critically important separate contributions of ice freeboard and snow depth to elevation are not known, although the authors make a valiant effort to generate "educated guesses" through use of a 1D thermodynamic

ice model; unfortunately, the "surface" air temperature used to drive this model was taken from the ERA5 reanalysis which has a 31-grid scale, much too large to achieve a realistic representation of surface weather conditions in Nares Strait.

The lack of contemporary or past oceanographic observations at the locations of interest is a serious shortcoming in a paper that strives to attribute polynya formation to oceanic heat flux. The FESOM global iceocean model has been harnessed in an effort to fill this gap. However, since this model neither assimilates contemporary ocean observations, nor seems to have been evaluated against existing ocean observations collected nearby, nor to incorporate tides, there is little basis for confidence in the minute (from a global perspective) thin-ice features that it is called upon to "explain".

In the particular instance of the Cape Jackson polynya, the authors could have saved themselves some trouble through a heavier reliance on Hannah et al. (2009). I examined CHS Chart No. 7072 to find a 43-fm (78 m) sounding 4 naut miles to the SW of Cape Jackson. I estimated depth beneath the polynya as half this, since the polynya is centered about 2 miles off the cape. WebTide (https://www.bio.gc.ca/science/research-recherche/ocean/webtide/index-en.php) predicts a 1 m/s spring tide here, so that Hannah's tidal mixing parameter is 2.1. This is comparable to values at polynyas in fast ice across the Canadian polar shelf, where turbulence generated by energetic tidal currents moves heat upward from relatively shallow depth. It seems unnecessary to look to the weaker general circulation to lift warm water from depths 3-4 times greater.

I believe that this paper should be published. However at present, it strives to be too comprehensive, is too speculative and therefore too long. There is valuable information therein and some pioneering use of remote sensing, but these strong points don't shine forth as well as they should. Specific suggestions for changes are listed below.

**Comments (major)**

- 1. The authors have chosen to refer to the fast ice that covers the full 500-km of Nares Strait during many winters as the "ice bridge": The terminology is confusing because a long strip of fast ice does not resemble a bridge. I believe that most readers will consider the bridge to be the arch that forms the boundary between fast and mobile ice, most often in southern Kane Basin. As in masonry, the arch is strongest geometry for a load bearing structure because it is everywhere under compression, thereby exploiting the stress-state where sea ice is strongest. I recommend that the authors devise a different term to refer to locations within the fast ice "above" the arch. For example at Cape Jackson, "more than 200 km north-east of the bridge".
- 2. Line 148, "The model was driven by the atmospheric reanalysis fields from JRA55-do": This was probably not a good choice. Samelson and Barbour (2008) concluded that a grid 10x finer than that of JRA55 was required to correctly represent weather conditions in Nares Strait. I recommend adding a discussion the capability JRA55 to represent the mesoscale meteorology of Nares Strait, a channel much narrower than 55 km in width for much of its length, bordered by high terrain and characterized by a strongly stable atmospheric boundary-layer the Arctic inversion during the freezing season. This could be perhaps achieved via comparison of simulations by the ERA55 and Polar MM5 models.
- 3. Line 166, "The 6-hourly records of 2 m air temperature, wind speed and humidity in Kane Basin were taken from the ERA-5 global reanalysis database": The 31-km grid of ERA5, comparable to the width of Nares Strait, does not come close to resolving the channel and very its steep surrounding terrain. I suspect that much of the strait's "sea surface" resolved on a 31-km scale will actually be above sea level and therefore "terrestrial". It is very difficult to accept that ERA5's 2-m air temperature values hold much credence for simulating ice growth in the real world. Since sea ice dissipates the upward flux of latent heat from wintertime freezing by long-wave radiation from its top surface, would you not be better using satellite-derived surface radiation temperature to model freezing? Please strive to persuade readers that my viewpoint is invalid.
- 4. Line 415, "It was found that the ocean heat flux at Cape Jackson needed to exceed 200 W m-2 to open the polynya as early as in March": The message intended here is unclear from the present text. Revision is required.

I don't believe that it is plausible for a polynya to suddenly melt itself into existence – oceanic heat fluxes just aren't large enough. The more likely role of oceanic heat flux is keeping the ice relatively thin (and relatively weak), so that more powerful mechanical (fracturing, rafting, flooding and downstream advection, etc.) and thermodynamic (radiation) processes can do their work. Principal among the former are the stresses exerted by wind, wind-waves and tidal current on already thin ice. Once these open a polynya, new ice (mainly frazil and nilas) created by high rates of heat loss (> 200 W m-2) from the surface will continue to be removed by current, while insolation and downwelling short-wave radiation may deliver appreciable heat directly to seawater. This "tag-team" approach to creating a polynya in fast ice, involving both dynamics and thermodynamics, has been discussed by Topham et al., (1983. JGR 88).

Actually lifting warm water 100 m or more to the surface to open a sensible-heat polynya takes a large input of kinetic energy to the ocean, which cannot occur with a continuous cover of fast ice. It is only possible when strong winds act on mobile ice or open water. There is a paradox because necessity for strong winds is the same requirement for the opening of latent heat polynyas; the distinction between the purported "two types" of polynyas is not clear (see discussion in Melling et al, 2001). Moreover, because the wind must be integrated over a large expanse of mobile pack ice to accumulate enough kinetic energy to drive upwelling, the formation of small polynyas in fast ice, such as that off Cape Jackson, via this mechanism is unlikely.

- 5. Lines 447-473: The speculation about iceberg melting provides an intriguing diversion, but it doesn't add much to the concepts central to this paper. I suggest that it be removed from the paper.
- 6. Section 4.2, "The formation of ice thickness anomalies along the western coast of Nares Strait": This is indeed an interesting feature. It did occur to me that is might possibly be an artifact in the southern half of the strait to the authors' referencing of elevation to the mean value measured along the full length of the strait. Has the possibility been investigated that the higher sea level in the north that drives the current might explain this anomaly?

Alternatively, could it be a manifestation of the undercurrent that hugs the Ellesmere coast when fast ice covers the strait? It is interesting that the elevation anomaly has roughly the same 10-km width as the undercurrent mapped by Rabe et al (2012; Fig. 4b). A connection to the dynamic relief reflecting geostrophy in this flow is implausible; it is only about 10% of the measured 20-cm drop in ice-plus-snow elevation adjacent to the coast. The inverse barometer effect, which might also contribute to lowered sea level in response to higher SLP at the western shore (not resolved at this scale by ERA5) is also too small: Sea-level pressure is only higher by 2-4 mb on the Ellesmere side of the strait (Samelson and Barbour, 2008: Fig. 8). Nonetheless, I recommend noted both possibilities as having been explored.

7. Line 484, "We suggest that the observed negative anomalies are attributable to the heat upwelled from the underlying mAW": I believe that this suggestion has merit, but that the details are incorrect.

The flow structure beneath fast ice in Nares Strait depicted by Rabe et al (2012; Fig. 4b), displays a jet of roughly 10-km width against the Ellesmere shore, centered at about 80-100-m depth. The baroclinic adjustment of the ocean to this jet (not shown) involves downwelling below the core of the flow and upwelling above. This leaves no mechanism to raise mAW through the core of the undercurrent to the surface on this side of the strait. Indeed, the cross-strait circulation that compensates for downwelling of mAW on the western side is upwelling on the other side, near Greenland!

However, upwelling does occur above the core of the jet. This would bring Pacific Winter Water as much as 0.2C warmer than the surface-freezing temperature (see Melling et al. 1984 Cont. Shelf Res 3) to the base of the surface mixed layer (see Melling et al. 1984 Cont. Shelf Res 3). This sensible heat in this Pacific Water could provide a heat flux to the underside of the sea ice via entrainment into the turbulent surface mixed layer. The needed turbulence kinetic energy could originate in part from brine-driven convection (ice growth) in the mixed layer and in part from shear between rough immobile ice and the rapid tidal flow. Melling et al. (2015) estimated an oceanic heat flux to the base of ice as 15 W/m2 under similar circumstances in Penny Strait, which would be sufficient to melt about 0.5 cm/d from 3-m ice (with 10-cm

snow) at -25C. It should be noted that the submerged jet, and the upwelling above it near the western shore, do not exist when the ice is moving, so that the oceanic flux would be much reduced in years without a fast-ice cover.

8. Lines 504-505, "Transformation of these currents over steep topography generates baroclinic semidiurnal tidal wave that may considerably enhance vertical mixing through benthic stresses and shear instabilities": I suggest that the steep cliffs on the western shore, indicative of deep water close to shore, make turbulence and internal waves generated in the benthic boundary layer irrelevant to the ice far above. However, I believe there is a good possibility to generate strong turbulence, mixing and entrainment through the action of the tidal flow (Pite et al., 1995. JPO 25) on the very rough under-ice topography of Nares Strait (Ryan & Munchow, 2017).

I suggest that the authors give some thought to this alternate, and I believe more plausible, explanation for the source of ocean sensible heat.

- 9. Line 523, "weakens the cohesion of landfast ice against the shoreline in Kane Basin": The tidal cycles in sea level ensure that the ice sheet is always fractured at the coast, not bonded to it. However, the word cohesion implies that the authors consider that bonding of ice to the shoreline is important. This line of thought runs contrary to decades-old discussions of fast ice in deep water, where it is the formation of ice arches across channels which stops the movement of ice behind them, not shear strength at the shoreline. The upwardly convex shape of a masonry arch is the key feature that allows it to resist downward loading; the shape ensures that all the stone in the arch is under compression, the stress state in which it is strongest. Indeed the stress is highest within the wedge-shaded stones of the arch and much less above them. Pack ice also is strongest in compression and much weaker in shear. Although there are likely several arch-shaped load-bearing features distributed in the fast ice along the length of Nares Strait during any winter, much of the fast ice cover will be in a low state of stress; cohesion at the shoreline is probably unnecessary for fast-ice stability, although its confinement by irregularly shaped shorelines may constrain it from moving locally. Conversely, weakening of that confinement by melting at the coast may allow it to shift around in response to wind and tide. It is quite common to see the ice in Kennedy Channel become mobile between arches at its northern and southern ends long before the collapse of the arch in Smith Sound allows the ice in Kane basin to do the same. The same phenomenon is seen annually in Prince Regent Inlet. I don't think that the authors' argument for up-channel polynyas hastening the break-up of fast ice further down-channel has much merit, as presently written. It is possible, of course, that phenomena may be correlated in time because of the influence of a third circumstance not identified.
- 10. Line 528-529, "This break-up appeared to release internal stresses in the ice bridge and led to concomitant ice cover break-ups in the main channel": This statement appears to rely upon a knowledge of the dynamical state of the ice cover. Nothing is known about stresses. In reality all you have access to is evidence of deformation (in the form of cracks) and of motion. Also see comment #9.
- 11. Line 531: This paragraph gives the impression that the polynya has played a role in the breakup, but really all that you demonstrate is that the breakup was correlated with expansion of the polynya. Perhaps the expansion of the polynya is just one event in the process. A more robust discussion, with a more useful take-away, would review the other factors in play, as listed in Line 520. If such completeness is thought to be beyond the scope of the paper, perhaps it should be covered in a separate paper. See comment #9.
- 12. Line 539-540, "The only oceanic heat source available to maintain such a polynya through winter is the modified Atlantic Water": This is not true. It may be the warmest source, but it is not the one closest to the ice. My comment on Line 484 raises the possibility that the less conspicuous warmth of the Pacific Water might be more influential than you give credit for. I recommend that you re-think the paper with this in mind.

**Comments (minor)**

Line 27: Shokr et al. (2020) is a weak reference for the role of the along-channel sea-level in driving flow down Nares Strait. Münchow & Melling, J Mar Res 66, doi.org/10.1357/002224008788064612 would be much better.

Line 30-31, "The ice bridge also helps prevent the loss of the thick, old ice from the Last Ice Area": The paper cited (Moore et al, 2019) is not helpful in substantiating this statement; it has very little to say about Nares Strait. To my knowledge, there has not yet been a study demonstrating that ice loss from the LIA, as distinct from ice export through Nares Strait, is reduced during years when an ice arch forms there. Nares Strait is only one of four pathways (and the narrowest) via which ice leaves the LIA – the others are to the NE via Fram Strait , to the SE through the QEI and to the SW to the Beaufort Sea. It is quite plausible that a blocked Nares Strait simply creates a diversion of ice to one of the other pathways, most likely Fram Strait. You need a citation that demonstrates convincingly that this is not so.

Line 38, "... peak in the fraction of sea ice with a draft between 2.6-2.8 m": It is important to note here, as was in the cited paper, that this range in draft was computed on the assumption of no snow cover, which may bias values appreciably high. Also, a referenced estimate of the empirical accuracy in draft estimates from CryoSat freeboard should be included here.

Lines 46-47, "That bridgeless years only occurred during last 15 years underscore a general shortening of bridge existence period and point to changes ...": It would be appropriate to clarify that this statement refers to the absence of an ice bridge at Smith Sound (think) and not to the much smaller number of years when there was no bridge anywhere between Baffin Bay and the Arctic Ocean.

In this clarified context, it should then be noted that there was one winter (1995) in the 1990s with no arch at Smith Sound – in 1995 the arch formed at Hans Island – and one (1993) essentially like 2007 with no arch anywhere; "essentially" because an arch in Smith Sound that year lasted only 10 days (Vincent 2019). With a 30-year perspective, the record looks less amenable to interpretation via trend: there is a cluster of 2 of 3 years with no arch at Smith Sound in the mid-1990s, then an 11-y period with annual arches, then a cluster of 3 of 4 years with no arch in the 2nd half of the 2000s, then a 6-y period with annual arches, then a cluster of 2 of 3 years with no arch in the second half of the 2010s. Disregarding clustering and estimating the probability of no bridge in any year from the data as 7/31, one uses the Poisson Distribution to estimate the likelihoods of the observed gaps between no-bridge winter – that is having 2 no-bridge years in 2 years, 2 in 3 y, 2 in 7 y and 2 in 12 y. These are 6.4%, 11.7%, 25.7%, 24.4%. The low values for the small gaps suggest there is clustering in play; the relatively high values for the large gaps suggest that such wide gaps are not unexpected, so that bridging despite weak clustering, looks like a Poisson process. On these grounds I suggest a re-examination the statistical confidence of the statement in lines 46-47, which is based on such a short time series.

Lines 48-49: I think that the date-based approach of Vincent (2019) is probably a more robust approach to a short 30-year time series than is the counting of the rare occurrences without arches, which the authors have used here.

Line 54-56, "it is the sensible heat polynyas ... that are more common in the Canadian Arctic (Hannah et al., 2009)": The authors appear to mis-quote Hannah et al. (2009), who state "... are widely distributed across the Canadian Arctic Archipelago"; Hannah at al. are clear that these sensible heat polynyas are features within fast ice in this region. Their map (Fig. 1) shows that the latent heat flaw-leads and polynyas that form along the perimeter of the fast ice are actually more widespread across the Canadian Arctic waters and occupy much more area.

Line 67: Refer the reader to Fig. 1 for the mapped location of Cape Jackson.

Line 67 et seq., "... at Cape Jackson in the central part of the bridge": The terminology is confusing. I believe that most readers will consider the bridge to be the arch that forms the boundary between fast ice and mobile ice in southern Kane Basin. It follows that the central part of the bridge is the "top" of the arch, halfway across the strait between Greenland and Ellesmere. However in this sentence, the authors are referring to a location in fast ice more than 200 km "above" the arch. I recommend that the authors devise a different term to refer to

locations within the fast ice "above" the arch. Simplest in this example would be "... at Cape Jackson, more than 200 km north-east of the bridge".

Line 93, "... maintaining water at Cape Jackson ice-free during winter": The reality is ""... maintaining water at Cape Jackson ice-free at times during winter".

Line 94 et seq., "under the bridge": See comment re line 67. I recommend using the phrase "beneath the fast ice" for the reason already given.

Line 100: Line 67: Refer the reader to Fig. 1 for the mapped location of Peabody Bay.

Line 130 et seq., "crossing the bridge": See comment re line 67.

Line 132, "Although ATL07 data are manifested to be adjusted for geoidal/tidal variations and inverted barometer effects": The correction for the inverted barometer effect is probably only accurate in wide deep ocean basins where the long ocean wave which is the ocean's response to changing atmospheric pressure can move as fast as, and in the same direction as, the SLP anomalies moving at 20-25 m/s. I suspect that the correction will not work well in a long (550 km) narrow (35 km) strait. I urge the authors to find and reference research that provides a discussion of the accuracy of the inverted barometer correction in confined coastal waters.

Line 133-134, "... may still contain unknown uncertainties related to the regional synoptic variability of sea level associated with wind forcing and/or with ocean dynamics": With respect to the atmosphere, I recommend replacing "wind forcing" with "strong wind, air-pressure and ocean dynamical effects on the mesoscale (10-30 km)", referencing Samelson and Barbour (2010).

With respect to the ocean, Münchow & Melling (J Mar Res 66) provide estimates of the anomalies of sea-level height relative to the mean. These have amplitudes as large as 10 cm along-channel and a few cm/s across-channel. These along-channel value is large enough to contribute appreciable fortuitous NE-SW varying anomalies in thickness that are computed relative to an along-track (approximately along-channel) mean. This source of error requires discussion.

Line 139-140, ">0.3 m mean snow depth in Kane Basin. However, as 140 we will show later, this height seems to be overestimated". Reference to Samelson and Barbour (2010) is again appropriate, since the extremely strong winds common in Kennedy Channel and the vicinity of Cape Jackson (see also Melling, Oceanography Mag, 2011) may indeed provide a strong disincentive for the accumulation of snow.

Line 159, "... generally have good agreement with the mooring records": It is necessary to provide an assessment that is more specific in relation to the comparison of model with data in relation to the cross-channel scale of flow features, their positions cross-channel and in depth and their intensity. Can the countercurrent on the Greenland side be simulated?

Line 182, "MODIS imagery confirm that a polynya is present every winter at Cape Jackson": The sentence that follows that quoted indicates that the following is more precise: "MODIS imagery confirms that in every winter when fast ice fills the strait, a polynya appears at Cape Jackson late in the season".

"... within the bridge": See comment re line 67.

Line 190, "may indicate either the ice-free surface or thinner ice": Clarification, "may indicate either the an ice-free sea surface, locally thinner ice, locally thinner snow or both the latter".

Line 205, "... If 50% of the 0.26 m surface elevation is attributed to a snow layer ...": The occurrence of very strong, very turbulent winds off sea capes is well known to mariners. Cape Horn and Cape Farewell, at the southern tip of Greenland, are perhaps the most famous. See Winant et al. (1988) J. Atmos. Sci. 45. Such conditions would be very effective at scouring snow from the surface of sea ice and moving it downwind. It is therefore quite plausible that both ice thickness and snow depth become thinner on approach to Cape Jackson, as the density-stratified oceanic and atmospheric flows accelerate in response to submarine and subaerial topography blockage, respectively. IceSat may be sensing environmental response to both these effects, not just to one or the other.

Lines 218-238 & Fig. 4, "along-track anomalies averaged over 1x1 km squares": On the "basin-wide scale" discussed here, the anomalies, calculated relative to mean height of any ascending or descending track crossing the bridge between 55-76°W and 78.25-82.5°N, may well be contaminated by a varying along-channel gradient is sea-surface height – see comment on lines 133-134. It is appropriate that the authors acknowledge this source of error and discuss its impact on results.

Lines 231-242 & Fig. 4, "In the main channel, the anomalies are highly irregular and form a speckled pattern, whereas the anomalies in Peabody Bay form a consistent pattern with positive anomalies in the southeast and negative anomalies to the northwest": It is unclear, with the continually moving ice of 2019, why the elevation anomalies are not smoothed out via averaging over time. The small scale of the speckle in elevation in 2019, not so different from that in the years with immobile ice is difficult to understand. Please explain.

A similar speckled pattern of h was observed over the landfast ice in Peabody Bay in 2020 (Fig. 4b), but not in 2021. What is the application in these instances?

Lines 233-234, "The difference in surface height anomalies between the southeastern and northwestern parts of Peabody Bay is supported by a similar difference in the observations of Tb" : In what sense do we interpret "is supported by"? Do you mean "is correlated with" or is there some physics behind the claim of support?

Line 234: Interpretation of AMSR brightness temperature. Please clarify whether the values depend on emissivity (ice type ) as well as on surface temperature (of snow, of ice, or of somewhere between?).

Line 235: Should "southwest" be changed to "southeast"?

Line 280, "we applied the 1-D thermodynamic ice growth model": Things like thermal coefficients, snow density, short and long-wave radiation, cloud cover do matter. Please provide a quick overview of the properties of this model, or an equivalent citation.

Lines 282-284. "We used 4 cm mo-1 snow accumulation rate to reach a modest snow thickness of 14 cm at the end of winter that is reasonably close to 19±2 cm obtained with AMSR2 data for Peabody Bay": As mentioned earlier, snow accumulation matching that in Peabody Bay may be unlikely. Ice off Cape Jackson may be blown clear of snow by frequent extreme winds in winter (see Samelson and Barbour, 2008: Fig. 6). It would be appropriate to mention this possibility.

Lines 288-289, "For having ice-free water in May, the heat flux should reach 70 W m-2 and be above 200 W m-2 to let polynya form in early March": These estimates presume that there is no advection of newly formed ice downstream and beneath thicker pre-existing level ice and, I believe, that there is no insolation.

Lines 440-441, "Although the northern branch is warmer and, being considerably faster, transports more heat compared to the southern branch ...": Unfortunately, the northern branch is partially blocked from entering eastern Kane Basin by a shallow (70-90 m) spur extending more than 100 km southwest from Cape Jackson. The deepest crossing is relatively shallow, a 220-m sill at 79 40'N close to the Ellesmere shore. Moreover, because of geostrophic adjustment in the Arctic outflow, the warm mAW is at it deepest on the western side of the basin. To make a convincing argument about the temperature of the water that gets over this sill, more careful thought is needed. Where does the mechanical energy to lift water of the sill come from? I don't believe that a numerical model unvalidated in Nares Strait is a substitute for data needed to substantiate an hypothesis. Perhaps the authors could strengthen their case by exploring what the model has to reveal about the energetics of the phenomenon that they propose?

Lines 456-457, "However, it is noteworthy that all these iceberg chains are located within the region with pronounced negative anomalies of ice surface heights in 2019 and 2020": Qualitatively, from the insets on Fig. 7, I estimate that the bergs cover only perhaps 10-20% of the sea surface; they could create point sources of turbulence kinetic energy through interactions with current, but are likely too sparse to form an area-wide source to explain the sea-surface anomalies which are manifest on the scale of the entire basin. Moreover the warm seawater contacting icebergs at depth has plenty of opportunity to transfer its heat directly to the bergs, rather than hoarding to create havoc on the sea ice. The authors' hypothesis is plausible, but it needs appreciable quantitative physics to convert it into an explanation appropriate to uplift from 100-250 m depth.

Lines 460-461, "However, the melting in this case is associated not with latent heat flux from water, but with dissolution controlled by solute transfer between water and ice-ocean interface (Woods, 1992)": I don't understand this point. I believe that a transfer of sensible heat to the iceberg is still required to free individual water molecules from the crystal lattice as dissolution proceeds. Please check whether you are citing Woods' work correctly.

Lines 496-497, "The stronger vertical mixing associated with the shear instability of the subsurface southward jet along the western coast ... . This statement is speculative and not supported by observations. It is trivial to show with data in the Rabe papers that the gradient Richardson Number in the shear layer above the jet is about 2.2, almost 10x the threshold for shear instability. The most plausible sources of turbulence kinetic energy are in the wintertime mixed layer, namely shear in tidal currents at the base of rough sea ice and, less important with thick ice, brine-driven convection. Both can be estimated. I recommend that the authors do so.

Lines 506-507, "generally support the idea of topographically controlled instabilities associated with the mean current and reversible tidal flow": I don't think it necessary to speculate about submerged topography generating instabilities. Headlands, by partially blocking along-shore currents, are notorious for strong tidal currents, and under-ice topography in Nares Strait is very rough.

Line 511, "probably through the local upwelling": What is the basis for "probably". I don't believe that there are any soundings in Flagler Bay, so the existence of a sill is speculative.

Lines 543-544, "Münchow (2011) reported a very similar warming in the southward branch of mAW of 0.23 °C/decade": Actually Münchow et al. (2011). This paper provides very weak evidence of long-term warming because the period of observation was only 6 years. The present authors have taken the liberty of extrapolating this to 10 years, and then referring to a supposed "as further warming of mAW progresses" – all this without having made a bullet-proof case for an influence of mAW on the sea ice of Nares Strait. It is one thing to have mAW affect glacial ice at the same depth, quite another to postulate an influence on sea ice at the surface hundreds of meters above. I suggest to the authors that the present evidence to make this projection is not statistically robust.

---

## Author Comment (AC1)

In the following, the comments by the reviewer are in normal blue characters and our responses to the comments are in cursive and indented. Modifications to the text are shown in quotation marks with bold characters indicating newly added text, and normal characters indicating text that was already present in the previous version. The line numbering in our responses corresponds to those in the revised manuscript with the tracked changes.

RC1: 'Comment on os-2022-16', Anonymous Referee #1, 10 May 2022 reply

"General comments":

The authors describe novel results showing visible and potentially invisible polynya formation within the Nares Strait ice bridge using visible and thermal imagery (MODIS), passive microwave (AMSR-E/AMSR-2), and altimetry measurements (ICESat-2). There is little discussion in the literature of these polynyas, the use of ICESat-2 and thermal data for this problem is novel, and polynya impacts on the North Water ice bridge has not been examined. The authors show evidence and make a reasonable argument that these polynyas impact the breakup of the ice bridge. They also make a convincing argument that upwelled Atlantic Water causes the polynyas and potentially the sea ice thinning patterns in Nares Strait and Peabody Bay. Some of the arguments, however, are unclear because of grammatical or organizational errors, or are missing counterarguments that are important to discuss with the reader. The treatment of MODIS and ICESat-2 data is unclear or inaccurate at points (noted below) and more clarity here will make it easier to assess the quality of the results. I appreciated the authors' thorough use of various methods to approach this problem and their clear goal to be transparent about the limitations in the analyses. The needed grammatical/organizational adjustments alone are substantial, and other potentially major revisions are included.

"Specific comments":

Although it does appear that the warm area in northern Peabody Bay is associated with warmer surface ocean heat, at least in years where there is landfast ice covering the entire area, a distinction should be noted in the text about other reasons that warm temperatures may be observed at the surface. Especially important for snapshots and short time periods, AMSR and MODIS surface temperatures will measure warmth merely because sea ice is broken (more surface ocean is exposed) or recently formed. That can happen because of mechanical wind forcing that has nothing to do with ocean/ice temperature. Further, when looking at snapshots of temperatures when ice is mobile, it is one thing to say the surface is warmer in a location because there is an open ocean surface (the surface is of course warmer if it is open ocean than if it is covered in sea ice) and to say that the open ocean surface is quantitatively warmer than freezing temperatures or than other years (might indicate AW coming to the surface). Differences in figure 10 are more likely to have resulted from synoptic scale variability than an interannual one.

> *We understand this reviewer's concern, but note that in this paper we mainly analyzed the data collected within solid landfast ice sheet from the middle to end of winter season. This automatically excludes such factors as 'broken ice' or 'recent formation' from consideration. Of course, if we are not talking about the mobile ice in the main channel in 2019 or in NOW area.*

*The warm temperatures in the channel in Fig.5a are linked to the fact that leads are present in the mobile sea ice drifting south. But the mobile ice is not a topic of our study.*

*The only exception is the discussion around Fig. 10 where the snapshot MODIS temperatures in early winter are shown. However, that figure is only for pointing at the existing (interannual) variability of surface water temperatures that may indicate the altering ocean heat flux. In respect to this comment, we changed text in Line 469 in Section 4.1 (see our response to your comment to Figure 10):*

*"The **MODIS** brightness temperatures, Tb, shown in Fig. 10 **generally** support the **idea that the thermal state of the ocean surface in Nares Strait varies interannually**. In December 2019 (Figure 10b), the high $T_b$ conditioned the ice-free (or covered with thin ice) area **in the northwestern part of Peabody Bay and at the eastern side of Kennedy Channel**. Although the signatures of warmer water **in Kennedy Channel** can also be traced **through leads within the mobile sea ice** in December 2018 and 2020, $T_b$ was observed to generally be lower **and may indicate reduced ocean heat transport towards the surface from below**."*

The authors' point that snow depth is really challenging to get an accurate measure of, is well-demonstrated and important for the community.

*Thank you, but it's really a basic conception. The importance of snow is well known for everybody who deals with the remotely measured sea ice thicknesses one way or another. We just tried to obtain very rough quantitative estimates of the snow impact on the observed anomalies. It may even turn out eventually, that all our estimations of the snow accumulation rates in the region are far from correct. And yes, an accurate measure of snow depths would improve our ability to obtain more accurate results.*

It is not yet clear to me if the sea ice appears to be thinner in northeast Peabody Bay because of sea ice thinning or winds scouring the sea ice. The ice temperature differences look like they may arise from atmospheric phenomena coming from the northeast corner of Peabody Bay rather than from warming from below/thinning of the sea ice. I think the wording of the modeling work was a little disorganized and could be streamlined to make the main arguments of the model more convincing. Taking a glance at the general weather patterns and wind direction/speed in the northeastern corner of the bay and commenting on that could also bolster the argument for or against AW being the cause for thinning of the sea ice. I am convinced that the persistent polynya at Cape Jackson is originating from AW upwelling.

*We fully agree that the moderate negative anomalies in the north**west** part of Peabody Bay may be partly (or even fully) associated with a wind effect on snow cover or to a specific spatial pattern of snow depth distribution in general. But it is what we say in Line 488 (Section 4.1): "... it is possible that an unknown spatial distribution of snow may considerably affect the magnitude of anomalies of ice freeboards and our suggestions about the ocean heat impact [on see ice thicknesses]."*

*We modified the Lines 497-503 (Section 4.1) and also made small changes in the following paragraph to represent the model results clearer:*

*"Several simulations with different snow accumulation rates and ocean heat fluxes were run to find an optimal combination of these parameters to match **the observed modal surface height of 0.26 m near the Cape Jackson polynya (Fig. 3). These simulations were made under consideration that the polynya is kept ice-free during winter by a large (>200 W m-2) ocean heat flux**.  Such large heat flux **within a relatively small polynya area** seems to be associated with a **local** upwelling **and followed mixing** of warm core of the southern branch of mAW rather than with vertical mixing **alone.**"*

*However, it is very difficult "to make the main arguments of the model more convincing", because one of the major outcomes of this part is that "Even though the [ice growth] model could reproduce the observed difference of elevations within landfast ice in vicinity of polynya reasonably well, the obtained results should be considered only as an approximation because steady snow accumulation rates and heat fluxes were used, which is likely not representative of their true change in times".*

Do years with earlier ice bridge breakups coincide with a larger or more persistent Cape Jackson polynya? This information could make the linkages clearer between the polynya and ice bridge.

*It's a really interesting idea, but its realization would be difficult. The quantitative estimate of size and persistency of the relatively small Cape Jackson polynya would require a good set of high-resolution true-color images that are limited to the recent (from 2015) Sentinel-2 dataset only. The longer series of MODIS images (from 2000) don't have a good spatial resolution (see our Figure 2) and these images are also limited to clear-sky periods only. A straightforward attempt to find a correlation between the visual sizes of polynya in Fig.2 and the dates of breakups (Kirillov et al., 2021) did not give any good results.*

*In addition, the polynya at Cape Jackson is not the only factor that supposedly affects the bridge breakups. The polynyas along the western coast, air temperatures (through sea ice thawing) and wind may contribute to a collapse. We don't think we are able to investigate all this within a scope of this paper.*

General grammatical and organizational errors throughout with some other writing errors (e.g., inconsistent figure/fig referencing in text). I've commented on some of these specifically in the technical corrections for the Introduction and Methods only, but they exist throughout.

*We did the best to find all mentioned errors in the text. Thank you for pointing at this problem.*

- L105-119 – It is unclear what specific products/levels of data were used for this work. Also, please clarify how MODIS "sea surface" brightness temperature (Tb) provides temperatures of sea ice.

*We added the information about the processing levels for MODIS, AMSR and Sentinel-2 products.*

*In respect to the second part of the comment, in this study we used Tb to highlight a temperature contrast between certain areas, but didn't interpret these temperatures as absolute*

*temperatures of the ice/snow surface, although MODIS 11µm brightness temperatures are reasonably close to the temperature of sea ice or snow and might be interpreted so.*

*We would like to clarify we never intended that either MODIS or AMSR provide information on the temperature of sea ice/snow or water. The brightness temperature is related to the temperature of underlying surface, but they are not the same. Because $T_b$ is dependent on both the actual temperature and the emissivity of the snow/ice, it leads to lower temperatures for satellite measurements such as the 89 GHz channel than the actual temperature values.*

*Also, see our response to your comment to Figure 5.*

- L122-123 – Segments are calculated using 150 signal (surface) photons. Photon density varies by surface type and can extend to as much as 150 m so the segment lengths listed here are not accurate as described. The ATL07 description should be adjusted to include this.

*We added the followed sentence to describe the variable resolution more carefully:*

**"The along-track resolution is variable according to the number of photons returned, but the typical lengths of segments are 15 and 60 m along strong and weak beams, respectively."**

- L124-131 – Authors mention they can use one method for determining freeboard heights in some instances and use relative heights to produce maps. It isn't clear what analyses in the results use the first method versus the second.

*We modified this part as follows:*

**"At a close distance from the polynya at Cape Jackson (up to 20 km), the ice-free area** *was used as a reference level for an estimation of absolute elevations in vicinity of the polynya. Although the absolute surface heights over narrow bridges may* **also** *be determined via linear interpolation of sea surface heights measured at the opposite sides of bridge (Babb et al., 2022), however, this approach requires ICESat-2 tracks cross a bridge from edge to edge which does not work for the* **long and narrow landfast ice-covered bridges that form in** *Nares Strait."*

- L135 – Why the uncertainties are small is unclear. References would help.

*Addressing this concern and also following the recommendations of the second reviewer, the new* Lines 286-292 *(Section 3.2) were added to explain why all these uncertainties are suggested to be small.*

**"A relatively short off-shore extension (about 10 km) of both coastal zones (along the western coast of Nares Strait and along the northern coast of Peabody Bay) eliminates the regional variations of sea level as a factor contributing considerably to the anomalies. For instance, Samelson and Barbour (2008) reported the relatively small spatial gradient of sea-level pressure over the full width of Kane Basin corresponding to about 2 cm of sea level difference with higher levels along the Greenlandic side of the strait. A geostrophic adjustment requires less than 2 cm sea-level drop from Ellesmere Island to Greenland in Kennedy Channel (Münchow et al., 2006). Also, using the tidal gauge records at Alert and at the opposite sides of Smith Sound, Münchow & Melling (2008) reported the across- and along-channel sea-level differences varying in Nares Strait from a few centimeters to about 10 cm, respectively.**

*However, these relatively large differences could be associated with the local dynamical effects as all bottom pressure sensors were deployed in shallow bays not far from the areas covered with mobile ice at Smith Sound and at Alert. The actual sea level gradients below the ice bridge in Nares Strait and their input to the observed ICESat-2 anomalies remain unknown, but are thought to be small comparing to the gradients associated with the anomalies observed along the western coast of Nares Strait and at the northern coast of Peabody Bay."*

*We also changed sentence in Line 157 (Section 2.2) to indicate that more explanations are followed:*

*"However, the presence of immobilized ice and relatively short along-track coast to coast distances within Kane Basin suggests that spatial variations of these uncertainties are relatively small **at least comparing to the calculated anomalies (more details are given in Section 3. 2)**."*

- L132-141 – The methods here are unclear. It is unclear what (if) the authors are doing about the uncertainties mentioned, implications for the study, and if these uncertainties are problematic enough to prevent being able to use them. See the subsequent comments.

*See our previous comment and the changes made in the text.*

- L137-141 – It isn't clear how this statement is pertinent to your method. Are you mentioning issues with determining snow height depth in general for ICESat-2, adjusting sea ice freeboard for discrepancies in mean snow depth, or stating ICESat-2 data is unusable? This needs more clarification here.

*To make this part clearer, we changed the text as follows:*

*"Another type of uncertainty related to estimating **sea ice thickness from the ICESat-2 elevation anomalies** is connected to **the presence of snow and, what is more important, to the generally unknown spatial and temporal variability of its accumulation rate**. The existing climatology or estimates of snow depths in the Arctic either do not cover the study area (Warren et al., 1999; Kwok et al., 2020b) or **are** too coarse to provide **a good spatial coverage** in Nares Strait (Rostosky et al., 2018; Glissenaar et al., 2021). Using DMSP/SSM/I-SSMIS brightness temperatures, Landy et al. (2017) reported >0.3 m mean snow depth in Kane Basin. However, as we will show later, this height seems to be overestimated and the more modest snow depth of 19±2 cm in Peabody Bay obtained from mean March-April AMSR2 data (Tedesco & Jeyaratnam, 2019) is thought to be a more reliable estimate **of the snow thickness in this area. The effect of the spatial variability of snow depth on the elevation anomalies remains unknown and is not accounted for in this study.**"*

- L149 – What is PHC3 climatology and JRA55-do? It hasn't been introduced. Also, need to write out "high resolution."

*We have specified that PHC3 is the **ocean state** climatology in that sentence. However, we don't think that an additional explanation is required for both PHC3 and JRA55. It's evidential that the atmospheric reanalysis data is used as an external forcing.*

- L160-163 – The phrase "We have to admit" can be cut. Additionally, it isn't clear from how this is worded whether this is a significant problem for the study or not, how it will create issues or not, or how

you mitigate the issues. My assumption is that this statement should read more like "Despite…, a dearth of bathymetry data in Kane Basin adds uncertainty to the models by… A large number of floating and grounded icebergs that originate from the Humboldt Glacier may also…."

*Cut as requested.*

*Unfortunately, we don't have a good answer to this question and we cannot do anything to mitigate a problem with the uncertain bathymetry. So, we could just state its existence.*

*We changed these sentences following your recommendations. Thank you for spending time to find a better wording!*

- L178-179 – What are the other parameterizations and why were they ruled out? I see this was mentioned later so it would be good to mention here that this analysis is included in a subsequent section.

*We changed the sentence to indicate where those parameterizations are exactly applied:*

*"The most often used parameterization of ice growth under average snow conditions in the Arctic was introduced by Lebedev (1938) in a form of $h(m)=0.0133×FDD^{0.58}$, although other possible parameterizations also exist (Bilello, 1961) and were considered **in Section 3.3**"*

- L188-189 – Why are sea surface brightness images not shown? It would be helpful to include these temps in a figure.

*We did not show these images because "…polynya signatures are less clear in those images because of their lower spatial resolution" (Line 234, Section 3.1). As an example, see the image below. It shows $T_b$ on 29 Jan 2021 (the ice bridge in place). One can see that there is a pronounced difference in the area around Cape Jackson compared to the surrounding ice, but the quality of this (one of the best!) image is relatively low to be used in the paper. In addition, it is difficult to determine whether the area around Cape Jackson is ice-free or covered with thin ice in these thermal images.*

[Figure]

- Figure 3 – Some of these images look like they may not have open ocean along the ICESat-2 tracks. Are all of the offsets being calculated by the same method in this figure?

*It's not true. The enlarged portion of Fig.3 is shown below to prove that open water (or very thin ice with negligibly small freeboards) is present in all panels. Please keep in mind while looking at these panels that the timing of the overlaid Sentinel-2 images and ICESat-2 tracks don't match and may differ by a few hours.*

[Figure]

- L233-234 – It is unclear what a "similar difference" in $T_b$ actually means. It would be helpful for this to be quantitative and to be accompanied by a figure.

*This sentence was changed as follows:*

*"The difference in surface height anomalies between the southeastern and northwestern parts of Peabody Bay is  **correlated with**  a similar difference in the observations of $T_b$.".*

*However, we think that the next two sentences along with referencing Fig.5 clearly explain what we mean under this similarity. The quantitative estimates of the $T_b$ difference are not given because the brightness temperature don't correspond to physical temperature of the ice surface, though they are strongly correlated.*

- The use of MODIS and AMSR temperature data is unclear. MODIS sea surface brightness temperatures were included in the study, but it isn't clear which product and no quantitative analysis were included in the manuscript. Tb was introduced as MODIS band 31 "sea surface" brightness temperatures, but Figure 5 references Tb as a "sea ice" temperature and the figure caption says it is from AMSR-E/AMSR-2 data.

> *This uncertainty seems to be the result of a few cases related to using wrong definitions in some parts of the text.*

> *We deleted "MODIS  brightness temperature" in the Data and Methods section. In the Fig.5 caption, the "ice surface temperature" was changed to "**brightness** temperature". We incorrectly used "ice" just because most of the strait is landfast ice-covered. Thank you for finding and pointing at these issues. We also went through the text to make sure that there are no other mistakes related to using "(ice/surface) temperature" where "brightness temperature" should be.*

> *Both MODIS and AMSR-2 brightness temperatures were used in our study: Figure 5 shows the AMSR-2 brightness temperatures ("AMSR-E" is a remnant from the earlier version that included that dataset - deleted) and Figure 10 - the MODIS brightness temperatures.*

> *The MODIS brightness temperatures in Fig.10 are for showing the contrast of surface water temperatures between different regions and point at their synoptic and/or interannual variability that may alter the ocean heat fluxes. This figure just visually supports the information mentioned earlier and doesn't require quantitative estimates.*

> *On the other hand, the AMSR-2 brightness temperatures in Fig.5 are for showing the correlation between the robust (all-weather permitting, though low-resolution) data on the thermal state of the surface and the observed anomalies of ICESat-2 elevations. The MODIS brightness temperatures can capture this linkage in individual images, but they are less convenient because of their strong dependence on cloudiness.*

- Figure 5 – Why is the $T_b$ only averaged for March instead of the same timespan as ICESat-2 (Jan-Apr)? The analysis would be stronger for a larger period and would make for an easier comparison to the ICESat-2 results.

> *Although there is no large difference between using only March compared to averaging over a longer period, we changed the timespan to January-March. April was excluded because the rising air temperatures (see Fig. 6a) and incoming shortwave radiation in the end of the winter lead to a significant increase of $T_b$ and, as a result, reduce the spatial contrasts.*

- Figure 9 – Error maps should be included here

> *Figure 9 represents the results of numerical simulation with the global oceanic model. There is no way of estimating and showing errors here.*

- There is currently no overlap between the ICESat-2 sea ice heights/AMSR temperatures (2019-2021) and the model results (2006-2010). It might be helpful to extend the AMSR temperatures back to 2006 to provide some comparison and context. It cannot necessarily be assumed that 2006-2010 have the same circulation conditions as 2019-2021.

> *We used the model of opportunity and were not able to choose a different simulated period.*

> *However, the main intention of the model was to demonstrate the circulation under the ice bridge. Since the main factors controlling water dynamics in this region (the along-strait sea-level gradient and the prevailing northern winds) don't vary a lot interannually, we reasonably suggest that the patterns shown in Fig.7-9 are generally valid and fairly represent (modeled) water dynamics and thermohaline state of Nares Strait in winter.*

- Figure 10 – It is unclear how the figure is making the main point the authors assert in L391-392. I believe the authors state that the warmth in Dec 2019 is probably related to the subsequent lack of ice bridge formation, or maybe "altered surface conditions" refers to something else? The ice bridge does form Jan-Apr 2020. So the early 2019 ice bridge failure to form should be unrelated to warm 2019 MODIS temperatures shown. Glancing at these images, this surface warmth was very short-lived and could have easily been associated with winds (synoptic-scale variability) moving sea ice away from the coast and causing sea-ice-free waters. Other years (e.g., 2018) have these same kinds of ephemeral open water conditions for a few days at a time. The authors would need to rule out that the MODIS Tb patterns derive from synoptic-scale variability here to make assertions about interannual variability here.

> *The main and only purpose of Fig. 10 was to demonstrate the existing spatial and interannual variability of surface water temperatures in the beginning of each winter. Taking into account the reviewer's phrase "Glancing at these images, this surface warmth was **very short-lived …",** we suggest that this figure was misinterpreted somehow.*

> *To make the message clearer, we altered the text in Lines 469-475 (Section 4.1):*

> *"The **MODIS** brightness temperatures, $T_b$, shown in Fig. 10 **generally** support the **idea that the thermal state of the ocean surface in Nares Strait varies interannually.** In December 2019 (Fig. 10b), the high $T_b$ conditioned the ice-free (or covered with thin ice) area **in the northwestern part of Peabody Bay and at the eastern side of Kennedy Channel**. Although the signatures of warmer water **in Kennedy Channel** can also be traced **through leads within the mobile sea ice** in December 2018 and 2020, $T_b$ was observed to generally be lower **and may indicate reduced ocean heat transport towards the surface from below**."*

- L456-474 – Are there more grounded icebergs in north Peabody Bay than in the south? I think this argument could be more succinct with references and this paper should be cited here. Theoretically the iceberg basal melt could create its own polynyas and bring AW all the way to the surface: Moon, T. et al. Subsurface iceberg melt key to Greenland fjord freshwater budget. Nature Geoscience 11, 49–54 (2018).

> *We didn't analyse the amount or density of icebergs in different parts of the bay carefully, but most of them seem to be grounded at the eastern flank of the mid-basin underwater ridge in Kane basin.*

*The mechanism of dissolution suggested in our paper doesn't suggest an intensive basal melt and, therefore, strong freshened plume dynamics that may deliver warm bottom water to the surface.*

*Anyway, this part of the paper was removed according to the other reviewer's recommendation.*

- L482-489 – This paper should support the notion that AW may come closer to the surface in the west in wintertime: SHROYER, E., PADMAN, L., SAMELSON, R., MÜNCHOW, A. & STEAS, L. Seasonal control of Petermann Gletscher ice-shelf melt by the ocean's response to sea-ice cover in Nares Strait. Journal of Glaciology 1–7 (2017) doi:10.1017/jog.2016.140. However, this study under review did not find warmer ice/ocean temperatures along the western edge of the strait and the polynyas in the west only open in early summer once the melt season has begun. It would be good to have a few more sentences of discussion on this.

*It's an interesting point. We didn't consider this effect closely, but the reviewer is right – we have to highlight that upwelling of water above the southward jet in winter may also contribute to the formation of thinner ice along the western coast in winter. Thank you for bringing this up.*

*The followed sentence was added in Lines 581-586 (Section 4.2):*

*"We suggest that the observed negative anomalies are attributable to the heat **transferred towards the base of the landfast ice** from **either upper thermocline mainly consisted of Pacific Water in this area (Jones et al., 2003) or warm** underlying mAW. **The baroclinic adjustment of the ocean to the intensification of the southward current in winter induces upwelling above the core that may shift upper thermocline water closer to the surface along the Ellesmere coast (Rabe et al., 2012; Shroyer et al., 2017) and, as a result, forms favourable conditions for a larger heat transport to the bottom of sea ice here.**"*

Technical corrections:

- Where paragraphs start and stop are unclear at times. Please add a space between paragraphs.

*The whole text is formatted in accordance with the rules of Ocean Science journal with using of its Word template.*

- Figures – isobaths in all figures need to span the entire length of the channel if they are included. They randomly stop in some parts of some figures.

*Thank you for this comment. However, the isobaths stop not randomly, but on purpose. In some regions, the bottom topography is complex and dense contours would have resulted in a worse displaying of useful data (e.g. along the western coast, in Kennedy Channel and Smith Sound). The isobaths in these regions are shown in Fig. 1 and we think it's more than enough for general understanding of the regional bathymetry.*

*The only purpose of plotting the selected isobaths in Fig.4 and Fig.5 is to show the relative positions of the observed elevation and temperature anomalies relative to the central ridge in Kane Basin.*

- L11 – controlling should be controls

    *Changed. Thank you for finding this.*

- L12 – earlier than what? Earlier in the year?

    *Yes, the bridge tends to form earlier in the season. Corrected.*

- L15 – semicolon should be a comma

    *Corrected.*

- L25 – "into the North Atlantic"

    *Corrected.*

- L26 – What is the direction of the sea level gradient?

    *The sea level decreases from Lincoln Sea towards Baffin Bay, but we think this explanation is redundant. The southward flow can be maintained by this particular direction of gradient only.*

- L27 – "situation" might be better as system

    *We checked both ways and think that "situation" works better here.*

- L29-30 – Awkward, needs to be reworded

    *The sentence was changed as follows:*

    *"An average **annual** ice export of ~141 km$^3$  **in years when the ice bridge exists** is about half of that exported during **bridgeless** years  (Kwok et al., 2010)."*

- L33 – icebergs

    *Changed.*

- L35 – Kennedy Channel not introduced

    *Changed as follows: "Under-ice sonar measurements in Kennedy Channel **in the northern part of the strait** showed the modal peak of ice…"*

- L47 – during the last 15 years underscores

    *This part was re-written in respect to the comment from the other reviewer.*

- L47-48 – a shortening of bridge annual or seasonal formation

    *In the modified text, it became more clear that we meant a shortening of the bridge existence duration in this sentence.*

- L54-56 – Awkward sentence

    *This sentence was changed as follows:*

*"Despite NOW is believed to be one the most studied polynya,  the sensible heat polynyas associated with **an impact of warm subsurface water**  are **also well-known feature in the landfast ice-covered**  Canadian Arctic (Hannah et al., 2009)."*

*Changed to "…latent NOW **polynya**…".*

*Changed. Than you for correcting.*

- L63 – Might want to restate this as "importance" rather than being interesting.

*The reviewer is right. "**Important**" seems to be a better way to highlight our interest to those polynyas.*

- L65-66 – Awkward last half of the sentence. Is this meant to say settlements existed here because there were hot spots? Why would that be the case? Are you meaning hot spots as in open water areas (sensible heat polynyas) or do you mean it's warmer here?

*We meant **biologically productive** hot spots, of course. Changed as follows:*

*"Those polynyas … seem to be **highly biologically productive**  spots as they evidence for nearby prehistoric settlements …"*

*Fixed.*

*Corrected.*

*Corrected.*

*Sorry, but we could not understand what awkwardness the reviewer meant. The sentence seems to be fine, but we made some changes to make sure that the message is clear:*

*"However, **these observations were** limited mainly to Kennedy Channel and **may not apply to** the entire strait, **limiting their use in** identifying the processes maintaining **ice-free conditions** at Cape Jackson **at times** during winter."*

*To avoid this, we changed the sentence as follows:*

*"This study intends to partially fill these gaps **by examining** the ice-ocean interactions **that occur** under the bridge during winter, **and subsequently examine their influence on the** formation of polynya at Cape Jackson and other yet unknown invisible polynyas in Nares Strait."*

- L98 – the observational evidence of the polynya

  *Corrected.*

- L100 – It would be helpful for Peabody Bay to be introduced before this. Cannot start sentence with "And".

  *We added the reference to Fig.1 where Peabody Bay is indicated and got rid of "And".*

- L105 – This first sentence needs to be more descriptive. What are the datasets for?

  *We modified this sentence as follows:*

  *"In this study, we used remote sensing data from different satellites **to demonstrate the presence of thinner ice and ice-free polynyas in Nares Strait**."*

- L125 – Vicinity

  *Thank you for finding this typo.*

- L127 – ICESat-2, not ICESat

  *Changed.*

- L128 – what ïf that. Also, need to state why the method doesn't work for Nares Strait. The next sentence seems to imply there are no leads, but it needs to be clearly stated at least in the previous sentence. Also, remove the second 'therefore' in the next sentence

  *We could not understand what does "what ïf that" stands for.*

  *In respect to the rest of the comment, we changed the sentence as follows:*

  *"…this approach requires ICESat-2 tracks crossing a bridge from edge to edge what does not work **in the long and narrow ice bridge in** Nares Strait."*

- L132 – I'm not sure what manifested to be adjusted means.

  *We removed the word "**manifested**".*

- L135 – "Basin suggests"

  *Changed.*

- L138 – "seem to be" should be "are" or "may be" but if you have the data, I would guess you know and this statement should be more firm.

  *Changed to "are".*

- Figure 2 – Need to mention that this is MODIS imagery

  *The mention of MODIS imagery was added.*

- Figure 3 - ICESat-2 transects are hard to see and contextualize here because there are no y-axes, portions of the lines disappear with the dark background, and plot orientations vary with each snapshot. These need to be larger and may need a y-axis.

*Unfortunately, any attempt to make the elevation even slightly larger results in overlapping of data from adjacent tracks. The figure caption was extended to mention how to interpret the data from each individual track:*

*"Figure 3: The ATL07 elevations **(red lines)** from 3 strong beams overlaid the Sentinel-2 image of polynya in March-May 2020 and in March-April 2021. **The elevations are plotted as absolute deviations from the ground tracks (black lines) in direction normal to the tracks.** The open water was used as a reference level to calculate the corresponding offsets for each subset. **The red scale bar in each panel correspond to 50 cm elevation.** The inset histogram demonstrates the probability distribution of all heights shown in the panels a-f."*

Figure caption needs to mention what the black and red lines are.

*Done. See the previous comment.*

It would be useful to mark where sea ice thinning around the polynya begins. It isn't clear in most of these plots because of how narrow the plots are.

*We really don't know how to do what the reviewer asks for. The elevations start to raise right from the polynya and increase gradually until they flatten about 10 km away from polynya. However, it's difficult to choose the exact position where ice thinning starts.*

The 50 cm scale bars are the same for all so can be removed from all but one plot.

*We prefer to keep it in each panel because it helps to see a relative amplitude of elevation by comparing it to a nearby scale.*

- L205 – It is unclear what 1.56 and 1.14 m ice thicknesses are associated with/changed from?

*These thicknesses correspond to the different presumed snow-to-ice freeboard ratios. We added comma, to clearly separate two parts of this sentence.*

- Figure 5 - The rainbow color map is unintuitive for representing temperature differences; please use a more intuitive and color-blind friendly scale (e.g., monotonic or smooth diverging).

*The color map was changed for better reception by color-blind readers.*

- L234-235 – This sentence kind of comes out of nowhere and it is unclear why this is important.

*We don't really understand why the reviewer does not see the connection with the previous sentence. However, after addressing the other reviewer's concern, these two sentences were changed as follows:*

*"The difference in elevation anomalies between the southeastern and northwestern parts of Peabody Bay is  **correlated with** a similar difference in the observations of $T_b$. The mean AMSR temperatures in March for all three years are shown in Fig. 5 and highlight the*

*presence of warmer (thinner) ice in the northwest compared to colder (thicker) ice in the south**east**."*

*We hope it helped make this part more correct and clearer.*

- Figure 5 – It would be helpful for brightness temperatures to be converted to Celsius to be more intuitive.

*Kelvin scale is a traditional way of presenting brightness temperatures.*

These scales appear to be incorrect. I would expect temperatures to only vary by a few 10s of degrees, not more than 100.

*It is correct that surface temperature would vary across a smaller range than 100 degrees. However, we clarify that 89 GHz AMSR brightness temperatures are not indicative of surface temperature alone, but measure the radiance of microwave radiation that is expressed in units of temperature (K) of an equivalent blackbody. Therefore, brightness temperatures are influenced by a combination of surface temperature, emissivity, and reflectance of the surface. In Figure 5, these variables are influenced by the different types, thicknesses, and surface properties of sea ice, accounting for the wide range in scale values.*

- L270-278 – Are these modeling results from this paper or from something else?

*The results presented in these lines are not from the model, but from the empirical relationships connecting ice thicknesses and amount of freezing-degree days (see the method description in the last paragraph in Section 2.4). All these estimations were obtained in the frame of this paper.*

- L317 – What depth is subsurface referring to?

*Below 100 m. This is the depth used to divide the water column into two layers in Fig.7. Reference to Fig. 7b was added to this sentence.*

- Figure 7 – Red and green are not color-blind friendly. Please use another color other than green.

*The green color was changed to grey.*

- L406-408 – Good point.

*Thank you*

- L410 – knee deep and 2.25 m where?

*The honest answer is we don't know. The "southeastern part of Peabody Bay" is the most precise positioning that could be drawn from Dr. Kane's report published in 1856.*

---

## Author Comment (AC2)

In the following, the comments by the reviewer are in normal blue characters and our responses to the comments are in cursive and indented. Modifications to the text are shown in quotation marks with bold characters indicating newly added text, and normal characters indicating text that was already present in the previous version. The line numbering in our responses corresponds to those in the revised manuscript with the tracked changes.

The role of oceanic heat flux in reducing thermodynamic ice growth in Nares Strait and promoting earlier collapse of the ice bridge

Sergei Kirillov, Igor Dmitrenko, David G. Babb, Jens K. Ehn, Nikolay Koldunov, Søren Rysgaard, David Jensen and David G. Barber

Overview

Nares Strait is an important oceanic connection between the Arctic Ocean and the Atlantic. It carries a sizeable fraction of the total outflows of Arctic surface and Pacific waters and provides a quick exit for thick old ice leaving the "last ice area". The rates of outflow of both seawater and ice are reduced when the strait is covered by fast ice, a condition of changeable duration that occurs in many but not all winters.

This paper explores factors that may reduce the viability of shore-to-shore fast ice in Nares Strait during winter. The discussion is based upon ice-cover observations acquired solely by satellite-based remote sensing instruments. These include Sentinel-2 SAR imagery (50 m), MODIS "true colour (250 m)", MODIS "mid-infrared brightness temperature (1000 m)", Sentinel 2 "high resolution optical imager (10 m)", AMSR "89-GHz brightness temperature (6250 m)" and IceSat's Advanced Topographic Laser Altimeter System "ice-plus-snow elevation (60 m)". No contemporary "ground-truth" data on the ice cover or the ocean were collected. A 1D sea-ice thermodynamic model was used in an attempt to distinguish the separate contributions of ice and snow to measured elevation. Oceanographic insight was garnered from a run of the Finite volumE Sea ice Ocean Model v2 for the 5-year interval, 2006-2010. The authors provide no indication that the viability of this model has been evaluated in Nares Strait.

The authors draw attention to a small polynya that forms off Cape Jackson on Greenland's coast at the southern end of Kennedy Channel during late winter of years when fast ice covers Nares Strait. They argue that this feature is indicative of a localized upward flux of sensible heat from the ocean to the ice. Using lidar data, they map a larger negative anomaly in surface elevation (ice plus snow) around the polynya. They use the measured elevations to constrain a 1D thermodynamic model of ice-plus-snow, driven by surface air temperature, to explore the complementary influences of ocean heat flux and snow depth in reducing ice-thickness. The "best match" corresponded to snow accumulation at 4-8 cm/mo and a 10-20 W/m2 heat flux from the ocean. Atlantic-derived water found below 150-m depth was proposed as the source of this heat, delivered directly to the underside of sea ice via upwelling. The paper also documents a band of relatively low ice-plus-snow surface elevation along the eastern coast of Ellesmere Island during two years with fast ice, but this was not observed during the one studied year (2019) when ice was mobile throughout the winter. Upwelled Atlantic-derived water was proposed as the source of this anomaly also. In closing remarks, the authors speculate that the zones of thermally weakened fast ice that they have identified on both sides of Kennedy Channel and Kane Basin in late winter weaken the stability of fast ice along the full 550-km length of Nares Strait and promote its earlier collapse in summer.

Assessment overall

The authors make a useful contribution in drawing attention to the influence of oceanic heat flux on the fast ice cover of Nares Strait. Oceanic heat flux has been shown to have a noticeably impact on the fast ice cover of the Canadian polar shelf, particularly in shallow waters with strong tidal currents where small polynyas form (Topham et al., 1983 [JGR 88(C5);]; Melling et al., 1984 [CSR 3(3)]; Melling, 2002 [JGR, 107]; Hannah et al., 2009; Melling et al., 2015). It might indeed be surprising if such features were not found along Nares Strait.

They make ingenious use of information from a variety satellite remote sensors to document ice-cover characteristics and state of motion, to detect polynyas within fast ice, measure ice-plus-snow elevation and map surface temperature. However no in situ data are brought into play. This is regrettable. In consequence, the accuracy, precision and possible bias of elevation and snow-surface temperature measurements for example were not determined, so that confidence in these data and the results derived from them is eroded. Moreover, the critically important separate contributions of ice freeboard and snow depth to elevation are not known, although the authors make a valiant effort to generate "educated guesses" through use of a 1D thermodynamic ice model; unfortunately, the "surface" air temperature used to drive this model was taken from the ERA5 re-analysis which has a 31-grid scale, much too large to achieve a realistic representation of surface weather conditions in Nares Strait.

The lack of contemporary or past oceanographic observations at the locations of interest is a serious shortcoming in a paper that strives to attribute polynya formation to oceanic heat flux. The FESOM global ice-ocean model has been harnessed in an effort to fill this gap. However, since this model neither assimilates contemporary ocean observations, nor seems to have been evaluated against existing ocean observations collected nearby, nor to incorporate tides, there is little basis for confidence in the minute (from a global perspective) thin-ice features that it is called upon to "explain".

In the particular instance of the Cape Jackson polynya, the authors could have saved themselves some trouble through a heavier reliance on Hannah et al. (2009). I examined CHS Chart No. 7072 to find a 43-fm (78 m) sounding 4 naut miles to the SW of Cape Jackson. I estimated depth beneath the polynya as half this, since the polynya is centered about 2 miles off the cape. WebTide (https://www.bio.gc.ca/science/research-recherche/ocean/webtide/index-en.php) predicts a 1 m/s spring tide here, so that Hannah's tidal mixing parameter is 2.1. This is comparable to values at polynyas in fast ice across the Canadian polar shelf, where turbulence generated by energetic tidal currents moves heat upward from relatively shallow depth. It seems unnecessary to look to the weaker general circulation to lift warm water from depths 3-4 times greater.

I believe that this paper should be published. However at present, it strives to be too comprehensive, is too speculative and therefore too long. There is valuable information therein and some pioneering use of remote sensing, but these strong points don't shine forth as well as they should. Specific suggestions for changes are listed below.

> *With all respect to the reviewer we have to disagree with the last argument about the tidal origin of the Cape Jackson polynya. There is a principal difference between this polynya and the polynyas in the narrow passages of CAA discussed in Hannah's paper. First of all, the polynya at Cape Jackson is not constrained by landmass. But our main argument would be that there are other shallows in the area with relatively strong tidal currents (see figure/table below and also supplemental figures in the end of the document showing the current speeds predicted by*

*WebTide). Although the tidal mixing parameter is indeed higher at Cape Jackson, it is also remarkably high in all other positions with known depths. However, an effect of tidal contribution to sensible polynya formation also requires a continuous lateral heat inflow at depth. The reversible nature of tidal motion cannot provide a consistent heat inflow by itself. In addition, the calculated horizontal excursions over one-quarter of the tidal cycle do not exceed 7 km in all considered positions (colored circles in the figure below). Even at Cape Jackson, where the predicted tidal currents are the largest, such excursion may result in only ~30m (59 to 43 fm) vertical displacement – not very large to upwell much heat. Therefore, even if a strong vertical mixing associated with tidal currents takes place at Cape Jackson, there should be a consistent mechanism "pumping" the heat from depth to the shallow.*

[Figure]

**Bathymetry in the vicinity of Cape Jackson. The circles show horizontal excursions over one-quarter of the tidal cycle (during spring tide) in several shallow regions based on WebTide current speed predictions.**

**The tidal mixing parameters. The colors correspond to the positions from the figure above**

|  | TideMarker 1 | TideMarker 2 | TideMarker 3 | TideMarker 4 |
|---|---|---|---|---|
| Depth, fm | 43 | 35 | 44 | 29 |
| Spring tide, m/s | 0.48 | 0.34 | 0.24 | 0.36 |
| Horizontal excursion, km | 6.8 | 4.8 | 3.4 | 5.1 |

| Mixing parameter | 2.85 | 3.20 | 3.79 | 3.06 |
|---|---|---|---|---|

*However, we have to admit that our suggestion that upwelling brings warm deep water from Peabody Bay **directly to the surface** may be too challenging. In combination with vertical mixing, it may be sufficient to upwell this water just closer to the surface – to the bottom layer over the "ridge" dividing Peabody Bay and the central channel of the strait. We changed the sentence in* *Line 536* *(Section 4.1) accordingly.*

*"This heat may **either be** upwelled **over the mid-basin ridge closer** to the surface*  **and/**or transported upward **to the lower surface of sea ice (or to the ice-free polynya)** by vertical mixing."*

Comments (major)

1. The authors have chosen to refer to the fast ice that covers the full 500-km of Nares Strait during many winters as the "ice bridge": The terminology is confusing because a long strip of fast ice does not resemble a bridge. I believe that most readers will consider the bridge to be the arch that forms the boundary between fast and mobile ice, most often in southern Kane Basin. As in masonry, the arch is strongest geometry for a load bearing structure because it is everywhere under compression, thereby exploiting the stress-state where sea ice is strongest. I recommend that the authors devise a different term to refer to locations within the fast ice "above" the arch. For example, at Cape Jackson, "more than 200 km north-east of the bridge".

*(This is also an answer to the reviewer's minor comments to lines 67, 94 and 130)*

*We understand this reviewer's concern about the terminology. However, even from a simple geometric point of view, arch is not an areal object but a line. In this research, we follow the terminology that was used in our previous paper (Kirillov et al., 2021; JGR) where we specifically point at the difference between an ice arch and an ice bridge: "…instead of using the term "arch", we prefer to use the term "bridge" for this structure in general and use "arch" to describe the characteristic dome-like shape of the bridge's leeward (southern) edge". We considered a bridge as an object connecting two opposite shores. From this point of view, "ice bridge" seems to be a good term for what we observe in Nares Strait. Historically, the landfast ice in Nares Strait was used as a migration route for Inuit. Mathieu Plante, an expert in sea-ice rheology who reviewed that paper, specifically mentioned that "it seems that there is no consensus on the terms ice arches and ice bridges" and agreed (supported) with our way of distinguishing them.*

*It might also be worth mentioning that there are some examples of the real bridges with width exceeding their length considerably.*

*We added the following footnote in Introduction to address this concern:*

**"In the absence of established consensus on the terminology, hereafter we prefer to use the term "bridge" for landfast ice blocking Nares Strait instead of "arch" which is used to describe the characteristic dome-like shape of the bridge's leeward (southern) edge."**

2. Line 148, "The model was driven by the atmospheric reanalysis fields from JRA55-do": This was probably not a good choice. Samelson and Barbour (2008) concluded that a grid 10x finer than that of JRA55 was required to correctly represent weather conditions in Nares Strait. I recommend adding a discussion the capability JRA55 to represent the mesoscale meteorology of Nares Strait, a channel much narrower than 55 km in width for much of its length, bordered by high terrain and characterized by a strongly stable atmospheric boundary-layer – the Arctic inversion – during the freezing season. This could be perhaps achieved via comparison of simulations by the ERA55 and Polar MM5 models.

*We agree that hi-resolution atmospheric models better represent weather conditions in the narrow Nares Strait with a strong impact of steep surrounding topography. However, a relative impact of wind forcing on water dynamics in the strait (not at CATs transect only) remains generally unknown and it's the along-channel sea level gradient that is thought to be a main factor controlling southward ice and water transport (Munchow and Melling, 2008).*

*Unfortunately, without running additional model experiments to investigate the effect of more realistic regional wind forcing on circulation in the strait, we can only try to address this reviewer's concern by comparing the results of our model of opportunity (FESOM-2) and results previously reported by Shroyer et al. (2015) who used wind forcing from the high resolution regional atmospheric model (after Samelson and Barbour, 2008). The figure below represents the along-channel velocities at two transects: across Kennedy channel at CATS mooring line and across Smith Sound. Both simulations show generally similar vertical structure of along-channel flow and fairly resembling current speeds. Slight differences might be attributed to different time interval used for simulations (2006-2010 in our paper, and 2005 in Shroyer et al., 2015).*

[Figure]

*To underline the fact of good correspondence between FESOM2 results and results obtained by Shroyer et al. (2015) and, therefore the suitability of using the low-resolution atmospheric forcing, the following sentence was added to the text (Line 177, Section 2.3):* **"Despite using a low resolution JRA55 product that is probably unable to reproduce orographic strengthening of wind within the relatively narrow Nares Strait (Moore and Våge, 2018), the modelled current velocities in Kennedy Channel and in Smith Sound fairly coincide (not shown) with velocities obtained by Shroyer et al. (2015) who used the Polar MM5 regional atmospheric model, which has a finer horizontal resolution of 6-km."**

3. Line 166, "The 6-hourly records of 2 m air temperature, wind speed and humidity in Kane Basin were taken from the ERA-5 global reanalysis database": The 31-km grid of ERA5, comparable to the width of Nares Strait, does not come close to resolving the channel and very its steep surrounding terrain. I suspect that much of the strait's "sea surface" resolved on a 31-km scale will actually be above sea level and therefore "terrestrial". It is very difficult to accept that ERA5's 2-m air temperature values hold much credence for simulating ice growth in the real world. Since sea ice dissipates the upward flux of latent heat from wintertime freezing by long-wave radiation from its top surface, would you not be better using satellite-derived surface radiation temperature to model freezing? Please strive to persuade readers that my viewpoint is invalid.

*We agree that the (relatively) low spatial resolution of any global atmospheric reanalysis makes it difficult to consider their products as good proxies of atmospheric conditions in Nares Strait (e.g. Moore and Vage, 2008). However, we would like to explain here our logic and justify feasibility of using ERA5 data in our ice growth model.*

*The key goal of using ice growth model in our study was not to obtain an absolute ice thickness in a certain position, but "to investigate a possible joint effect of snow and ocean heat flux on the observed spatial variations of surface heights in the vicinity of Cape Jackson" (i.e. over a distance of ~10 km or so). Despite a possible bias in ERA5 data in Nares Strait, a spatial variation of any meteorological parameter over such a short distance is negligibly small. Therefore, any local anomalies of ice thicknesses here are thought to be controlled by spatial variations of ocean heat flux rather than varying meteorological parameters. It means that although our estimates of absolute sea ice thicknesses presented in Figure 6b and 6c could have been biased, the difference of elevations presented in Figure 6d (and it is specifically mentioned that this plot is particularly important) are exposed to these errors to considerably lesser extent.*

*To underline this aspect, we added the followed sentence to Line 210 (Section 2.4):*

**"Although the spatial resolution of ERA5 reanalysis data is relatively low to resolve orographic effects in the narrow and steep Nares Strait (Moore and Våge, 2018), the key goal of the ice growth modeling was not obtaining the absolute ice thicknesses, but reproducing the spatial variations of combined ice and snow surface heights in the vicinity of the polynya. From this perspective, even though the modelled absolute ice thicknesses calculated with using ERA-5 data may be meteorologically biased, the accuracy of meteorological data seems to have a considerably smaller effect on the investigated spatial differences of ice thickness compared to unknown snow accumulation rate and possible spatial variations of ocean heat flux."**

*Another aspect that we would like to point at is that the magnitudes observed anomalies of surface elevations around polynya (ICESat-2 data, Figure 3) were very similar in 2020 and 2021. It implies that interannual variations of air temperature and/or wind speed may insignificantly affect the maximum ice thickness at the end of winter and that it's the ocean heat flux that seems to determine the thinner sea ice around Cape Jackson. Therefore, using just realistic meteorological parameters is believed to be fine for reproducing the ice growth during winter. The unknown snow accumulation rate and its wide range used in the model is believed to have much stronger effect compared to the coarseness of ERA5 grid.*

*To demonstrate that ERA5 parameters are realistic, we generated here two histograms showing probabilities of winter (December-April, 2014-2018) air temperatures and wind speeds measured with Automatic Weather Station set on Hans Island (red line) and from the nearby node of gridded ERA5 product during the same period. AWS on Hans Island is installed at 168m elevation (mean pressure 991 hPa), so for comparison we used the closest 1000 hPa level from ERA5 dataset. It was found that, despite some differences between reanalysis and measured data distributions, the means of both parameters are not largely biased. For instance, the mean measured wind speed was 7.1 m/s that is almost equal to 7.0 m/s in ERA5. For the mean air temperatures these numbers are -18.0C and -18.4C, respectively.*

[Figure]

*Summarizing our response, we would like to say that, although we generally agree that ERA5 reanalysis is not the best dataset for reproducing meteorological data in Nares Strait, we found that ERA5 wind and temperatures are only slightly biased with observations. The effect of such biases on the local thinning of sea ice is negligibly smaller then unknown snow accumulation rate and possible spatial variations of ocean heat flux – the parameters which effect was investigated with the ice growth model.*

4. Line 415, "It was found that the ocean heat flux at Cape Jackson needed to exceed 200 W m-2 to open the polynya as early as in March": The message intended here is unclear from the present text. Revision is required.

I don't believe that it is plausible for a polynya to suddenly melt itself into existence – oceanic heat fluxes just aren't large enough. The more likely role of oceanic heat flux is keeping the ice relatively thin (and relatively weak), so that more powerful mechanical (fracturing, rafting, flooding and downstream

advection, etc.) and thermodynamic (radiation) processes can do their work. Principal among the former are the stresses exerted by wind, wind-waves and tidal current on already thin ice. Once these open a polynya, new ice (mainly frazil and nilas) created by high rates of heat loss (> 200 W m-2) from the surface will continue to be removed by current, while insolation and downwelling short-wave radiation may deliver appreciable heat directly to seawater. This "tag-team" approach to creating a polynya in fast ice, involving both dynamics and thermodynamics, has been discussed by Topham et al., (1983. JGR 88).

Actually lifting warm water 100 m or more to the surface to open a sensible-heat polynya takes a large input of kinetic energy to the ocean, which cannot occur with a continuous cover of fast ice. It is only possible when strong winds act on mobile ice or open water. There is a paradox because necessity for strong winds is the same requirement for the opening of latent heat polynyas; the distinction between the purported "two types" of polynyas is not clear (see discussion in Melling et al, 2001). Moreover, because the wind must be integrated over a large expanse of mobile pack ice to accumulate enough kinetic energy to drive upwelling, the formation of small polynyas in fast ice, such as that off Cape Jackson, via this mechanism is unlikely.

*The estimated heat flux was obtained based on the fact of presence of open water near Cape Jackson in early March. Below, we give an example of one of the earliest (March 16, 2020) good Sentinel-2 images showing polynya in more details. One may clearly see the presence of thin new ice along with the areas of open water. If water surface is ice-free, 200 W/m² does not seem to be extraordinarily high.*

[Figure]

*The mechanical processes maintaining water at Cape Jackson ice-free suggested by reviewer can work to a certain extent only. Occurring within very small, landfast-ice constrained area, none of these processes may prevent the polynya to eventually become covered with ice unless the ocean heat flux is strong enough to melt it. From this perspective, the situation at Cape Jackson is thought to be different from the polynya in Pioneer Channel discussed in Topham et al, 1983. The authors showed that the polynya was kept ice-free by a removal of new ice by strong wind (and, we suppose, by extremely strong tide with up to 1.2 m/s current speeds) that pushes newly formed ice beneath the landfast ice sheet. If such process had taken place at Cape Jackson, the*

*continuous accumulation of large amounts of frazil/new ice from polynya below landfast ice sheet would have resulted in much thicker surrounding ice close to edge (at least in certain directions). However, ICESat-2 data shows a more less linear increase of ice thickness from ice-free polynya area in all directions (Figure 3).*

*To address the reviewer's concerns, we changed the sentence in Line 497 (Section 4.1) as follows: "Several simulations with different snow accumulation rates and ocean heat fluxes were run to find an optimal combination of these parameters to match **the observed modal surface height of** 0.26 m **near the Cape Jackson** polynya **(Fig. 3). These simulations were made under consideration that the polynya is kept ice-free during winter by a large (>200 W m-2) ocean heat flux.** It was found that the ocean heat flux at Cape Jackson needed to exceed 200 W m-2 to open the polynya as early as in March. Such large heat flux **within a relatively small polynya area** seems to be associated with a **local** upwelling **and followed mixing** of warm core of the southern branch of mAW rather than with vertical mixing **alone."***

*We also added the followed sentence in Line 230 (Section 3.1): "**The appearance of open water near Cape Jackson in March is also evident in a few high-resolution Sentinel-2 images obtained in 2020 and 2021 (Fig. 3).**"*

*And we also changed the sentence in Line 358 (section 3.3): "**Within the polynya, in order to have open** water in May the heat flux should reach 70 W m$^{-2}$, **while a heat flux** above 200 W m$^{-2}$ **is required to form an ice-free polynya** to let polynya form in early March"*

*Addressing the other reviewer's comment, we would like to say that we never attributed upwelling near Cape Jackson to wind forcing. The kinetic energy is available from the consistent inflow of AW through Smith Sound. This flow has nothing to do, but overflow the ridge separating the semi-enclosed trough in Peabody Bay from the main Nares Strait channel. We understand that it's difficult to prove this suggestion without in-situ current measurements, T/S data and water sampling, but our whole paper is a combination of lines of indirect evidence that all together support the hypothesis of upwelled heat.*

*In respect to the concern about "lifting warm water 100 m or more", please see our comment in your Assessment overall. We changed the sentence in Line 536 (Section 4.1) as followed: "This heat may either be upwelled **over the mid-basin ridge closer** to the surface (leading to formation of sensible heat polynya at Cape Jackson) **and/**or transported upward **to the lower surface of sea ice (or to the ice-free polynya)** by vertical mixing."*

**5. Lines 447-473: The speculation about iceberg melting provides an intriguing diversion, but it doesn't add much to the concepts central to this paper. I suggest that it be removed from the paper.**

*We agree with the reviewer. Although the process of iceberg base dissolution may support the idea of a steady ocean heat transport in the bottom layer of Peabody Bay, we don't have any solid evidence of its occurrence. Therefore, we have to admit that it adds little to the main idea of our paper. This part was removed completely.*

6. Section 4.2, "The formation of ice thickness anomalies along the western coast of Nares Strait": This is indeed an interesting feature. It did occur to me that is might possibly be an artifact in the southern half of the strait to the authors' referencing of elevation to the mean value measured along the full length of the strait. Has the possibility been investigated that the higher sea level in the north that drives the current might explain this anomaly?

*Although using the along-track anomalies generates interesting results, we do understand all its weaknesses. It can be used only at short distances where dynamical/steric variations of SLH are relatively small. The narrow Nares Straits is believed to be such a place, but even here the along-track and tracks-to-tracks changes of sea level may affect the obtained results.*

*However, the wedge of negative anomalies along the western coast is certainly not related to this weakness. The observed coastal zone is too narrow and has a very strong local gradient of elevations for being explained by spatially varying reference level. It's also worth noting that the pattern and magnitude of anomalies are very similar for descending (~along the strait) and ascending (~across the strait) ICESat-2 tracks (see figure below).*

[Figure]

[Figure]

*Figure. The ascending (a-c) and descending (d-f) along-track anomalies of ATL07 heights averaged over 1.5x1.5 km cells in January-April 2019 (a, d), 2020 (b, e) and 2021 (c, f).*

*We changed the Line 277 (Section 3.2) for making the latest aspect clearer:*

*"This discrepancy is partly attributed to the averaging of data **from both ascending and descending tracks (ICESat-2 repetition cycle is 90 days) that were** used to compute the **mean h̃ in each 1.5x1.5 km cell**, but **it is** a combining **of both these tracks** with different along-track regional means **together that seems to** result in **some** smoothing of spatial anomalies presented in Fig. 4. **Note, however, that both the patterns and the magnitudes of anomalies are very similar when calculated from only descending (~along the strait) or ascending (~across the strait) tracks (not shown)."***

Alternatively, could it be a manifestation of the undercurrent that hugs the Ellesmere coast when fast ice covers the strait? It is interesting that the elevation anomaly has roughly the same 10-km width as the undercurrent mapped by Rabe et al (2012; Fig. 4b). A connection to the dynamic relief reflecting geostrophy in this flow is implausible; it is only about 10% of the measured 20-cm drop in ice-plus-snow elevation adjacent to the coast. The inverse barometer effect, which might also contribute to lowered sea level in response to higher SLP at the western shore (not resolved at this scale by ERA5) is also too small: Sea-level pressure is only higher by 2-4 mb on the Ellesmere side of the strait (Samelson and Barbour, 2008: Fig. 8). Nonetheless, I recommend noted both possibilities as having been explored.

*We appreciate the reviewer's help in attempting to attribute the observed anomalies to factors other than ocean heat flux. We agree that the suggested factors should not have a large effect on the observed anomalies. The inverse barometer effect is believed to form a smooth gradient of sea level across the entire strait, not a large gradient within the narrow 10-km wedge and a relatively smooth pattern over the rest of the channel. A connection to a geostrophic balance is also implausible. The mentioned dynamic relief of the geostrophic flow is less than 2 cm across the channel (Munchow et al., 2006), but sea level rises towards the coast of Ellesmere coast though it has to be opposite to explain the negative anomalies in ICESat-2 data.*

*Also, see our answer to the minor comment to Line 133-134) and the proposed changes in the text.*

7. Line 484, "We suggest that the observed negative anomalies are attributable to the heat upwelled from the underlying mAW": I believe that this suggestion has merit, but that the details are incorrect.

The flow structure beneath fast ice in Nares Strait depicted by Rabe et al (2012; Fig. 4b), displays a jet of roughly 10-km width against the Ellesmere shore, centered at about 80-100-m depth. The baroclinic adjustment of the ocean to this jet (not shown) involves downwelling below the core of the flow and upwelling above. This leaves no mechanism to raise mAW through the core of the undercurrent to the surface on this side of the strait. Indeed, the cross-strait circulation that compensates for downwelling of mAW on the western side is upwelling on the other side, near Greenland!

However, upwelling does occur above the core of the jet. This would bring Pacific Winter Water as much as 0.2C warmer than the surface-freezing temperature (see Melling et al. 1984 Cont. Shelf Res 3) to the base of the surface mixed layer (see Melling et al. 1984 Cont. Shelf Res 3). This sensible heat in this Pacific Water could provide a heat flux to the underside of the sea ice via entrainment into the turbulent surface mixed layer. The needed turbulence kinetic energy could originate in part from brine-driven convection (ice growth) in the mixed layer and in part from shear between rough immobile ice and the rapid tidal flow. Melling et al. (2015) estimated an oceanic heat flux to the base of ice as 15 W/m2 under similar circumstances in Penny Strait, which would be sufficient to melt about 0.5 cm/d from 3-m ice (with 10-cm snow) at -25C. It should be noted that the submerged jet, and the upwelling above it near the western shore, do not exist when the ice is moving, so that the oceanic flux would be much reduced in years without a fast-ice cover.

> *The word "upwelled" was definitely used there by mistake. We didn't really imply that it's upwelling that is responsible for heat transport towards the surface along the western coast of Nares Strait. The next sentence in the text clearly showed what we meant. We changed "**upwelled**" to "**transferred**" and also mentioned Winter Pacific Water as a possible source of sensible heat transferred to the base of the landfast ice along the western coast. Citations of Münchow et al. (2006) and Jones et al. (2003) were added.*
>
> *However, we agree with the reviewer that the mentioned upwelling above the core may facilitate the heat from the Pacific Water layer (which warmer temperature may just reflect an upstream mixing with the underlying mAW layer; see our answer to the major comment #12) reaching the base of sea ice in winter. The followed sentence was added in Line 581 (Section 4.2):*
>
> *"We suggest that the observed negative anomalies are attributed to the heat **transferred towards the base of the landfast ice** from **either upper thermocline mainly consisted of Pacific Water in this area (Jones et al., 2003) or warm** underlying mAW. **The baroclinic adjustment of the ocean to the intensification of southward current in winter induces upwelling above the core that may shift the upper thermocline water closer to the surface along the Ellesmere coast (Rabe et al., 2012; Shroyer et al., 2017) and, as a result, forms more favorable conditions for a larger heat transport to the bottom of sea ice here**."*
>
> *We deleted few sentences further in the text:*

*"The temperature in the northern branch of mAW in Kane Basin is about 0.3 °C higher compared to the temperature in the southern branch (Fig. 9c, d). In combination with higher current velocities within the subsurface jet (Fig. 9a, b) this would result in increased shear instabilities within the flow and higher upward heat flux that may have considerably stronger impact on ice growth compared to the northwestern Peabody Bay."*

*and modified sentences in Line 607:*

*"Transformation of these currents over steep topography generates baroclinic semidiurnal tidal wave that may considerably enhance vertical mixing through benthic stresses and shear instabilities (Davis et al., 2019). From this perspective, the fact that most of the western polynyas first appear near prominent headlands (Fig. **11**) generally support the idea **that the enhanced heat fluxes along Ellesmere Island are attributed to the** topographically controlled instabilities associated with the mean current and reversible tidal flow. **Another mechanism that may enhance the heat flux in the area is associated with the sub-ice turbulence generated by interaction of tidal flow and the very rough under-ice topography (Ryan and Münchow, 2017). In combination with the upwelling of the upper thermocline water along the western coast in winter (Shroyer et al., 2017), this mechanism may result in a considerable increase of vertical heat flux towards the bottom of sea ice."***

8. Lines 504-505, "Transformation of these currents over steep topography generates baroclinic semidiurnal tidal wave that may considerably enhance vertical mixing through benthic stresses and shear instabilities": I suggest that the steep cliffs on the western shore, indicative of deep water close to shore, make turbulence and internal waves generated in the benthic boundary layer irrelevant to the ice far above. However, I believe there is a good possibility to generate strong turbulence, mixing and entrainment through the action of the tidal flow (Pite et al., 1995. JPO 25) on the very rough under-ice topography of Nares Strait (Ryan & Munchow, 2017).

I suggest that the authors give some thought to this alternate, and I believe more plausible, explanation for the source of ocean sensible heat.

*See our response to the previous comment and also the suggested changes in the text.*

*Not for the paper, but we want to point at on thing related to the idea of sub-ice tidal mixing that makes us confused. The semidiurnal tide forms a standing wave pattern in Kane Basin with the lowest tidal velocities in its central part (Davis et al., 2019). If so and if the negative anomalies along the western coast are tidally driven, they should have become smaller around Cape Frazer. However, we don't clearly see it in Fig. 4b-c.*

9. Line 523, "weakens the cohesion of landfast ice against the shoreline in Kane Basin": The tidal cycles in sea level ensure that the ice sheet is always fractured at the coast, not bonded to it. However, the word cohesion implies that the authors consider that bonding of ice to the shoreline is important. This line of thought runs contrary to decades-old discussions of fast ice in deep water, where it is the formation of ice arches across channels which stops the movement of ice behind them, not shear strength at the shoreline. The upwardly convex shape of a masonry arch is the key feature that allows it

to resist downward loading; the shape ensures that all the stone in the arch is under compression, the stress state in which it is strongest. Indeed the stress is highest within the wedge-shaded stones of the arch and much less above them. Pack ice also is strongest in compression and much weaker in shear. Although there are likely several arch-shaped load-bearing features distributed in the fast ice along the length of Nares Strait during any winter, much of the fast ice cover will be in a low state of stress; cohesion at the shoreline is probably unnecessary for fast-ice stability, although its confinement by irregularly shaped shorelines may constrain it from moving locally. Conversely, weakening of that confinement by melting at the coast may allow it to shift around in response to wind and tide. It is quite common to see the ice in Kennedy Channel become mobile between arches at its northern and southern ends long before the collapse of the arch in Smith Sound allows the ice in Kane basin to do the same. The same phenomenon is seen annually in Prince Regent Inlet. I don't think that the authors' argument for up-channel polynyas hastening the break-up of fast ice further down-channel has much merit, as presently written. It is possible, of course, that phenomena may be correlated in time because of the influence of a third circumstance not identified.

> *We understand this reviewer's concern about weakening the cohesion and its potential impact on ice bridge break-up. There is no doubt that it's the arch in Smith Sound that keeps the entire Kane Basin bridge in place. Indeed, we used the fact that breaking and moving of ice in the middle of bridge often occurs before the arch collapse. And suggested that it may facilitate the following collapse because mobile sea ice in the middle of bridge may gain more kinetic energy (from wind, mean flow fluctuations or tidal current) and put additional load on the arch that is becoming less and less strong in summer due to gradual thawing.*
>
> *We agree that more explanations are needed in the text. We changed the Line 628 (Section 4.3) as followed:*
>
> *"Based on the results presented here, we* **suggest** *that thinner coastal ice, formed under conditions of enhanced oceanic heat flux, weakens the cohesion of landfast ice against the shoreline in Kane Basin.* **We can further speculate that such weakening may facilitate** *an earlier ice bridge break-up (comparing to a supposed no-polynyas situation)* **as it leads to formation of patches of mobile ice in the middle of the ice bridge in Kane Basin. While shifting around, this ice may gain some kinetic energy from wind and tide and eventually result in additional dynamical load on parts of the bridge that still remain in place.***"*

10. Line 528-529, "*This break-up appeared to release internal stresses in the ice bridge and led to concomitant ice cover break-ups in the main channel*": This statement appears to rely upon a knowledge of the dynamical state of the ice cover. Nothing is known about stresses. In reality all you have access to is evidence of deformation (in the form of cracks) and of motion. Also see comment #9.

> *We agree with this criticism. The sentence was changed as follows: "This break-up appeared to* **initiate the further fracturing of** *ice cover in the main channel."*

11. Line 531: This paragraph gives the impression that the polynya has played a role in the breakup, but really all that you demonstrate is that the breakup was correlated with expansion of the polynya.

Perhaps the expansion of the polynya is just one event in the process. A more robust discussion, with a more useful take-away, would review the other factors in play, as listed in Line 520. If such completeness is thought to be beyond the scope of the paper, perhaps it should be covered in a separate paper. See comment #9.

*We added new sentences (Line 628, Section 4.3) explaining how polynyas and thinner ice may facilitate the bridge breaking-up (see our response to comment #9).*

*Although we agree that the investigation of a complex possible role of polynya(s) in bridge collapse requires additional research, we think it is worth at least mentioning such a hypothesis even based on a simple correlation of timing of polynya(s) development and bridge collapse. And we noted honestly that (Line 646) "although our hypothesis that polynyas facilitate ice bridge break-up in Nares Strait is speculative, we would like to emphasize the observations that the first movements of the immobilized ice cover occurred in areas with negative ice thickness anomalies during winter and where polynyas are observed."*

*We also changed the title of Section 4.3 to "The **inferred** role of thinner ice in Kane Basin in ice bridge break-up".*

12. Line 539-540, "The only oceanic heat source available to maintain such a polynya through winter is the modified Atlantic Water": This is not true. It may be the warmest source, but it is not the one closest to the ice. My comment on Line 484 raises the possibility that the less conspicuous warmth of the Pacific Water might be more influential than you give credit for. I recommend that you re-think the paper with this in mind.

*Although we don't support the reviewer's idea that colder Pacific Water plays a greater role as a source of sensible heat transferred to the sea ice, we agree that this source has to be mentioned. Our major initial mistake was related to not ignoring this water mass, but to suggesting that its heat is associated with a thermal impact of underlying Atlantic Water. Based on summer observations, Jones and Eert (2006) showed that surface mixed layer occupied ~90-100 m of the water column at the western side of Kennedy Channel. From this depth, where the fraction of Atlantic Water was 20-30%, water temperature started to increase. It made us think that the elevated temperatures in the Pacific Water layer below the surface mixed layer (within the upper thermocline) is simply attributed to this AW fraction and the upstream mixing with the underlying AW core.*

*Without specialized experiments and in-situ observations, we can only give credit to Pacific Water as another potential source of ocean heat limiting ice growth, but not provide solid proof of its greater input compared to mAW.*

*Addressing this concern, we changed the text in Lines 386 (Section 3.4), 526 (Section 4.1) and 581 (section 4.2) as follows:*

*"This flow occupies the entire water column and consists of 3 distinctive layers; i) cold brackish polar mixed water within the upper 50-60 m, ii) the **upper thermocline coinciding with** halocline **that is** observed at 70-110 m and (iii) the relatively warm underlying modified Atlantic Water (mAW) which originated in the North Atlantic and was transported a long way from Fram Strait*

*into the AO and to Northern Greenland (e.g. Melling et al., 2001).* **The first two layers mainly consist of water of Pacific origin (Jones et al., 2003; Jones and Eert, 2006).**"

"*The only available source of ocean heat in Kane Basin during winter is associated with the relatively warm modified Atlantic Water penetrating into the basin from the Lincoln Sea (northern branch) and Baffin Bay (southern branch).* **The inflow from the Lincoln Sea may also transport some heat with Pacific Water below the surface mixed layer. However, this heat may just reflect an upstream mixing with warm underlying mAW. For instance, based on the data collected in Kennedy Channel, Jones and Eert (2006) showed the fraction of Pacific and Atlantic Water in the upper part of thermocline at depth 90-100 m of about 70-80% and 20-30%, respectively.**"

"*We suggest that the observed negative anomalies are attributable to the heat* **transferred towards the base of the landfast ice** *from* **either the upper thermocline mainly consisting of Pacific Water in this area (Jones et al., 2003) or** *the warm underlying mAW.* **The baroclinic adjustment of the ocean to the intensification of the southward current in winter induces upwelling above the core that may shift upper thermocline water closer to the surface along the Ellesmere coast (Rabe et al., 2012; Shroyer et al., 2017) and, as a result, to form favourable conditions for a larger heat transport to the bottom of sea ice here.**"

*And also in the Abstract and Summary:*

"*This work provides new insight into the Nares Strait ice bridge, and highlights that* **an impact of warming** *modified Atlantic and/or Pacific Waters entering the Strait* **may** *contribute to its [bridge] further decline.*"

"*The ice thickness anomalies along the western coast were considered to be associated with heat released* **either from the upper thermocline water of Pacific origin** *or from the* **underlying mAW** *that carry relatively warm water southward from Lincoln Sea.*"

Comments (minor)

Line 27: Shokr et al. (2020) is a weak reference for the role of the along-channel sea-level in driving flow down Nares Strait. Münchow & Melling, J Mar Res 66, doi.org/10.1357/002224008788064612 would be much better.

> *The reference to Münchow and Melling (2008) was added.*

Line 30-31, "The ice bridge also helps prevent the loss of the thick, old ice from the Last Ice Area": The paper cited (Moore et al, 2019) is not helpful in substantiating this statement; it has very little to say about Nares Strait. To my knowledge, there has not yet been a study demonstrating that ice loss from the LIA, as distinct from ice export through Nares Strait, is reduced during years when an ice arch forms there. Nares Strait is only one of four pathways (and the narrowest) via which ice leaves the LIA – the others are to the NE via Fram Strait, to the SE through the QEI and to the SW to the Beaufort Sea. It is quite plausible that a blocked Nares Strait simply creates a diversion of ice to one of the other pathways, most likely Fram Strait. You need a citation that demonstrates convincingly that this is not so.

*The reference to Moore et al. (2019) here was used here only as a reference to the Last Ice Area, not to confirm the statement in the beginning of this sentence. To address this uncertainty, we just moved the reference into the relative clause and specified that the loss through Nares Strait was meant: "The ice bridge also helps prevent the loss of the thick, old ice* **through the strait** *from the Last Ice Area* , *located north of Ellesmere Island and Greenland* **(Moore et al., 2019)**, *by hindering its transport south, …"*

Line 38, "… peak in the fraction of sea ice with a draft between 2.6-2.8 m": It is important to note here, as was in the cited paper, that this range in draft was computed on the assumption of no snow cover, which may bias values appreciably high. Also, a referenced estimate of the empirical accuracy in draft estimates from CryoSat freeboard should be included here.

*We added that these estimates were made under no-snow assumption. However, 2.6-2.8 m range represents a mean characteristic that is not directly related to the accuracy of individual ICESat-2 readings. The same for the recent paper – we used the elevation anomalies averaged over 1.5x1.5 km cell (Fig.4) that means ~200 readings per a single pair of ascending and descending tracks or even more for repeated tracks. With a nominal accuracy of ICESat-2 measurements of few centimeters, the accuracy of the calculated elevation anomalies is at least ~15 times smaller (few millimetres).*

*The followed sentence was added in* Line 153 *(Section 2.2):*

**"Even though the accuracy of individual ICESat-2 readings is relatively high (less than 5 cm, Brunt et al., 2019), the accuracy of the averaged anomalies calculated with this method is estimated to not exceed a few millimeters."**

Lines 46-47, "That bridgeless years only occurred during last 15 years underscore a general shortening of bridge existence period and point to changes …": It would be appropriate to clarify that this statement refers to the absence of an ice bridge at Smith Sound (think) and not to the much smaller number of years when there was no bridge anywhere between Baffin Bay and the Arctic Ocean.

In this clarified context, it should then be noted that there was one winter (1995) in the 1990s with no arch at Smith Sound – in 1995 the arch formed at Hans Island – and one (1993) essentially like 2007 with no arch anywhere; "essentially" because an arch in Smith Sound that year lasted only 10 days (Vincent 2019). With a 30-year perspective, the record looks less amenable to interpretation via trend: there is a cluster of 2 of 3 years with no arch at Smith Sound in the mid-1990s, then an 11-y period with annual arches, then a cluster of 3 of 4 years with no arch in the 2nd half of the 2000s, then a 6-y period with annual arches, then a cluster of 2 of 3 years with no arch in the second half of the 2010s. Disregarding clustering and estimating the probability of no bridge in any year from the data as 7/31, one uses the Poisson Distribution to estimate the likelihoods of the observed gaps between no-bridge winter – that is having 2 no-bridge years in 2 years, 2 in 3 y, 2 in 7 y and 2 in 12 y. These are 6.4%, 11.7%, 25.7%, 24.4%. The low values for the small gaps suggest there is clustering in play; the relatively high values for the large gaps suggest that such wide gaps are not unexpected, so that bridging despite weak clustering, looks like a Poisson process. On these grounds I suggest a re-examination the statistical confidence of the statement in lines 46-47, which is based on such a short time series.

*Thank you for bringing all these details up. We agree that 1993 and 1995 have to be also referred as bridgeless years according to the data reported by Vincent (2019). At least for Kane Basin. We changed the corresponding lines and also specified that we are talking about Kane Basin in this paragraph:*

*"Analysis of 16 bridge formations during the past two decades2001-2021 revealed that consolidation occurred at cold air temperatures (less than -15°C), around neap tide, and during a cessation or even reversal in the prevailing north-northeasterly winds in the strait. However, the bridge* **in Kane Basin may have** *failed to form* **even under atmospheric and oceanic conditions that are favourable for consolidation (Kirillov et al., 2021).** *Based on AVHRR satellite data from 1979 to 2019, Vincent (2019) reported on a recent trend towards later formation and earlier breakup of the ice bridge.* **The fact that the ice bridge failed to form only two times during the first two decades of observational records (in 1993 and 1995; Vincent, 2019) and six times during last two decades (in 2007, 2009, 2010, 2017, 2019 and the last bridgeless winter 2022)** *underscore a general shortening of bridge existence period and point to changes in environmental conditions."*

Lines 48-49: I think that the date-based approach of Vincent (2019) is probably a more robust approach to a short 30-year time series than is the counting of the rare occurrences without arches, which the authors have used here.

*See the changes made while answering the previous comment.*

Line 54-56, "it is the sensible heat polynyas … that are more common in the Canadian Arctic (Hannah et al., 2009)": The authors appear to mis-quote Hannah et al. (2009), who state "… are widely distributed across the Canadian Arctic Archipelago"; Hannah at al. are clear that these sensible heat polynyas are features within fast ice in this region. Their map (Fig. 1) shows that the latent heat flaw-leads and polynyas that form along the perimeter of the fast ice are actually more widespread across the Canadian Arctic waters and occupy much more area.

*This criticism is fair. We used bad way of saying that sensible heat polynyas are commonly met in the Canadian Arctic. We re-wrote this sentence as followed: "**Beyond the NOW and other latent heat polynyas, there are several sensible heat polynyas that form within the landfast ice cover of the Canadian Arctic that are associated with warm subsurface waters opposing ice growth** (Hannah et al., 2009)."*

Line 67: Refer the reader to Fig. 1 for the mapped location of Cape Jackson.

*Done as requested*

Line 67 et seq., "… at Cape Jackson in the central part of the bridge": The terminology is confusing. I believe that most readers will consider the bridge to be the arch that forms the boundary between fast ice and mobile ice in southern Kane Basin. It follows that the central part of the bridge is the "top" of the arch, halfway across the strait between Greenland and Ellesmere. However in this sentence, the authors

are referring to a location in fast ice more than 200 km "above" the arch. I recommend that the authors devise a different term to refer to locations within the fast ice "above" the arch. Simplest in this example would be "… at Cape Jackson, more than 200 km north-east of the bridge".

*Please see our response to the first major comment.*

Line 93, "… maintaining water at Cape Jackson ice-free during winter": The reality is ""… maintaining water at Cape Jackson ice-free at times during winter".

*Changed as requested.*

Line 94 et seq., "under the bridge": See comment re line 67. I recommend using the phrase "beneath the fast ice" for the reason already given.

*See our response to the first major comment.*

Line 100: Line 67: Refer the reader to Fig. 1 for the mapped location of Peabody Bay.

*Reference was added.*

Line 130 et seq., "crossing the bridge": See comment re line 67.

*See our response to the first major comment.*

Line 132, "Although ATL07 data are manifested to be adjusted for geoidal/tidal variations and inverted barometer effects": The correction for the inverted barometer effect is probably only accurate in wide deep ocean basins where the long ocean wave which is the ocean's response to changing atmospheric pressure can move as fast as, and in the same direction as, the SLP anomalies moving at 20-25 m/s. I suspect that the correction will not work well in a long (550 km) narrow (35 km) strait. I urge the authors to find and reference research that provides a discussion of the accuracy of the inverted barometer correction in confined coastal waters.

> *This is a good point. Although the depth of the main channel (>200 m) allows long wave to travel with the speed of more than 45 m/s, we generally agree that the inverted barometer effect may work not very well in narrow Nares Strait. However, we don't think it somehow affects the obtained results. Even if the strait would have reacted "normally" to changing SLP (as in wide deep ocean), the width of the strait is too small for spatial SLP variations having a large effect on the cross-channel sea level difference. And all observed large coastal anomalies are either too small (at Cape Jackson) or too narrow (western coast) for being attributed to this factor. See our response to the major comment #6 for more details.*

Line 133-134, "… may still contain unknown uncertainties related to the regional synoptic variability of sea level associated with wind forcing and/or with ocean dynamics": With respect to the atmosphere, I recommend replacing "wind forcing" with "strong wind, air-pressure and ocean dynamical effects on the mesoscale (10-30 km)", referencing Samelson and Barbour (2010).

> *We don't really think that varying air pressure may considerably affect the amplitude of observed ICESat-2 anomalies, even if the inverted barometer effect is not adjusted well in Nares*

*Strait (see our response to the previous comment). The main discovered anomalies (at Cape Jackson and along the western coast) are relatively small/narrow for being affected by any of mentioned uncertainties from our point of view. But they have to be mentioned, of course.*

*We changed the sentence accordingly.*

With respect to the ocean, Münchow & Melling (J Mar Res 66) provide estimates of the anomalies of sea-level height relative to the mean. These have amplitudes as large as 10 cm along-channel and a few cm/s across-channel. These along-channel value is large enough to contribute appreciable fortuitous NE-SW varying anomalies in thickness that are computed relative to an along-track (approximately along-channel) mean. This source of error requires discussion.

*With all respect to the reviewer, we don't agree with this point of view. The reference to the sea-level differences obtained from the tidal gauges deployed at Alert and in the interiors of the fjords at both sides of Smith Sound can't be used as a good argument. All those gauges were installed in a relative vicinity to the mobile ice areas and, therefore, may contain a large portion of variability attributed to the dynamical effects associated with local wind and currents. One can see this variability in the Münchow & Melling's Fig. 14 as a continuous alteration of differences between positive and negative values. The magnitude of sea-level gradients below the central part of the ice bridge and its temporal variability in winter remains unknown.*

*However, it's not the main reason why we would like to reject the reviewer's concern. We have already mentioned that from our point of view the spatial configuration of the observed ICESat-2 anomalies doesn't admit any other explanation rather than the local ocean heat impact. Those areas are either too small (at Cape Jackson) or too narrow (along the western coast, aligned along-channel) for letting other factors explain the large gradients of anomalies observed in these areas.*

*The distance between individual ICESat-2 tracks is 3.6 km. It implies, that MOST of 1.5x1.5 km cells accumulate data from a single ascending and a single descending track. MOST, because there are some tracks that were repeated twice: ICESat-2 repetition cycle is 90 days and we used 120-day (January-April) period when calculating anomalies in Fig. 4. Even from this perspective, the random variation of cross- and along-channel sea-level differences (from Münchow & Melling) would have resulted in random elevation anomalies along adjacent tracks (=cells) made at different dates. However, the observed anomalies demonstrate a fair correlation between nearby cells and, moreover, the spatial pattern of anomalies was found to be very similar when calculated by using only descending or ascending tracks (see our response to the major comment #6). Such result would be highly implausible, if the along- or across-channel sea-level gradients (with their altering directions) determine the observed elevation anomalies.*

*To address this concern and show why we think sea-level gradients or SLP could not explain or even contribute much to the formation of anomalies, we added the followed sentences in Line 286 (section 3.2):*

***"A relatively short off-shore extension (about 10 km) of both coastal zones (along the western coast of Nares Strait and along the northern coast of Peabody Bay) eliminates the regional***

*variations of sea level as a factor contributing considerably to the anomalies. For instance, Samelson and Barbour (2008) reported the relatively small spatial gradient of sea-level pressure over the full width of Kane Basin corresponding to about 2 cm of sea level difference with higher level at Greenlandic side. A geostrophic adjustment requires less than 2 cm sea-level drop from Ellesmere Island to Greenland in Kennedy Channel (Münchow et al., 2006). Also, using the tidal gauge records at Alert and at the opposite sides of Smith Sound, Münchow & Melling (2008) reported the across- and along-channel sea-level differences varying in Nares Strait from a few centimeters to about 10 cm, respectively. However, these relatively large differences could be associated with the local dynamical effects as all bottom pressure sensors were deployed in shallow bays not far from the areas covered with mobile ice at Smith Sound and at Alert. The actual sea level gradients below the ice bridge in Nares Strait and their input to the observed ICESat-2 anomalies remain unknown, but are thought to be small comparing to the gradients associated with the anomalies observed along the western coast of Nares Strait and at the northern coast of Peabody Bay."*

Line 139-140, ">0.3 m mean snow depth in Kane Basin. However, as we will show later, this height seems to be overestimated". Reference to Samelson and Barbour (2010) is again appropriate, since the extremely strong winds common in Kennedy Channel and the vicinity of Cape Jackson (see also Melling, Oceanography Mag, 2011) may indeed provide a strong disincentive for the accumulation of snow.

*Thank you for this comment. However, here we are talking about the central Kane Basin and Peabody Bay. We don't think the orographic effect plays the same role in snow accumulation rates as it does in Kennedy Channel.*

Line 159, "… generally have good agreement with the mooring records": It is necessary to provide an assessment that is more specific in relation to the comparison of model with data in relation to the cross-channel scale of flow features, their positions cross-channel and in depth and their intensity. Can the countercurrent on the Greenland side be simulated?

*To address this concern, the followed sentences were added in Line 180 (Section 2.3):*

*"It was found that both vertical and cross-channel distributions of temperature/salinity and current velocities in FESOM2 simulations generally have good agreement with the mooring records in this region (Münchow & Melling, 2008; Rabe et al., 2010; Münchow, 2016). **For instance, the model reproduces well the shift of the southward jet towards the Ellesmere coast (at ~1/4 of the channel width) and also the existence of countercurrent on the Greenlandic side, although with lower velocities. The mean modeled temperatures and salinities both demonstrate the presence of cross-channel gradients towards Greenland that become stronger at depth that is in a good accordance with observational data. In addition, the model fairly reproduces the uplifting of isohalines and isotherms over western slope in winter.**"*

*We can confirm that the countercurrent on the Greenland side was fairly reproduced by FESOM-2. Similar to the simulation results of Shroyer (2015), this flow is only presented as undercurrent in winter. See the figure in our response to the major comment #2.*

Line 182, "MODIS imagery confirm that a polynya is present every winter at Cape Jackson": The sentence that follows that quoted indicates that the following is more precise: "MODIS imagery

confirms that in every winter when fast ice fills the strait, a polynya appears at Cape Jackson late in the season".

> *Thank you for helping to make this sentence clearer. It was changed as followed: "MODIS imagery confirm that **every winter when the ice bridge is formed in the strait** since the MODIS observations began in 2000**, a polynya **appears at Cape Jackson late in the season".*

Line 190, "may indicate either the ice-free surface or thinner ice": Clarification, "may indicate either the ice-free sea surface, locally thinner ice, locally thinner snow or both the latter".

> *Changed as requested. Thank you for this comment.*

Line 205, "… If 50% of the 0.26 m surface elevation is attributed to a snow layer …": The occurrence of very strong, very turbulent winds off sea capes is well known to mariners. Cape Horn and Cape Farewell, at the southern tip of Greenland, are perhaps the most famous. See Winant et al. (1988) J. Atmos. Sci. 45. Such conditions would be very effective at scouring snow from the surface of sea ice and moving it downwind. It is therefore quite plausible that both ice thickness and snow depth become thinner on approach to Cape Jackson, as the density-stratified oceanic and atmospheric flows accelerate in response to submarine and subaerial topography blockage, respectively. IceSat may be sensing environmental response to both these effects, not just to one or the other.

> *Thank you for this comment. However, in this paragraph we just demonstrated how much the snow layer could contribute to the ice thickness if the observed 0.26 m anomaly away of the polynya at Cape Jackson is partly associated with the snow.*

> *Further in the Section 3.3, we also used 1D ice growth model to investigate the effect of snow (and ocean heat) on ice growth. But we also applied different snow accumulation rates to simulate the ice growth at some distance from polynya. In the polynya, with very strong winds suggested by the reviewer, no-snow assumption was used. We made some changes in the text to get rid of these unclear parts and to underline that no-snow was used for modeling the evolution of ice thickness in the polynya.*

Lines 218-238 & Fig. 4, "along-track anomalies averaged over 1x1 km squares": On the "basin-wide scale" discussed here, the anomalies, calculated relative to mean height of any ascending or descending track crossing the bridge between 55-76°W and 78.25-82.5°N, may well be contaminated by a varying along-channel gradient is sea-surface height – see comment on lines 133-134. It is appropriate that the authors acknowledge this source of error and discuss its impact on results.

> *We already addressed this concern while answering the major comment #6.*

> *There was also a mistake throughout the paper. We used 1.5x1.5, not 1x1 km, mesh. The 1x1 km corresponded to the old version of Fig.4. Corrected everywhere.*

Lines 221-232 & Fig. 4, "In the main channel, the anomalies are highly irregular and form a speckled pattern, whereas the anomalies in Peabody Bay form a consistent pattern with positive anomalies in the southeast and negative anomalies to the northwest": It is unclear, with the continually moving ice of 2019, why the elevation anomalies are not smoothed out via averaging over time. The small scale of the speckle in elevation in 2019, not so different from that in the years with immobile ice is difficult to understand. Please explain.

*For averaging, we used 1.5x1.5-km cells in order to keep data from the individual ICESat-2 tracks (~3 km apart from each other) separated. Therefore, time averaging mainly means averaging of a pair of ascending and descending tracks. See our answer to the major comment #6 and the corresponding changes in the text for more details.*

*Using coarser mesh gives smoother pattern of anomalies in the main channel in 2019, but also results in vanishing gradients in the areas with large anomalies and strong gradients (see the anomalies calculated at 10x10 km mesh in 2019, 2020 and 2021 in the figure below). We preferred to show the latest.*

[Figure]

A similar speckled pattern of $h$ was observed over the landfast ice in Peabody Bay in 2020 (Fig. 4b), but not in 2021. What is the application in these instances?

*There is no particular application of these instances because they don't affect the main feature – the negative anomaly along the northern coast of Peabody Bay. The sentence was deleted. Thank you.*

Lines 233-234, "The difference in surface height anomalies between the southeastern and northwestern parts of Peabody Bay is supported by a similar difference in the observations of Tb" : In what sense do we interpret "is supported by"? Do you mean "is correlated with" or is there some physics behind the claim of support?

*Yes, "correlated with" seems to be a better way of saying what we meant. Changed.*

Line 234: Interpretation of AMSR brightness temperature. Please clarify whether the values depend on emissivity (ice type) as well as on surface temperature (of snow, of ice, or of somewhere between?).

*AMSR brightness temperature values depend on a combination of emissivity, surface temperature, and surface roughness (reflectance). To underline the difference between the actual surface temperatures and used brightness temperatures, we added the followed sentence in* *Line 131**:*

**"Note neither AMSR2 nor MODIS brightness temperatures are indicative of surface temperature alone, but measure the radiance of microwave radiation that is expressed in units of temperature (K) of an equivalent blackbody. Therefore, brightness temperatures are influenced by a combination of surface temperature, emissivity, and reflectance of the surface. In this study, we used $T_b$ to highlight a temperature contrast between adjacent regions, but didn't interpret it as absolute temperatures of the ice/snow surface."**

Line 235: Should "southwest" be changed to "southeast"?

*Yes. Thank you for finding this mistake. Changed to "southeast".*

Line 280, "we applied the 1-D thermodynamic ice growth model": Things like thermal coefficients, snow density, short and long-wave radiation, cloud cover do matter. Please provide a quick overview of the properties of this model, or an equivalent citation.

*Changed to "…we applied the 1-D thermodynamic ice growth model **with the same parameters as in Kirillov et al. (2015)."***

Lines 282-284. "We used 4 cm mo-1 snow accumulation rate to reach a modest snow thickness of 14 cm at the end of winter that is reasonably close to 19±2 cm obtained with AMSR2 data for Peabody Bay": As mentioned earlier, snow accumulation matching that in Peabody Bay may be unlikely. Ice off Cape Jackson may be blown clear of snow by frequent extreme winds in winter (see Samelson and Barbour, 2008: Fig. 6). It would be appropriate to mention this possibility.

*We don't think there is any issue here. We used different snow accumulations to simulate the sea ice growth away from the polynya at Cape Jackson (Fig.6b) and no-snow condition in the polynya (Fig.6c). Also see our answer to one of the previous comments on this.*

*For clarity, we changed this sentence as follows: "**Away of polynya,** we used 4 cm mo$^{-1}$ snow accumulation rate to reach a modest snow thickness of 14 cm at the end of winter that is reasonably close to 19±2 cm obtained with AMSR2 data for Peabody Bay (not shown)."*

Lines 288-289, "For having ice-free water in May, the heat flux should reach 70 W m-2 and be above 200 W m$^{-2}$ to let polynya form in early March": These estimates presume that there is no advection of newly formed ice downstream and beneath thicker pre-existing level ice and, I believe, that there is no insolation.

*We already addressed this concern while answering the major comment #4.*

*In respect to insolation, 1D ice growth model takes the incoming shortwave radiation into account (Kirillov et al., 2015). However, the estimated 200 W m$^{-2}$ were obtained for early March when the sun just starts rising above horizon at this latitude.*

Lines 440-441, "Although the northern branch is warmer and, being considerably faster, transports more heat compared to the southern branch …": Unfortunately, the northern branch is partially blocked from entering eastern Kane Basin by a shallow (70-90 m) spur extending more than 100 km southwest from Cape Jackson. The deepest crossing is relatively shallow, a 220-m sill at 79 40'N close to the Ellesmere shore. Moreover, because of geostrophic adjustment in the Arctic outflow, the warm mAW is at it deepest on the western side of the basin. To make a convincing argument about the temperature of the water that gets over this sill, more careful thought is needed. Where does the mechanical energy to lift water of the sill come from? I don't believe that a numerical model unvalidated in Nares Strait is a substitute for data needed to substantiate an hypothesis. Perhaps the authors could strengthen their case by exploring what the model has to reveal about the energetics of the phenomenon that they propose?

*We don't understand this reviewer's concern. We never mentioned that the northern branch penetrates into Peabody Bay (Fig.7) and somehow affects the observed anomalies there. In this sentence, we just underline the fact that even though the southern branch is colder than the northern one, it is warm enough to have a thermal impact on the sea ice. However, understanding why this concern appeared, we changed this paragraph considerably as follows:*

*"The only available source of ocean heat in Kane Basin during winter is associated with the relatively warm modified Atlantic Water penetrating into the basin from the Lincoln Sea (northern branch) and Baffin Bay (southern branch). **The inflow from the Lincoln Sea may also transport some heat within the upper thermocline layer consisting of Pacific Water. However, this heat may just reflect an upstream mixing with warm underlying warm mAW. For instance, based on the data collected in Kennedy Channel, Jones and Eert (2006) showed the fraction of Pacific and Atlantic Water in the upper part of thermocline at depth 90-100 m of about 70-80% and 20-30%, respectively.** Although the northern branch **of mAW** is warmer and, being considerably faster, transports more heat compared to the southern branch, **this water is thought to not to be present in Peabody Bay and can be mainly found in the western part of Nares Strait (Fig. 7a-b)**. According to FESOM-2 simulations, the mean temperature of **the southern branch of** mAW core in the central Peabody Bay is -0.15 °C or ~1.75 °C above freezing with a maximum observed at depth below 200 m. This heat may either be upwelled **over the mid-basin ridge closer** to the surface (leading to formation of sensible heat polynya at Cape Jackson) **and/**or transported upward **to the lower surface of sea ice (or to the ice-free polynya)** by vertical mixing…"*

Lines 456-457, "However, it is noteworthy that all these iceberg chains are located within the region with pronounced negative anomalies of ice surface heights in 2019 and 2020": Qualitatively, from the insets on Fig. 7, I estimate that the bergs cover only perhaps 10-20% of the sea surface; they could create point sources of turbulence kinetic energy through interactions with current, but are likely too sparse to form an area-wide source to explain the sea-surface anomalies which are manifest on the scale of the entire basin. Moreover, the warm seawater contacting icebergs at depth has plenty of opportunity to transfer its heat directly to the bergs, rather than hoarding to create havoc on the sea ice. The authors' hypothesis is plausible, but it needs appreciable quantitative physics to convert it into an explanation appropriate to uplift from 100-250 m depth.

*Even covering only 10-20% of total area within the mentioned chains, these icebergs represent a dense fence with no strait passages allowing flow to pass these chains undisturbed. One may*

*suppose that the flow jostles through the icebergs, change direction of streams and form a complex highly variable dynamics in this area.*

*However, following the reviewer's recommendation (the major comment #5), this part was completely removed.*

Lines 460-461, "However, the melting in this case is associated not with latent heat flux from water, but with dissolution controlled by solute transfer between water and ice-ocean interface (Woods, 1992)": I don't understand this point. I believe that a transfer of sensible heat to the iceberg is still required to free individual water molecules from the crystal lattice as dissolution proceeds. Please check whether you are citing Woods' work correctly.

*The reviewer is right - it was not written correctly. Of course, it's the heat flux that eventually melt the glacial ice. We just meant that the dissolution process and its rate are controlled by salt exchange. This part was completely removed, so there is no need of correction.*

Lines 496-497, "The stronger vertical mixing associated with the shear instability of the subsurface southward jet along the western coast … . This statement is speculative and not supported by observations. It is trivial to show with data in the Rabe papers that the gradient Richardson Number in the shear layer above the jet is about 2.2, almost 10x the threshold for shear instability. The most plausible sources of turbulence kinetic energy are in the wintertime mixed layer, namely shear in tidal currents at the base of rough sea ice and, less important with thick ice, brine-driven convection. Both can be estimated. I recommend that the authors do so.

*The Lines 611-615 (Section 4.2) were changed to address this concern. See our response to the major comment #7.*

*However, with all respect to the reviewer's opinion, we don't think that geostrophic velocities from Rabe et al. (2012) represent appropriate dataset for estimating Ri numbers. The good choice would be using the data of individual pings from 75kHz ADCP deployed at KS02 in 2003-2006. However, to our best knowledge, the highest frequency domain analysed and published based on those records was limited to the tidal variability (Munchow and Melling, 2008). But even if we had access to those records, we think such analysis is beyond the scope of the current research and it should be investigated in a separate paper.*

Lines 506-507, "generally support the idea of topographically controlled instabilities associated with the mean current and reversible tidal flow": I don't think it necessary to speculate about submerged topography generating instabilities. Headlands, by partially blocking along-shore currents, are notorious for strong tidal currents, and under-ice topography in Nares Strait is very rough.

*Our main objection to this reviewer's comment is that headlands can be found in many places at both sides of Kane Basin and Kennedy Channel, but it is only the western coast where negative anomalies of elevations (and, likely, ice thicknesses) are observed. We still persuade the idea that it is the southward subsurface flow amplified by semidiurnal tide, that generates the instabilities along its pathway. However, we gave credit for the mechanism related to the interaction of tidal flow and under-ice topography (see our response to the major comment #7)*

Line 511, "probably through the local upwelling": What is the basis for "probably". I don't believe that there are any soundings in Flagler Bay, so the existence of a sill is speculative.

> *It is a speculation based on a general understanding how the polynya in Flagler Bay can form. We could not figure out any other mechanism maintaining this polynya rather than a tidal upwelling, but said "probably" exactly because of the absence of information about water dynamics and bathymetry there.*

> *This part was deleted in the new version of the text.*

Lines 543-544, "Münchow (2011) reported a very similar warming in the southward branch of mAW of 0.23 °C/decade": Actually Münchow et al. (2011). This paper provides very weak evidence of long-term warming because the period of observation was only 6 years. The present authors have taken the liberty of extrapolating this to 10 years, and then referring to a supposed "as further warming of mAW progresses" – all this without having made a bullet-proof case for an influence of mAW on the sea ice of Nares Strait. It is one thing to have mAW affect glacial ice at the same depth, quite another to postulate an influence on sea ice at the surface hundreds of meters above. I suggest to the authors that the present evidence to make this projection is not statistically robust.

> *In respect to "influence of mAW on the sea ice", we agree that all our paper is built on the basis of indirect lines of evidence. Unfortunately, there is no possibility to provide a bullet-proof evidence without specialized direct measurements in the zones with large anomalies. However, it is the number of those lines that made us think that it is mAW that influences sea ice at both sides of Nares Strait, though through different mechanisms.*

> *We admit our mistake with using results of Münchow et al. (2011) who showed the positive trend of the mean cross-channel temperatures of 0.027 °C/year in 2007-2009 mooring records only. We changed this part and have reduced the emphasis on the future projections. However, we can't simply reject it because, if our suggestion about the impact of mAW on landfast ice is right, the warming of this water is important to mention.*

**SUPPLEMENTAL FIGURES**

**TideMarker 1**

[Figure]

**TideMarker 2**

[Figure]

**TideMarker3**

[Figure]

**TideMarker 4**

---

## Referee Report (RR1)

**The role of oceanic heat flux in reducing thermodynamic ice growth in Nares Strait and promoting earlier collapse of the ice bridge, version 2**

Sergei Kirillov, Igor Dmitrenko, David G. Babb, Jens K. Ehn, Nikolay Koldunov, Søren Rysgaard,

David Jensen and David G. Barber

**Overview**

I have restricted my present comments on the revision to the author's diligence in addressing my original comments. However, I admit that I did find the authors' responses to seemingly valid comments by the other reviewer unsatisfactory at times. For example:

> Other reviewer's comment: "There is currently no overlap between the ICESat-2 sea ice heights/AMSR temperatures (2019-2021) and the model results (2006-2010). It might be helpful to extend the AMSR temperatures back to 2006 to provide some comparison and context. It cannot necessarily be assumed that 2006-2010 have the same circulation conditions as 2019-2021".

> Authors' responses:

> 1. "We used the model of opportunity and were not able to choose a different simulated period."

>   *There is nothing wrong with saying this, but the future readers of the paper need to know this unfortunate reality too; therefore the gist of this response should be added to the text.*

> 2. "However, the main intention of the model was to demonstrate the circulation under the ice bridge. Since the main factors controlling water dynamics in this region (the along-strait sea-level gradient and the prevailing northern winds) don't vary a lot interannually, we reasonably suggest that the patterns shown in Fig.7-9 are generally valid and fairly represent (modeled) water dynamics and thermohaline state of Nares Strait in winter."

>   *I very much doubt that "the along-strait sea-level gradient and the prevailing northern winds do not vary a lot interannually", whatever "a lot" means. However, since this purported lack of variation is crucial to the authors' line of reasoning, the paper must provide citations or include data and discussion to persuade the reader that this is so. Without documentation, the statement is unacceptable.*

I originally provided three overall comments, supported by more detailed comments cross-referenced to specific places in the first draft. The overall comments were:

- The authors neglected the established role of tidal currents in maintaining polynyas in the fast ice of the North American High Arctic.

- The accuracy, precision and possible bias of ice/snow-surface elevation and temperature measurements were not determined, because no in situ data were brought into play. Without error analysis of uncertainty, readers' confidence in the results obtained from satellite remote sensing is eroded.

- The lack of contemporary or past oceanographic observations at the locations of interest is a serious shortcoming in a paper that strives to attribute polynya formation to oceanic heat flux. The FESOM global ice-ocean model that has been harnessed in an effort to fill this gap seems not to have been evaluated in this tiny – from a global perspective – area. Readers' skepticism about the study' results is therefore likely to be high.

Here I am taking the same approach in assessing the authors' responses. I provide an overall assessment of the revision in the next section. Subsequently, I provide my original comment (black font) followed by my assessment of the authors' response in blue font.

**Assessment overall**

*Concerning "The authors neglected the established role of tidal currents in maintaining polynyas in the fast ice of the North American High Arctic".*

The authors' begin a semi-quantitative (and incomplete) speculation to argue that tidal currents can't work to create a polynya at Cape Jackson, but later fall back on a more plausible hypothesis: "we have to admit that our suggestion that upwelling brings warm deep water from Peabody Bay directly to the surface may be too challenging. In combination with vertical mixing, it may be sufficient to upwell this water just closer to the surface – to the bottom layer over the "ridge" dividing Peabody Bay and the central channel of the strait".

Correct! This is the hypothesis developed in the citations to which I directed the authors in my first review: Polynyas beneath fast ice in the Arctic need two things: 1) "warm" (probably at least 0.05C" above freezing) water locally at depth, almost certainly of higher-than-surface density; 2) a strong local source of turbulence kinetic energy. In Nares Strait, baroclinic adjustment of the southward mean flow causes shoaling of warm water into shallower areas on the Greenland side, providing the shallow warm-water source beneath the polynyas. The strong local source of turbulence kinetic energy is almost certainly shear in the benthic and under-ice boundary layers of the strong tidal current. Tidal current, contrary to the authors' contention but well-known to mariners, is commonly accelerated around headlands like Cape Jackson; a narrow strait is not required. The generated turbulence erodes the stratification in density between the seabed and the surface and diffuses oceanic sensible heat, if present at depth, to the under-surface of the ice. An internal tide generated by the tidal flow across bottom topography, often serves to enhance to warm-water uplift in such areas. The strong tidal current additionally plays a role in sweeping newly formed frazil and nilas ice beneath or onto adjacent fast ice, except perhaps during neap tides and very cold conditions.

So, the authors are on the right track, but the suggested modest modification to line 536, namely "This heat may either be upwelled over the mid-basin ridge closer to the surface and/or transported upward to the lower surface of sea ice (or to the ice-free polynya) by vertical mixing" remains inadequate. The authors needs to lay out the requirements: warm water at depth, upwelling and a source of turbulence kinetic energy. Second, a paragraph is needed to properly enlighten the reader about the ocean and ice physics involved and third, the authors need to acknowledge earlier work and third,.

*Concerning "The accuracy, precision and possible bias of ice/snow-surface elevation and temperature measurements were not determined."*

See my critical responses below to comment 2, 3, and comments on lines 38, 46-47, 132 and 533-534. Work is still required

*Concerning "The lack of contemporary or past oceanographic observations at the locations of interest is a serious shortcoming in a paper that strives to attribute polynya formation to oceanic heat flux."*

This comment remains difficult to redress, but it should be straightforward to acknowledge and document shortfalls of in situ data. I have continued in the detailed comment below to urge the authors to keep readers aware of all the ifs and buts associated with story that they are telling.

*Overall the manuscript remains too speculative in its "explanations". It lacks brief digressions to present simple physic or statistics that could provide some reassurance of validity to speculative interpretations offered. In detailed comment below, I have continued to remind the authors of their responsibility in scientific publication to anticipate the queries of skeptical readers, and to provide enough scientific underpinning to bring this readers along with them set these readers at ease.*

**Major revision required: This revision is an improvement, but remains in need of further revision.**

**Comments (major)**

1. The authors have chosen to refer to the fast ice that covers the full 500-km of Nares Strait during many winters as the "ice bridge": The terminology is confusing because a long strip of fast ice does not resemble a bridge. I believe that most readers will consider the bridge to be the arch that forms the boundary between fast and mobile ice, most often in southern Kane Basin. As in masonry, the arch is strongest geometry for a load bearing structure because it is everywhere under compression, thereby exploiting the stress-state where sea ice is strongest. I recommend that the authors devise a different term to refer to locations within the fast ice "above" the arch. For example at Cape Jackson, "more than 200 km north-east of the bridge".

   I accept the authors' sentence clarifying their use of the terms, and their choice to retain "ice bridge". I am disappointed they have chosen not to clean up the terminology. "Arch" and "bridge" sound like much the same thing to me, but are used for different purposes in the text.

2. Line 148, "The model was driven by the atmospheric reanalysis fields from JRA55-do": This was probably not a good choice. Samelson and Barbour (2008) concluded that a grid 10x finer than that of JRA55 was required to correctly represent weather conditions in Nares Strait. I recommend adding a discussion the capability JRA55 to represent the mesoscale meteorology of Nares Strait, a channel much narrower than 55 km in width for much of its length, bordered by high terrain and characterized by a strongly stable atmospheric boundary-layer – the Arctic inversion – during the freezing season. This could be perhaps achieved via comparison of simulations by the ERA55 and Polar MM5 models.

   I am pleased that the authors' have added a sentence discussing an intercomparison of ocean current modeled independently using wind from 55-km and 6-km atmospheric grids. However because the graphics they provided in their response will not appear in the paper, I take issue with their claim that "… the modelled current velocities in Kennedy Channel and in Smith Sound fairly coincide (not shown) with velocities obtained by Shroyer et al. (2015)". I acknowledge that the models produce flow cross-sections that are qualitatively similar. However, quantitatively they are appreciably different; Schroyer's sections show the speed of the undercurrent reaching about 45 and 65 cm/s in the two sections shown, whereas those in the FESOM sections reach only 25 and 40 cm/s, just 55-60% of those with high-resolution forcing.

   Because the models do not honestly "fairly coincide", I recommend that the quoted clause be revised to read "… the modelled cross-sections (not shown) of average November-June ocean current through Kennedy Channel and through Smith Sound driven by winds from FESOM-2 and from MM5 (9x higher resolution: Shroyer et al., 2015) are qualitatively similar but those derived using MM5 forcing yield currents that are 60-80% larger".

3. Line 166, "The 6-hourly records of 2 m air temperature, wind speed and humidity in Kane Basin were taken from the ERA-5 global reanalysis database": The 31-km grid of ERA5, comparable to the width of Nares Strait, does not come close to resolving the channel and very its steep surrounding terrain. I suspect that much of the strait's "sea surface" resolved on a 31-km scale will actually be above sea level and therefore "terrestrial". It is very difficult to accept that ERA5's 2-m air temperature values hold much credence for simulating ice growth in the real world. Since sea ice loses the upward flux of latent heat from wintertime freezing by long-wave radiation from its top surface, would you not be better using satellite-derived surface radiation temperature to model freezing? Please strive to persuade readers that my viewpoint is invalid.

   I am relieved that the authors acknowledge the serious limitations of the ERA5 meteorological analyses in Nares Strait where variations in substrate (water, sea ice, land, ice cap) and elevation (sea level to 1000m) occur on a scale of a few km. I am pleased that they have added a paragraph that explains their goal in using the ERA5 forcing. They are also correct in stating that local-scale surface air temperature takes second seat to local snow depth in driving variations in ice growth rate. Unfortunately, as mentioned in my earlier review, the true sea surface off Cape Jackson, which would take the second seat, will likely lie hundreds of metres below where ERA5 "thinks" it is, because sea-level features as narrow as Nares Strait (40 km) probably do not exist within the ERA5 grid. Hundreds of metres in the polar atmospheric boundary layer span tens of degrees in temperature, which is non-trivial for ice growth.

Moreover, the authors state in the response (but not in the added paragraph) that "a spatial variation of any meteorological parameter over such a short distance is negligibly small. Therefore, any local anomalies of ice thicknesses here are thought to be controlled by spatial variations of ocean heat flux rather than varying meteorological parameters". This statement may be true on the 1000-km scale of synoptic weather forecasting, but it is not true at the surface within a few tens of km of a change in substrate, such as the edge of a polynya. The transition from fast ice with a surface temperature of perhaps -25C to open water or thin ice is likely at least 20C. Moreover, the atmospheric boundary layer, warmed and moistened as it moves over the polynya, is quickly covered by ice fog and further downwind stratus cloud (the cause of "water sky" over polynyas). Cloud droplets have high emissivity and their presence can reduce or perhaps reverse the upward heat loss by long-wave radiation that drives ice growth. Here huge differences occur on the mesoscale in this environment.

The authors have also responded with a couple of plots (to me, but not in the revised manuscript) showing probabilities of wintertime air temperatures and wind speeds measured with the Automatic Weather on Hans Island (red line) and from the nearest node of gridded ERA5 product during the same period "… to demonstrate that ERA5 parameters are realistic". Unfortunately, for this comparison to have any value, we need to be absolutely sure that the Hans Island data were NOT USED in the ERA5. If they were, then the re-analysis result would be strongly biased to the data there, and quite possibly nowhere near as good elsewhere. Also, the conditions compared are for a rocky site high up (170 m) in the Polar Inversion. Its relevance to ERA5's capability to predict conditions at marine locations at the bottom of this inversion is questionable.

This is all to say that there is clear potential for atmospheric forcing to vary appreciably on the scale of the polynya because the polynya is capable of creating its own weather.

This is also to say that it is not good science to make a statement such as the one that I quoted two paragraphs up without being very careful about the distinctions between physics of the atmosphere on a synoptic scale above the planetary boundary layers and that on the mesoscale within the planetary boundary layer. Sweeping generalization can lead one astray.

I don't know what to recommend here to move your paper along, since once again you/we are stymied by the absence of information that you really need, specific to your locations of interest. Rather than torpedoing the paper, I prefer to recommend that you add straightforward discussion to acknowledge and describe the complexity of the analysis that you are undertaking, and the fact that many of your assumptions are ad hoc rather than readily justifiable. Such an approach will enlarge the document, and the result will not be definitive, but a full declaration will be of greater value to future scientific studies than a pretense that everything is just fine.

4. Line 415, "It was found that the ocean heat flux at Cape Jackson needed to exceed 200 W m$^{-2}$ to open the polynya as early as in March": The message intended here is unclear from the present text. Revision is required.

I don't believe that it is plausible for a polynya to suddenly melt itself into existence – oceanic heat fluxes just aren't large enough. The more likely role of oceanic heat flux is keeping the ice relatively thin (and relatively weak), so that more powerful mechanical (fracturing, rafting, flooding and downstream advection, etc.) and thermodynamic (radiation) processes can do their work. Principal among the former are the stresses exerted by wind, wind-waves and tidal current on already thin ice. Once these open a polynya, new ice (mainly frazil and nilas) created by high rates of heat loss (> 200 W m$^{-2}$) from the surface will continue to be removed by current, while insolation and downwelling short-wave radiation may deliver appreciable heat directly to seawater. This "tag-team" approach to creating a polynya in fast ice, involving both dynamics and thermodynamics, has been discussed by Topham et al., (1983. JGR 88).

Actually lifting warm water 100 m or more to the surface to open a sensible-heat polynya requires a large input of kinetic energy to the ocean, which cannot occur with a continuous cover of fast ice. It is only possible when strong winds act on mobile ice or open water. There is a paradox because necessity for strong

winds  is the same requirement for the opening of latent heat polynyas; the distinction between the purported "two types" of polynyas is not clear (see discussion in Melling et al, 2001). Moreover, because the wind must be integrated over a large expanse of mobile pack ice to accumulate enough kinetic energy to drive upwelling, the formation of small polynyas in fast ice, such as that off Cape Jackson, via this mechanism is unlikely.

The authors appear to have misunderstood my response to the authors' original statement "It was found that the ocean heat flux at Cape Jackson needed to exceed 200 W m-2 to open the polynya as early as in March". However, this statement does clearly say, perhaps unintentionally, that the OCEAN HEAT FLUX must exceed 200 W m-2 to open the polynya as early as in March. They have come back in their response with an image of the polynya, showing thin ice and perhaps open areas and a statement that "If water surface is ice-free, 200 W/m2 does not seem to be extraordinarily high". I agree with this statement, with the reminder that if the surface is ice-free, that heat flux is derived from the latent heat of freezing for seawater and not from sensible heat in the ocean. The authors appear to have confounded these two fluxes.

Continuing their response, the authors write "The mechanical processes maintaining water at Cape Jackson ice-free suggested by reviewer can work to a certain extent only. Occurring within very small, landfast-ice constrained area, none of these processes may prevent the polynya to eventually become covered with ice unless the ocean heat flux is strong enough to melt it". The authors are correct in writing "none of these processes may prevent the polynya to eventually become covered with ice" but a cover of thin and young ice does not terminate the existence of a polynya. Polynyas are not necessarily open-water areas; they are areas with ice cover "appreciably thinner" than adjacent areas of ice on all sides. Many polynyas exist with a thin ice cover through the winter; at times when covered by a new fall of snow, they may be difficult to distinguish from surrounding ice, but they are still there. They only become ice free in the darkness of winter via the action of wind and current in fracturing the thin ice cover and carrying it away – a so called "latent heat polynya". As winter wanes, sometimes in April but more reliably in May and June, insolation will strong enough, at hundreds of W/m^2, to melt the ice.

The authors last point is "If such process had taken place at Cape Jackson, the continuous accumulation of large amounts of frazil/new ice from polynya below landfast ice sheet would have resulted in much thicker surrounding ice close to edge (at least in certain directions)". The advection of frazil is well known in both oceans and rivers; under level ice, the crystals remain suspended in the turbulent surface waters and travel far, spreading their influence thinly over an area much larger than the polynya. Remember that tidal currents rotate and typically flow in every direction in the course of 12 hours. Under sea ice, frazil may settle against ridge keels, or within their porous structure, with a consequent increase in keel consolidation that is undetectable  from space. Flooding, downfolding and submergence of sheets of new and young ice on the downwind, down-wave or down-current sides of polynyas in fast ice is well documented. It the process responsible Joseph-René Bellot's drowning in 1853 and has provided a close call for many others. Sheets of young ice have low buoyancy and can travel large distances in turbulent flows before the water between them and the overlying ice sheets is squeezed out.

The authors' arbitrary dismissal of earlier research on and knowledge of High Arctic polynyas in fast ice is not acceptable. Their continued insistence that the ocean has the capability of providing an unprecedented 200 W/m2 flux of SENSIBLE heat by some mysterious physical mechanism to an open sea surface continuously throughout the winter is highly unlikely. If they wish to stick with this concept, a great deal of careful work and explanation will be required to make a convincing case, if such can actually be done..

The revised text in lines 358 and 497 is therefore unacceptable. That in line 230, concerning the appearance of open water in March off Cape Jackson is okay if the phrase "as the daily downwelling shortwave flux increases with the coming of spring" is appended.

5. Lines 447-473: The speculation about iceberg melting provides an intriguing diversion, but it doesn't add much to the concepts central to this paper. I suggest that it be removed from the paper.

6. Section 4.2, "The formation of ice thickness anomalies along the western coast of Nares Strait": This is indeed an interesting feature. It did occur to me that is might possibly be an artifact in the southern half of the strait to the authors' referencing of elevation to the mean value measured along the full length of the strait. Has the possibility been investigated that the higher sea level in the north that drives the current might explain this anomaly?

   Alternatively, could it be a manifestation of the undercurrent that hugs the Ellesmere coast when fast ice covers the strait? It is interesting that the elevation anomaly has roughly the same 10-km width as the undercurrent mapped by Rabe et al (2012; Fig. 4b). A connection to the dynamic relief reflecting geostrophy in this flow is implausible; it is only about 10% of the measured 20-cm drop in ice-plus-snow elevation adjacent to the coast. The inverse barometer effect, which might also contribute to lowered sea level in response to higher SLP at the western shore (not resolved at this scale by ERA5) is also too small: Sea-level pressure is only higher by 2-4 mb on the Ellesmere side of the strait (Samelson and Barbour, 2008: Fig. 8). Nonetheless, I recommend noted both possibilities as having been explored.

7. Line 484, "We suggest that the observed negative anomalies are attributable to the heat upwelled from the underlying mAW": I believe that this suggestion has merit, but that the details are incorrect.

   The flow structure beneath fast ice in Nares Strait depicted by Rabe et al (2012; Fig. 4b), displays a jet of roughly 10-km width against the Ellesmere shore, centered at about 80-100-m depth. The baroclinic adjustment of the ocean to this jet (not shown) involves downwelling below the core of the flow and upwelling above. This leaves no mechanism to raise mAW through the core of the undercurrent to the surface on this side of the strait. Indeed, the cross-strait circulation that compensates for downwelling of mAW on the western side is upwelling on the other side, near Greenland!

   However, upwelling does occur above the core of the jet. This would bring Pacific Winter Water as much as 0.2C warmer than the surface-freezing temperature (see Melling et al. 1984 Cont. Shelf Res 3) to the base of the surface mixed layer (see Melling et al. 1984 Cont. Shelf Res 3). This sensible heat in this Pacific Water could provide a heat flux to the underside of the sea ice via entrainment into the turbulent surface mixed layer. The needed turbulence kinetic energy could originate in part from brine-driven convection (ice growth) in the mixed layer and in part from shear between rough immobile ice and the rapid tidal flow. Melling et al. (2015) estimated an oceanic heat flux to the base of ice as 15 W/m² under similar circumstances in Penny Strait, which would be sufficient to melt about 0.5 cm/d from 3-m ice (with 10-cm snow) at -25C. It should be noted that the submerged jet, and the upwelling above it near the western shore, do not exist when the ice is moving, so that the oceanic flux would be much reduced in years without a fast-ice cover.

   The authors response to my suggestion is gratifying. Thank you.

   As I have mentioned, the baroclinic adjustment associated with the undercurrent core centred near 100 m only moves water above this depth to the surface; baroclinic adjustment below this depth drives water downward. Münchow et al. (2007) might be a better citation for the fractional presence of Pacific Water at different depths on sections along Nares Strait than Jones (2003) [*Münchow, A., Falkner, K. K., & Melling, H. (2007). Spatial continuity of measured seawater and tracer fluxes through Nares Strait, a dynamically wide channel bordering the Canadian Archipelago. Journal of Marine Research, 65(6), 759-788. figs. 6, 88, 11*]. You can see in their figures how the 50% Pacific Water contour corresponds roughly to the core of the undercurrent, with the remainder being predominately what you call mAW below this depth and meteoric water above it. But you also see that the mAW designation is a bit clumsy, since to be consistent you should be calling the PW, mPW – no purity so far from the sources. You might be better off abandoning use of mAW (I'm am not sure if this is an accepted designation anyway) and talking about the water above the core being predominately PW and that below as being predominately AW. Incidentally the upper PW, from the Alaskan Coastal Current in summer, near 80m depth in western Nares Strait, is plenty warm

enough to melt ice. It doesn't need help from the Atlantic and is much better positioned in the water column to do so.

Adjustments to the text based on my assessment are recommended.

8. Lines 504-505, "Transformation of these currents over steep topography generates baroclinic semidiurnal tidal wave that may considerably enhance vertical mixing through benthic stresses and shear instabilities": I suggest that the steep cliffs on the western shore, indicative of deep water close to shore, make turbulence and internal waves generated in the benthic boundary layer irrelevant to the ice far above. However, I believe there is a good possibility to generate strong turbulence, mixing and entrainment through the action of the tidal flow (Pite et al., 1995. JPO 25) on the very rough under-ice topography of Nares Strait (Ryan & Munchow, 2017).

I suggest that the authors give some thought to this alternate, and I believe more plausible, explanation for the source of ocean sensible heat.

The modifications in line 607 have created a paragraph that is a little unclear, since it deals with two features in the ice cover that appear traceable to different physical processes: 1) the continuous band of thinner ice within 10 km of shore, most plausibly attributable to warm water upwelling above the upwelling; 2) The early appearance of water near headlands, most plausibly attributable to upward diffusion of sensible heat driven by turbulent instability of tidal current accelerated around headlines. I suggest that you could be more easily understood by having a separate paragraph for each issue.

*Query by the authors: Not for the paper, but we want to point at one thing related to the idea of sub-ice tidal mixing that makes us confused. The semidiurnal tide forms a standing wave pattern in Kane Basin with the lowest tidal velocities in its central part (Davis et al., 2019). If so and if the negative anomalies along the western coast are tidally driven, they should have become smaller around Cape Frazer. However, we don't clearly see it in Fig. 4b-c.*

Perhaps you are referring to Davis et al. (2018) fig. 4? This actually depicts the amplitude of the semi-diurnally varying flux of tidal energy, not the speed of tidal current. The flux is $P = \rho \cdot g \cdot h \cdot $, where $u$ is the depth-averaged tidal current and η is the tidal elevation; $P$ and $u$ are vectors. Both these properties vary semi-diurnally and can have different phase. Fig. 4 illustrates that differences in phase result in a northward flux of tidal energy on the Greenland side of Kane Basin and a southward flux on the Ellesmere side. Note also that the tidal energy flux increases with water depth if $u$ remains constant. I think that the strength of tidal currents off headlands – a local effect – depends very much on the character of the obstruction that the headland and adjacent submerged terrain pose to tidal flow. It is also possible that at Cape Frazer – a steep high cliff – drainage of snow melt-water from the heights onto the ice may help to expand the polynya in June.

9. Line 523, "weakens the cohesion of landfast ice against the shoreline in Kane Basin": The tidal cycles in sea level ensure that the ice sheet is always fractured at the coast, not bonded to it. However, the word cohesion implies that the authors consider that bonding of ice to the shoreline is important. This line of thought runs contrary to decades-old discussions of fast ice in deep water, where it is the formation of ice arches across channels which stops the movement of ice behind them, not shear strength at the shoreline. The upwardly convex shape of a masonry arch is the key feature that allows it to resist downward loading; the shape ensures that all the stone in the arch is under compression, the stress state in which it is strongest. Indeed the stress is highest within the wedge-shaded stones of the arch and much less above them. Pack ice also is strongest in compression and much weaker in shear. Although there are likely several arch-shaped load-bearing features distributed in the fast ice along the length of Nares Strait during any winter, much of the fast ice cover will be in a low state of stress; cohesion at the shoreline is probably unnecessary for fast-ice stability, although its confinement by irregularly shaped shorelines may constrain it from moving locally. Conversely, weakening of that confinement by melting at the coast may allow it to shift around in response to wind and tide. It is quite common to see the ice in Kennedy Channel become mobile between arches at its northern and southern ends long before the collapse of the arch in Smith Sound

allows the ice in Kane basin to do the same. The same phenomenon is seen annually in Prince Regent Inlet. I don't think that the authors' argument for up-channel polynyas hastening the break-up of fast ice further down-channel has much merit, as presently written. It is possible, of course, that phenomena may be correlated in time because of the influence of a third circumstance not identified.

The authors appear to agree with my comment that cohesion of ice at the coast is not important to the longevity of the ice bridge. However, they continue to use the word 'cohesion' in the revised text at line 628, which I think is misleading. I suggest that they substitute something along the lines of "… we suggest that thinner ice and open leads at the shoreline of Kane Basin, linked to locally enhanced oceanic heat flux, provides weakness that allows patches of fast ice further from the coast to fracture and move in response to wind and current".

They subsequently speculate that collisions of these freed patches with remaining parts of the bridge "may" contribute to an earlier collapse of the bridge. This speculation is okay as an hypothesis, but the comment is of little value to the reader if it is just left hanging there. The reader has no idea whether or not this suggestion is worthy. How about adding a little "back of an envelope" physics to see whether the speculation has merit?

I decided to calculate the shock-loading of the fast-ice arch by the impact of a free-drifting floe for comparison with the sustained loading of an equal segment of the arch by the accumulated drag forces of wind and current on upstream fast ice. I assumed a 1-km width of contact by a square floe, 2.5 m thick, moving at 0.3 m/s and brought to a stop after ridging 30 m of ice along the zone of contact; with assumed constant deceleration, the force of impact was 3.45 million newtons, sustained over 200s. For the comparison, I assumed that wind and current stresses were accumulated over a fast-ice strip of the same width stretching 100 km up-channel from the arch, subject to the same 0.3 m/s current and to 15 m/s wind; the calculated loading was 37.5 million newtons on just this 1 km segment of the arch, sustained for days to months. In engineering terms, the arch has a safety factor of 10.9 for resilience against impacts of the specified type. The colliding floe would have to stretch more than 10.9 km up-drift for the safety factor to drop below 1, but this vulnerability would last for only a brief 200-s interval.

In my mind, this calculation suggests that the authors' idea that collisions with freed patches may contribute to an earlier collapse of the bridge is a non-starter … but perhaps the authors can devise a different scenario and simple physical model wherein the odds are not heavily stacked against their idea. If they cannot, I suggest that it be dropped from the text.

10. Line 528-529, "This break-up appeared to release internal stresses in the ice bridge and led to concomitant ice cover break-ups in the main channel": This statement appears to rely upon a knowledge of the dynamical state of the ice cover. Nothing is known about stresses. In reality all you have access to is evidence of deformation (in the form of cracks) and of motion. Also see comment #9.

The authors have voiced agreement with my comment. They have changed the sentence to read "This break-up appeared to initiate the further fracturing of ice cover in the main channel". Since break-up and fracturing of the ice cover are essentially the same thing, this sentence seems circular and of little import. I may have missed the point, but either way, the sentence needs to be re-worked for improved clarity.

11. Line 531: This paragraph gives the impression that the polynya has played a role in the breakup, but really all that you demonstrate is that the breakup was correlated with expansion of the polynya. Perhaps the expansion of the polynya is just one event in the process. A more robust discussion, with a more useful take-away, would review the other factors in play, as listed in Line 520. If such completeness is thought to be beyond the scope of the paper, perhaps it should be covered in a separate paper. See comment #9.

The revised manuscript has new text in line 646: "although our hypothesis that polynyas facilitate ice bridge break-up in Nares Strait is speculative, we would like to emphasize the observations that the first movements of the immobilized ice cover occurred in areas with negative ice thickness anomalies during winter and where polynyas are observed."

Hypotheses are typically the second stage of the scientific method, following collection and examination of data relevant to the phenomenon to be studied. They remain as hypotheses until a mechanism of interaction has been identified and the direction of causation established.

Since both these subsequent steps remain undone, the authors' science is incomplete, meaning in strict terms that this paper is premature. However, if what appears in the conclusions is a limited list of plausible hypotheses to guide future research, with mechanisms tentatively identified) and with the list clearly flagged as such conjecture, with nothing really "proven", this could be sufficient for acceptance of this paper.

12. Line 539-540, "The only oceanic heat source available to maintain such a polynya through winter is the modified Atlantic Water": This is not true. It may be the warmest source, but it is not the one closest to the ice. My comment on Line 484 raises the possibility that the less conspicuous warmth of the Pacific Water might be more influential than you give credit for. I recommend that you re-think the paper with this in mind.

The authors response to this comment appears entangled with the definitions of the water masses themselves; see my concluding comments under item (7) above. All water masses are only pure at their sources, so every water mass is a modified something. Moreover, the relationship is reciprocal, if PW mixes with AW, is the result mAW or mPW? The distinction between AW and mAW seems irrelevant. Admittedly, some sensible heat from the AW has reached the overlain Pacific Winter Water by the time the mixture enters Nares Strait, but less dense varieties of PW higher in the water column, namely Bering Sea Summer Water and Alaskan Coastal Current (summer) Water are less likely to have been influenced from below and more likely to have been influenced from above, by summertime Meteoric and ice-melt waters and wintertime freezing-associated brines. As stated under (7), I consider it less confusing to label a water mass PW if it is more than 50% virgin Pacific, and similarly AW if more than 50% virgin Atlantic. Then the hair-splitting about which water mass is most influential can be abandoned.

Definitions aside, I still maintain that if AW is to be lifted somehow to the base of the ice, the overlying PW will by necessity have to have gotten there first, and with a smaller expenditure of energy. So I judge the hypothesis that sensible heat reaches the ice from PW and perhaps also from AW to be more plausible than the converse, namely that sensible heat reaches the ice from AW and perhaps also from PW. Only if the authors could come up with an energetically viable mechanism to explain the latter version would I be more believing of it.

I recommend that the revised text provided by the authors and quoted under this item be re-worked either in terms of a simpler stack of water masses (i.e., ASW, PW, AW) which perforce are blended across their interfaces, or instead referenced to a partition of source waters in terms of T & S thresholds. Either approach would make this discussion much less enigmatic.

With reference to "This flow occupies the entire water column and consists of 3 distinctive layers; i) cold brackish polar mixed water within the upper 50-60 m, ii) the upper thermocline coinciding with halocline that is observed at 70-110 m …" in the revised version of the text, I caution the authors to look carefully at the Rabe papers which show appreciable differences between T & S profiles in winter and those in summer, which are quoted from the Jones papers. Since the present paper concerns winter, the former reference may be a more relevant source.

**Comments (minor)**

Line 27: Shokr et al. (2020) is a weak reference for the role of the along-channel sea-level in driving flow down Nares Strait. Münchow & Melling, J Mar Res 66, doi.org/10.1357/002224008788064612 would be much better.

Good

Line 30-31, "The ice bridge also helps prevent the loss of the thick, old ice from the Last Ice Area": The paper cited (Moore et al, 2019) is not helpful in substantiating this statement; it has very little to say about Nares

Strait. To my knowledge, there has not yet been a study demonstrating that ice loss from the LIA, as distinct from ice export through Nares Strait, is reduced during years when an ice arch forms there. Nares Strait is only one of four pathways (and the narrowest) via which ice leaves the LIA – the others are to the NE via Fram Strait , to the SE through the QEI and to the SW to the Beaufort Sea. It is quite plausible that a blocked Nares Strait simply creates a diversion of ice to one of the other pathways, most likely Fram Strait. You need a citation that demonstrates convincingly that this is not so.

Good

Line 38, "… peak in the fraction of sea ice with a draft between 2.6-2.8 m": It is important to note here, as was in the cited paper, that this range in draft was computed on the assumption of no snow cover, which may bias values appreciably high. Also, a referenced estimate of the empirical accuracy in draft estimates from CryoSat freeboard should be included here.

In their response, the authors do not draw a distinction between random errors, which do shrink as the number of values averaged increases, and bias errors that do not. Since all values in a 1.5x1.5 km square share the same bias, traceable to errors in estimating propagation delays from space, in the geoid and in the (unseen under fast ice) sea level, the average is biased by the same amount, whatever it is. The authors still need to provide information on the magnitude and character of bias in CryoSat as part of the necessary error analysis.

Lines 46-47, "That bridgeless years only occurred during last 15 years underscore a general shortening of bridge existence period and point to changes …": It would be appropriate to clarify that this statement refers to the absence of an ice bridge at Smith Sound (think) and not to the much smaller number of years when there was no bridge anywhere between Baffin Bay and the Arctic Ocean.

In this clarified context, it should then be noted that there was one winter (1995) in the 1990s with no arch at Smith Sound – in 1995 the arch formed at Hans Island – and one (1993) essentially like 2007 with no arch anywhere; "essentially" because an arch in Smith Sound that year lasted only 10 days (Vincent 2019). With a 30-year perspective, the record looks less amenable to interpretation via trend: there is a cluster of 2 of 3 years with no arch at Smith Sound in the mid-1990s, then an 11-y period with annual arches, then a cluster of 3 of 4 years with no arch in the 2$^{nd}$ half of the 2000s, then a 6-y period with annual arches, then a cluster of 2 of 3 years with no arch in the second half of the 2010s. Disregarding clustering and estimating the probability of no bridge in any year from the data as 7/31, one uses the Poisson Distribution to estimate the likelihoods of the observed gaps between no-bridge winter – that is having 2 no-bridge years in 2 years, 2 in 3 y, 2 in 7 y and 2 in 12 y. These are 6.4%, 11.7%, 25.7%, 24.4%. The low values for the small gaps suggest there is clustering in play; the relatively high values for the large gaps suggest that such wide gaps are not unexpected, so that bridging despite weak clustering, looks like a Poisson process. On these grounds I suggest a re-examination the statistical confidence of the statement in lines 46-47, which is based on such a short time series.

I took pains in my first review to demonstrate that the small number of bridgeless events is insufficient to draw with high confidence a distinction in their occurrence between the first two and the last two decades.

I stress that it remains important to provide limits of confidence on 2 as the mean for the first two decades and on 6 as the mean for the last two, and to calculate the confidence with which the data allows these two numbers to be considered different. Without error analysis, this is just handwaving.

Lines 48-49: I think that the date-based approach of Vincent (2019) is probably a more robust approach to a short 30-year time series than is the counting of the rare occurrences without arches, which the authors have used here.

Okay.

Line 54-56, "it is the sensible heat polynyas … that are more common in the Canadian Arctic (Hannah et al., 2009)": The authors appear to mis-quote Hannah et al. (2009), who state "… are widely distributed across the Canadian Arctic Archipelago"; Hannah at al. are clear that these sensible heat polynyas are features within fast ice in this region. Their map (Fig. 1) shows that the latent heat flaw-leads and polynyas that form along the

perimeter  of the fast ice are actually more widespread across the Canadian Arctic waters and occupy much more area.

*Okay, except that Hannah's paper actually did not consider heat sources. It simply correlated existence of polynyas in fast ice with h/u^3. Areas with small values of this parameter typically have strong current, maintaining the likelihood that both latent and sensible heat processes are in play. The Topham reference documented just this situation at the Dundas Island polymya. Please adjust your text accordingly.*

Line 67: Refer the reader to Fig. 1 for the mapped location of Cape Jackson.

*Good*

Line 67 et seq., "… at Cape Jackson in the central part of the bridge": The terminology is confusing. I believe that most readers will consider the bridge to be the arch that forms the boundary between fast ice and mobile ice in southern Kane Basin. It follows that the central part of the bridge is the "top" of the arch, halfway across the strait between Greenland and Ellesmere. However in this sentence, the authors are referring to a location in fast ice more than 200 km "above" the arch. I recommend that the authors devise a different term to refer to locations within the fast ice "above" the arch. Simplest in this example would be "… at Cape Jackson, more than 200 km north-east of the bridge".

*Okay*

Line 93, "… maintaining water at Cape Jackson ice-free during winter": The reality is ""… maintaining water at Cape Jackson ice-free at times during winter".

*Good*

Line 94 et seq., "under the bridge": See comment re line 67. I recommend using the phrase "beneath the fast ice" for the reason already given.

*Okay*

Line 100: Line 67: Refer the reader to Fig. 1 for the mapped location of Peabody Bay.

*Good*

Line 130 et seq., "crossing the bridge": See comment re line 67.

*Okay*

Line 132, "Although ATL07 data are manifested to be adjusted for geoidal/tidal variations and inverted barometer effects": The correction for the inverted barometer effect is probably only accurate in wide deep ocean basins where the long ocean wave which is the ocean's response to changing atmospheric pressure can move as fast as, and in the same direction as, the SLP anomalies moving at 20-25 m/s. I suspect that the correction will not work well in a long (550 km) narrow (35 km) strait. I urge the authors to find and reference research that provides a discussion of the accuracy of the inverted barometer correction in confined coastal waters.

*Notwithstanding the authors' hopeful "don't worry" comment in their response, this issue remains unresolved in my mind without reassuring observations. End-to-end SLP difference of 25 mb are quite common in Nares Strait (Samelson papers); who knows how much of the 25 cm of sea-level adjustment is actually distributed along the strait?*

*I therefore urge the authors to at least acknowledge the possibility that bias from inaccurate inverse-barometer correction could be a source of error, unless they can find data do better.*

Line 133-134, "… may still contain unknown uncertainties related to the regional synoptic variability of sea level associated with wind forcing and/or with ocean dynamics": With respect to the atmosphere, I recommend replacing "wind forcing" with "strong wind, air-pressure and ocean dynamical effects on the mesoscale (10-30 km)", referencing Samelson and Barbour (2010).

*Okay*

With respect to the ocean, Münchow & Melling (J Mar Res 66) provide estimates of the anomalies of sea-level height relative to the mean. These have amplitudes as large as 10 cm along-channel and a few cm/s acrosschannel. These along-channel value is large enough to contribute appreciable fortuitous NE-SW varying anomalies in thickness that are computed relative to an along-track (approximately along-channel) mean. This source of error requires discussion.

Okay

Line 139-140, ">0.3 m mean snow depth in Kane Basin. However, as 140 we will show later, this height seems to be overestimated". Reference to Samelson and Barbour (2010) is again appropriate, since the extremely strong winds common in Kennedy Channel and the vicinity of Cape Jackson (see also Melling, Oceanography Mag, 2011) may indeed provide a strong disincentive for the accumulation of snow.

The authors must not have looked carefully at the suggested reference before making their comment: "we are talking about the central Kane Basin and Peabody Bay. We don't think the orographic effect plays the same role in snow accumulation rates as it does in Kennedy Channel". In fact, as clear in Samelson's paper, fig. 2, 6 and particularly 9, the areas of strongest gap-flow winds are not in Kennedy Channel but in north-central Kane Basin, off Cape Jackson and south of Smith Sound. Also, I suspect that there is a strong likelihood of katabatic winds off the Humboldt Glacier into Peabody Bay in winter, but the authors seem not to explored this possibility. A renewes look at these issues is highly recommended.

Line 159, "… generally have good agreement with the mooring records": It is necessary to provide an assessment that is more specific in relation to the comparison of model with data in relation to the cross-channel scale of flow features, their positions cross-channel and in depth and their intensity. Can the countercurrent on the Greenland side be simulated?

Good

Line 182, "MODIS imagery confirm that a polynya is present every winter at Cape Jackson": The sentence that follows that quoted indicates that the following is more precise: "MODIS imagery confirms that in every winter when fast ice fills the strait, a polynya appears at Cape Jackson late in the season".

"… within the bridge": See comment re line 67.

Good

Line 190, "may indicate either the ice-free surface or thinner ice": Clarification, "may indicate either the an ice-free sea surface, locally thinner ice, locally thinner snow or both the latter".

Good

Line 205, "… If 50% of the 0.26 m surface elevation is attributed to a snow layer …": The occurrence of very strong, very turbulent winds off sea capes is well known to mariners. Cape Horn and Cape Farewell, at the southern tip of Greenland, are perhaps the most famous. See Winant et al. (1988) J. Atmos. Sci. 45. Such conditions would be very effective at scouring snow from the surface of sea ice and moving it downwind. It is therefore quite plausible that both ice thickness and snow depth become thinner on approach to Cape Jackson, as the density-stratified oceanic and atmospheric flows accelerate in response to submarine and subaerial topography blockage, respectively. IceSat may be sensing environmental response to both these effects, not just to one or the other.

Okay

Lines 218-238 & Fig. 4, "along-track anomalies averaged over 1x1 km squares": On the "basin-wide scale" discussed here, the anomalies, calculated relative to mean height of any ascending or descending track crossing the bridge between 55-76°W and 78.25-82.5°N, may well be contaminated by a varying along-channel gradient is sea-surface height – see comment on lines 133-134. It is appropriate that the authors acknowledge this source of error and discuss its impact on results.

Okay

Lines 231-242 & Fig. 4, "In the main channel, the anomalies are highly irregular and form a speckled pattern, whereas the anomalies in Peabody Bay form a consistent pattern with positive anomalies in the southeast and negative anomalies to the northwest": It is unclear, with the continually moving ice of 2019, why the elevation

anomalies are not smoothed out via averaging over time. The small scale of the speckle in elevation in 2019, not so different from that in the years with immobile ice is difficult to understand. Please explain.

Perhaps the authors could include in the manuscript, for readers' benefit, a truncated version of the explanation provided in their response to me?

A similar speckled pattern of $h$ was observed over the landfast ice in Peabody Bay in 2020 (Fig. 4b), but not in 2021. What is the application in these instances?

Good

Lines 233-234, "The difference in surface height anomalies between the southeastern and northwestern parts of Peabody Bay is supported by a similar difference in the observations of Tb" : In what sense do we interpret "is supported by"? Do you mean "is correlated with" or is there some physics behind the claim of support?

Good

Line 234: Interpretation of AMSR brightness temperature. Please clarify whether the values depend on emissivity (ice type ) as well as on surface temperature (of snow, of ice, or of somewhere between?).

Good

Line 235: Should "southwest" be changed to "southeast"?

Good

Line 280, "we applied the 1-D thermodynamic ice growth model": Things like thermal coefficients, snow density, short and long-wave radiation, cloud cover do matter. Please provide a quick overview of the properties of this model, or an equivalent citation.

Good

Lines 282-284. "We used 4 cm mo$^{-1}$ snow accumulation rate to reach a modest snow thickness of 14 cm at the end of winter that is reasonably close to 19±2 cm obtained with AMSR2 data for Peabody Bay": As mentioned earlier, snow accumulation matching that in Peabody Bay may be unlikely. Ice off Cape Jackson may be blown clear of snow by frequent extreme winds in winter (see Samelson and Barbour, 2008: Fig. 6). It would be appropriate to mention this possibility.

Good

Lines 288-289, "For having ice-free water in May, the heat flux should reach 70 W m$^{-2}$ and be above 200 W m$^{-2}$ to let polynya form in early March": These estimates presume that there is no advection of newly formed ice downstream and beneath thicker pre-existing level ice and, I believe, that there is no insolation.

See  my comment under (4)

Lines 440-441, "Although the northern branch is warmer and, being considerably faster, transports more heat compared to the southern branch …": Unfortunately, the northern branch is partially blocked from entering eastern Kane Basin by a shallow (70-90 m) spur extending more than 100 km southwest from Cape Jackson. The deepest crossing is relatively shallow, a 220-m sill at 79 40'N close to the Ellesmere shore. Moreover, because of geostrophic adjustment in the Arctic outflow, the warm mAW is at it deepest on the western side of the basin. To make a convincing argument about the temperature of the water that gets over this sill, more careful thought is needed. Where does the mechanical energy to lift water of the sill come from? I don't believe that a numerical model unvalidated in Nares Strait is a substitute for data needed to substantiate an hypothesis. Perhaps the authors could strengthen their case by exploring what the model has to reveal about the energetics of the phenomenon that they propose?

Please excuse my misunderstanding. On re-reading, I don't know how I was misled, but good that you made an effort to keep others off the wrong path.

Lines 456-457, "However, it is noteworthy that all these iceberg chains are located within the region with pronounced negative anomalies of ice surface heights in 2019 and 2020": Qualitatively, from the insets on Fig. 7, I estimate that the bergs cover only perhaps 10-20% of the sea surface; they could create point sources of turbulence kinetic energy through interactions with current, but are likely too sparse to form an area-wide

source to explain the sea-surface anomalies which are manifest on the scale of the entire basin. Moreover the warm seawater contacting icebergs at depth has plenty of opportunity to transfer its heat directly to the bergs, rather than hoarding to create havoc on the sea ice. The authors' hypothesis is plausible, but it needs appreciable quantitative physics to convert it into an explanation appropriate to uplift from 100-250 m depth.

Good

Lines 460-461, "However, the melting in this case is associated not with latent heat flux from water, but with dissolution controlled by solute transfer between water and ice-ocean interface (Woods, 1992)": I don't understand this point. I believe that a transfer of sensible heat to the iceberg is still required to free individual water molecules from the crystal lattice as dissolution proceeds. Please check whether you are citing Woods' work correctly.

Now irrelevant because of changes made in the revision.

Lines 496-497, "The stronger vertical mixing associated with the shear instability of the subsurface southward jet along the western coast … . This statement is speculative and not supported by observations. It is trivial to show with data in the Rabe papers that the gradient Richardson Number in the shear layer above the jet is about 2.2, almost 10x the threshold for shear instability. The most plausible sources of turbulence kinetic energy are in the wintertime mixed layer, namely shear in tidal currents at the base of rough sea ice and, less important with thick ice, brine-driven convection. Both can be estimated. I recommend that the authors do so.

Good

Lines 506-507, "generally support the idea of topographically controlled instabilities associated with the mean current and reversible tidal flow": I don't think it necessary to speculate about submerged topography generating instabilities. Headlands, by partially blocking along-shore currents, are notorious for strong tidal currents, and under-ice topography in Nares Strait is very rough.

Okay

Line 511, "probably through the local upwelling": What is the basis for "probably". I don't believe that there are any soundings in Flagler Bay, so the existence of a sill is speculative.

Good

Lines 543-544, "Münchow (2011) reported a very similar warming in the southward branch of mAW of 0.23 °C/decade": Actually Münchow et al. (2011). This paper provides very weak evidence of long-term warming because the period of observation was only 6 years. The present authors have taken the liberty of extrapolating this to 10 years, and then referring to a supposed "as further warming of mAW progresses" – all this without having made a bullet-proof case for an influence of mAW on the sea ice of Nares Strait. It is one thing to have mAW affect glacial ice at the same depth, quite another to postulate an influence on sea ice at the surface hundreds of meters above. I suggest to the authors that the present evidence to make this projection is not statistically robust.

I am okay with the first part of the authors' response, but not with the second, namely "We … have reduced the emphasis on the future projections. However, we can't simply reject it because, if our suggestion about the impact of mAW on landfast ice is right, the warming of this water is important to mention".

The issue in contention is the difference between scientific publication and news publication. In the latter, the primary goal is to sell newspapers. In science we strive for robust results with carefully estimated bounds of uncertainty, numbers that we can provide with confidence to other scientists to help in their work. We can all speculate that something may or may not happen. However, the key factors are how confident we are in what we know has happened, how confident we are that it will continue to happen and what will be the value some years hence, plus or minus what. Since these key factors have not been addressed, the statement about warming, "if it is right" as the authors say, is of little value to science and can be omitted. If the authors can address these key factors, then the statement certainly belongs in the paper.

---

## Referee Report (RR2)

General comments:

This manuscript presents a scientifically interesting body of work, with great use of a variety of remote-sensing datasets and a modeling exercise together to make novel inferences about Nares Strait ice bridge structure and breakup. The authors have done a good job addressing reviewers' questions and concerns, and have produced a much clearer and polished revised manuscript. Assumptions and limitations of the results are much improved, and the storyline is a lot tighter. I am very impressed with the creativity in how these datasets were used and am looking forward to referencing it once published. I have a few comments and points of clarification (stated in the attached document) that should be addressed, but otherwise I recommend acceptance with minor revisions.

Specific comments:

121-125– Great addition so far. However, MODIS is thermal infrared not microwave, and brightness temperature is also impacted by cloud and atmospheric constituents/temperature.

444-450/Figure 10 – Apologies for the not-quite-complete comment on this during the first round of review. Your edits here improved this part of the discussion a lot despite a lack of complete info on what I was meaning. What I had meant to say is that when I looked at the timeseries of MODIS thermal images in WorldView, the image you show from 2019 seems like it may be a high wind event pushing ice away from the coast (maybe a latent heat polynya of sorts, which would not necessarily be indicative of high ocean temperatures and could cause high sea ice production) and not the best representation of the Tb for that month. It's an extreme and ephemeral event for the month. A few days earlier and later in Dec 2019 looked very different. A similarly "warm-looking" event occurred in the 2018/2019 season for just a day or two. However, looking at all of the images from Dec 2019, I would agree that Tb was generally higher. A monthly mean Tb or timeseries of clear-sky days for the area would be much more useful in supporting your statements here for this reason. I don't think that a change is required for publication here, although it would make for a stronger statement and it could give you more insight about the relationship between sea ice height and Tb, as well as understanding mechanisms impacting bridge formation and breakup. I also use this as a caution against using extreme synoptic event days (Dec 18, 2019) to represent a monthly/interannual difference. That said, in general this section of text is oversimplifying a lot about Tb and not accounting for the complex sea ice-ocean-atmosphere interactions that are taking place. These statements are not quite accurate to what you can say from the figures you show without further explanation:

444-446 – "The high Tb conditioned the ice-free area in Peabody Bay", "signatures of warmer water in Kennedy Channel can also be traced through leads within the mobile sea ice" – this ends up being a little misleading for the reader. Since the temperatures you are showing aren't sea surface temperature and sit below saltwater freezing temperatures, it isn't clear what is causing the high Tb in 2019 or in the leads. The high Tb may merely be arising from a lack of sea ice at the surface and cold ocean surface temperatures (ocean is warmer than sea ice). The high

Tb and lower sea ice concentrations could arise from low sea ice transport out of the Arctic and/or a windstorm blowing sea ice out of this area (which also could drive upwelling of warm water but doesn't necessarily need to do so to produce the Tb pattern you see). It may or may not also be from warmer ocean water coming to the surface, but I doubt that is happening in isolation from the other phenomena I mentioned. These comments also play into line 488-489.

447 - "may indicate reduced ocean heat transport towards the surface from below" – this is technically true with the use of "may", but it would be helpful to present the other alternatives so that the argument is balanced for the reader.

447-449 - "the difference in temperatures presented in these early winter snapshot images cannot explain the seasonally averaged elevation anomalies shown in Fig. 4" - The notion of high winds/low sea ice transport into the strait causing the high Tb can more easily explain why sea ice thicknesses aren't consistent with the high Tb in 2019 than higher ocean heat flux to the surface can. For instance, more sea ice cover on the surface early in the season could insulate and prevent ocean heat escape that impacts sea ice formation afterwards. This could be discussed a little more in the text.

605-608 – Since most of the iceberg discussion was removed from the paper, this no longer applies, correct?

Line comments:

29 – ", which" makes the sentence a little clearer here

50 – not sure it makes sense to include "the last bridgeless winter" here since it is the most recent winter. I would cut this.

51 – points

67 – spots based on evidence from nearby…?

85 – Cape

103 – Spell out FESTOM-2 acronym here for first use, rather than on L170

125 - "did not"

130-131 – the segment lengths are in reference to ATL07 specifically and not all ICESat-2 data, so I recommend moving this sentence down one line to come after ATL07 is introduced.

134-137 – Have "although" and "however" in same sentence so recommend removing "Although"

140 – into "a" 1.5….

145 – Compared to

150 – Spell out for the first use of acronym (DMSP/SSM/I-ISSMIS), slashes are in the wrong spot at the end also.

161, 162, 166 – Spell out acronyms for first use?

166 – "a" finer

178 – the model performs well in reproducing the shift…

181 – depth, which is in good…

196 – lower than ideal for resolving

198 – calculated using

199-201 – what is your evidence for this statement?

217 – presence of a sensible

218 – remove 'sea surface'

220 – I think you don't want "the latter" here

238 – central main seems redundant so maybe this should just be central

271 – corresponds

278 – compared to, also if you have it, it would be good to give a quantitative order of magnitude estimate for each of the gradients

292 – spell out SAR

303-304 – we will roughly estimate … or even "will estimate"

331 – to the 19….

336 – would need to reach 70

338 – at some distance from the polynya

339-40 – is large enough () to keep it ice-free…

341 – to the 0.26 m mode

341-343 – A set of model experiments showed that the maximum…corresponds to … and a snow accumulation rate…

From L343 to the end  – stopped noting grammatical errors, but quite a few still exist beyond this point

365 – spell out first use of AO – but since it is the only use could remove the acronym

383 – how much shorter? Could give ranges or the average for each

Figure 10 – What is the vertical-ish line in (b) and (c)?

459 – what does "contrast of the latest" mean?

462 – away from Cape Jackson

491 – the "only" available source – this isn't accurate based on your second sentence

527 – "ice elevation anomalies"

530-532 – "by the fact of forming the chain of polynyas" to "by the fact that chains of polynyas for…"

532 – Fig 11 should probably be referenced the sentence before.

---

## Author Response (AR2)

In the following, the comments by the reviewer are in green (old comments) and black (new comments) characters and our responses to the comments are made in blue and indented. The comments where the reviewer was satisfied with our initial answers and/or changes were left out of this document for simplicity. Modifications to the text are shown in quotation marks with bold characters indicating newly added text, and normal characters indicating text that was already present in the previous version.

**The role of oceanic heat flux in reducing thermodynamic ice growth in Nares Strait and promoting earlier collapse of the ice bridge, version 2**

Sergei Kirillov, Igor Dmitrenko, David G. Babb, Jens K. Ehn, Nikolay Koldunov, Søren Rysgaard, David Jensen and David G. Barber

**Overview**

I have restricted my present comments on the revision to the author's diligence in addressing my original comments. However, I admit that I did find the authors' responses to seemingly valid comments by the other reviewer unsatisfactory at times. For example:

Other reviewer's comment: "There is currently no overlap between the ICESat-2 sea ice heights/AMSR temperatures (2019-2021) and the model results (2006-2010). It might be helpful to extend the AMSR temperatures back to 2006 to provide some comparison and context. It cannot necessarily be assumed that 2006-2010 have the same circulation conditions as 2019-2021".

Authors' responses:
1. "We used the model of opportunity and were not able to choose a different simulated period."

*There is nothing wrong with saying this, but the future readers of the paper need to know this unfortunate reality too; therefore the gist of this response should be added to the text.*

> *Done*

2. "However, the main intention of the model was to demonstrate the circulation under the ice bridge. Since the main factors controlling water dynamics in this region (the along-strait sea-level gradient and the prevailing northern winds) don't vary a lot interannually, we reasonably suggest that the patterns shown in Fig.7-9 are generally valid and fairly represent (modeled) water dynamics and thermohaline state of Nares Strait in winter."

*I very much doubt that "the along-strait sea-level gradient and the prevailing northern winds do not vary a lot interannually", whatever "a lot" means. However, since this purported lack of variation is crucial to the authors' line of reasoning, the paper must provide citations or include data and discussion to persuade the reader that this is so. Without documentation, the statement is unacceptable.*

> *To address this concern, we analysed the sea level data in Alert (data taken from the Canadian Tides and Water Levels Data Archive) and Thule (University of Hawaii Sea Level Center) and found no considerable difference between these two periods (Dec-Jun 2006-2010 and Dec-Jun 2019-2021).*
> *The corresponding information was added in Section 2.3:*

*"Although there is no overlap between the ICESat-2/AMSR data (2019-2021) and the available output of the FESOM2 model (2006-2010), we assume that there is very little interannual variability in the circulation under the ice-bridge given that the two major factors controlling water dynamics in this region (the along-strait sea-level gradient and wind) do not vary much interannually. The mean difference in sea level between Alert (data from the Canadian Tides and Water Levels Data Archive) and Thule (University of Hawaii Sea Level Center) from December to June was 1.56±0.04 m in 2006-2010 and 1.59±0.05 m in 2019-2021. The mean difference of sea level pressure between Alert and Thule, that may be used as a fair proxy of wind speed in Nares Strait (Samelson and Barbour, 2008), also revealed no significant changes from 2006-2010 (3.26±0.22 hPa) to 2019-2021 (3.33±0.23 hPa)."*

I originally provided three overall comments, supported by more detailed comments cross-referenced to specific places in the first draft. The overall comments were:

- The authors neglected the established role of tidal currents in maintaining polynyas in the fast ice of the North American High Arctic.
- The accuracy, precision and possible bias of ice/snow-surface elevation and temperature measurements were not determined, because no in situ data were brought into play. Without error analysis of uncertainty, readers' confidence in the results obtained from satellite remote sensing is eroded.
- The lack of contemporary or past oceanographic observations at the locations of interest is a serious shortcoming in a paper that strives to attribute polynya formation to oceanic heat flux. The FESOM global ice-ocean model that has been harnessed in an effort to fill this gap seems not to have been evaluated in this tiny – from a global perspective – area. Readers' skepticism about the study' results is therefore likely to be high.

Here I am taking the same approach in assessing the authors' responses. I provide an overall assessment of the revision in the next section. Subsequently, I provide my original comment (black font) followed by my assessment of the authors' response in blue font.

**Assessment overall**

*Concerning "The authors neglected the established role of tidal currents in maintaining polynyas in the fast ice of the North American High Arctic".*

The authors' begin a semi-quantitative (and incomplete) speculation to argue that tidal currents can't work to create a polynya at Cape Jackson, but later fall back on a more plausible hypothesis: "we have to admit that our suggestion that upwelling brings warm deep water from Peabody Bay directly to the surface may be too challenging. In combination with vertical mixing, it may be sufficient to upwell this water just closer to the surface – to the bottom layer over the "ridge" dividing Peabody Bay and the central channel of the strait".

Correct! This is the hypothesis developed in the citations to which I directed the authors in my first review: Polynyas beneath fast ice in the Arctic need two things: 1) "warm" (probably at least 0.05C" above freezing) water locally at depth, almost certainly of higher-than-surface density; 2) a strong

local source of turbulence kinetic energy. In Nares Strait, baroclinic adjustment of the southward mean flow causes shoaling of warm water into shallower areas on the Greenland side, providing the shallow warm-water source beneath the polynyas. The strong local source of turbulence kinetic energy is almost certainly shear in the benthic and under-ice boundary layers of the strong tidal current. Tidal current, contrary to the authors' contention but well-known to mariners, is commonly accelerated around headlands like Cape Jackson; a narrow strait is not required. The generated turbulence erodes the stratification in density between the seabed and the surface and diffuses oceanic sensible heat, if present at depth, to the under-surface of the ice. An internal tide generated by the tidal flow across bottom topography, often serves to enhance to warm-water uplift in such areas. The strong tidal current additionally plays a role in sweeping newly formed frazil and nilas ice beneath or onto adjacent fast ice, except perhaps during neap tides and very cold conditions.

So, the authors are on the right track, but the suggested modest modification to line 536, namely "This heat may either be upwelled over the mid-basin ridge closer to the surface and/or transported upward to the lower surface of sea ice (or to the ice-free polynya) by vertical mixing" remains inadequate. The authors needs to lay out the requirements: warm water at depth, upwelling and a source of turbulence kinetic energy. Second, a paragraph is needed to properly enlighten the reader about the ocean and ice physics involved and third, the authors need to acknowledge earlier work.

As the reviewer mentioned, we already admitted the role of vertical tidal mixing in the formation of the polynya at Cape Jackson and the corresponding changes were made in the paper. Therefore, one of major disagreements now seem to be about the "established role of tidal currents in maintaining polynya" and "sweeping newly formed frazil and nilas ice" that are suggested by the reviewer to play a significant role in the maintenance of this polynya. Unfortunately, without in-situ measurements, we can only speculate about their roles and about a role of advection and upwelling as suggested in our paper. But the reviewer is definitely right in one thing – we have to at least acknowledge all mechanisms that could result in formation of the Cape Jackson polynya.

In the following, we will try to convince the reviewer (and readers) why we suggest that the mechanical removal of new ice and/or frazil ice and nilas removal play only a secondary role in a formation of this polynya and why a tidally-driven upwelling and mixing is not thought to be the only mechanism that is responsible for vertical heat exchange in this area. Note, that the suggested text changes are presented further in our responses to the particular reviewer's comments.

1)  We don't argue with the reviewer that tidal currents are "commonly accelerated around headlands like Cape Jackson". However, this is a general statement and we can't use it as granted without more robust data on the tidal flow structure there. Although the reviewer also mentioned that "a narrow strait is not required", all sensible heat polynyas surrounded by landfast ice are located in the narrows according to Hannah et al. (2009) and other papers. If there are any examples of such polynyas not constrained between landmasses, we are not aware of them. According to WebTide model, the spring tide current velocities in the vicinity of Cape Jackson is about 48 cm/s. We understand, that the model may not distinguish some local effects from prominent coastline features like Cape Jackson, but this is the only number we can operate with in this area. Interestingly, this speed is very close to what Munchow and Melling (2008) reported for KS10 position in Kennedy Channel (the combined effect of $M_2$, $K_1$ and $S_2$ constituents is 44 cm/s, whereas $M_2$ and $K_1$ constituents at KS14 were even larger).
    The 48 cm/s is much smaller than the peak 1.2 m/s measured in the polynya at Dundas Island during spring tide and for which Topham et al. (1983) described a mechanism of the mechanical removal of new ice. Within other sensible heat polynyas in the Canadian Arctic, the tidal currents

may be even faster (for example, in Cardigan Strait and Hell Gate) that makes all those areas not very comparable with the not-extremely-prominent Cape Jackson.

2) The horizontal excursion of a tidal flow with the peak velocity of 48 cm/s is 6.8 km. Using sparse bathymetric data, we could estimate the vertical displacement near the bottom corresponding to such excursion as ~30 m - from about 110 m to 80 m. We agree that such displacement is enough to uplift warm upper halocline water to the base of the water column in the polynya area (where it can be further diffused to the surface by the enhanced turbulent mixing). Although this argument may support the idea of tidally-driven vertical upwelling of heat, we don't agree with the reviewer that upwelling is a "mysterious" process. The wind-driven mechanism is not the only possible way to uplift water. A horizontal convergence of mean currents near a wall may also result in a local upwelling. Unfortunately, the dearth of bathymetry data and some limitation of FESOM-2 model don't allow us to estimate its real occurrence in the northern Peabody Bay. However, the regional circulation within the bay generally supports an existence of a convergence zone in its northern part as streamlines congregate along the northern coast:

[Figure]

3) As a continuation of the previous point. The area of thinner ice in the northern Peabody Bay is associated not only with the polynya at Cape Jackson. From Fig.4b and 4c, one can see that the negative anomalies of ice elevations spread over about 50-km from Cape Jackson to Cape Webster and further east. We strongly doubt that this entire area is affected by tidally-driven upwelling. We could admit that local tidal upwellings take place at specific hot spots (like Cape Webster and Cape Jackson) with further advection of the upwelled heat over a relatively long distance along the coast. However, if such combined mechanism works here, why don't we see negative coastal anomalies beyond Cape Jackson – at the entrance to Kennedy Channel?

4) We may partly agree with the idea of mechanical removal of nilas and frazil ice from the polynya. However, in some years the polynya is almost invisible even at the end of winter season (e.g. in 2008, 2011, 2018 in Fig. 2). And the polynya is invisible in early March during most of the years, opposite to the polynya at Dundas Island where an ice-free area is present almost every March since 2000 according to Worldview data. The presence of a relatively thin but solid ice cover near Cape Jackson and also along the entire northern coast of Peabody Bay suggests that mechanical breaking and removal of thin ice or frazil ice can not be considered as factors limiting the ice

growth in these areas and that the sea ice would continue to grow (Topham et al., 1983) if there was no ocean sensible heat flux.

5) To demonstrate that the tidal forcing at Cape Jackson is not strong enough to play an essential role in a mechanical breaking of thin ice, we also analyzed the time series of composed daily images of the polynyas at Dundas Island and Cape Jackson in April-May 2020. We used ALL daily images where clouds allowed to see individual ice floes.

[Figure]

Figure. The polynya at Dundas Island

A combined effect of the strong tidal dynamics and wind forcing in the Dundas polynya result in clustering of the ice floes near polynya edges most of time (see the figure above) that is favorable for preconditioning the mechanical removal of ice. But the situation is different in the polynya at Cape Jackson (see the figure below).
Taking 3 km as a length of opening at Cape Jackson we can find that it would take less than 3 h for sea ice to move from edge to edge by tide with a maximum speed of 48 cm/s. Therefore, one may

expect that satellite images would have captured ice floes clustering near edges in more than 50% of all images. Or even more often, if the tidal flow at Cape Jackson is accelerated around the cape and exceeds 48 cm/s. However, we mainly see a quite random distribution of ice floes with numerous floes idling in the middle of the polynya almost in every satellite image. Note that we didn't select "good" daily images, but took ALL images when cloud conditions allowed to see the individual ice floes in both polynyas.

Based on this visualisation, we may suggest that the effect of tides in the Cape Jackson polynya is not very strong. Or at least considerably weaker than in the Dundas polynya where the process of mechanical removal of new ice was considered by Topham as a key factor responsible for keeping the Dundas polynya ice-free.

[Figure]

Figure. The polynya at Cape Jackson

Summarizing all these points:

We admit that tidally-driven upwelling and enhanced vertical mixing are likely responsible for maintaining the invisible or ice-free polynya at Cape Jackson. However, few moments allow us to suggest that upwelling associated with a convergence of mean currents in the northern Peabody Bay may also be present in this area. Regardless of this, we don't think that mechanical breaking and removal of ice plays a considerable role in formation of the polynya at Cape Jackson. The removal of frazil ice and nilas is not thought to contribute much either because most of the areas with negative anomalies are invisible polynyas, i.e. covered with thinner but solid ice. Please see the suggested changes addressing these and other reviewer's concerns further in this document.

*Concerning "The accuracy, precision and possible bias of ice/snow-surface elevation and temperature measurements were not determined."*

See my critical responses below to comment 2, 3, and comments on lines 38, 46-47, 132 and 533-534. Work is still required

*Concerning "The lack of contemporary or past oceanographic observations at the locations of interest is a serious shortcoming in a paper that strives to attribute polynya formation to oceanic heat flux."*

This comment remains difficult to redress, but it should be straightforward to acknowledge and document shortfalls of in situ data. I have continued in the detailed comment below to urge the authors to keep readers aware of all the ifs and buts associated with story that they are telling.

*Overall the manuscript remains too speculative in its "explanations". It lacks brief digressions to present simple physic or statistics that could provide some reassurance of validity to speculative interpretations offered. In detailed comment below, I have continued to remind the authors of their responsibility in scientific publication to anticipate the queries of skeptical readers, and to provide enough scientific underpinning to bring this readers along with them set these readers at ease.*

**Major revision required: This revision is an improvement, but remains in need of further revision.**

**Comments (major)**

2. Line 148, "The model was driven by the atmospheric reanalysis fields from JRA55-do": This was probably not a good choice. Samelson and Barbour (2008) concluded that a grid 10x finer than that of JRA55 was required to correctly represent weather conditions in Nares Strait. I recommend adding a discussion the capability JRA55 to represent the mesoscale meteorology of Nares Strait, a channel much narrower than 55 km in width for much of its length, bordered by high terrain and characterized by a strongly stable atmospheric boundary-layer – the Arctic inversion – during the freezing season. This could be perhaps achieved via comparison of simulations by the JRA55 and Polar MM5 models.

I am pleased that the authors' have added a sentence discussing an intercomparison of ocean current modeled independently using wind from 55-km and 6-km atmospheric grids. However, because the graphics they provided in their response will not appear in the paper, I take issue with their claim that "… the modelled current velocities in Kennedy Channel and in Smith Sound fairly coincide (not shown) with velocities obtained by Shroyer et al. (2015)". I acknowledge that the models produce flow crosssections that are qualitatively similar. However, quantitatively they are appreciably different; Schroyer's sections show the speed of the undercurrent reaching about 45 and 65 cm/s in the two sections shown, whereas those in the FESOM sections reach only 25 and 40 cm/s, just 55-60% of those with high-resolution forcing.

Because the models do not honestly "fairly coincide", I recommend that the quoted clause be revised to read "… the modelled cross-sections (not shown) of average November-June ocean current through Kennedy Channel and through Smith Sound driven by winds from FESOM-2 and from MM5 (9x higher resolution: Shroyer et al., 2015) are qualitatively similar but those derived using MM5 forcing yield currents that are 60-80% larger".

> The top speed within the undercurrent jet at CATS line is 26 cm/s in the FESOM simulations and from 35 to 40 cm/s in Shroyer's paper (isotaches are separated by 0.05 m/s in their plots, not by 0.1 m/s!). In Smith Sound these numbers are 47 cm/s in FESOM and from 45 to 50 cm/s in Shroyer et al. It means that FESOM gives lower speeds in the Kennedy Channel (65-74%), but almost similar results in Smith Sound (94-104%), which is better than the reviewer initially suggested. We would also like to note that the observed difference in the Kennedy Channel may not be entirely attributed to a different resolution of atmospheric forcing. Using the different period of averaging (2006-2010 in FESOM and 2005 in Shroyer et al.), different parameterizations of ice cover etc. could also play a role.
>
> But the main purpose of the model was to qualitatively show the circulation patterns in Nares Strait and particularly in Peabody Bay. The model speeds themselves were not used in our paper.
>
> To address this concern, we changed the sentence as follows:
>
> "*Despite using a low resolution JRA55 product that is probably unable to reproduce orographic strengthening of wind within the relatively narrow Nares Strait (Moore and Våge, 2018; **Moore, 2021**), the **structure of flow** in Kennedy Channel and in Smith Sound **obtained with FESOM2 qualitatively** coincides (not shown) with **that** obtained by Shroyer et al. (2015) who used the Polar MM5 regional atmospheric model, which has a finer horizontal resolution of 6-km. **Although the maximum speed of the southward jet in Kennedy Channel in FESOM2 simulations is about 30% lower compared to Shroyer's results, the speeds in Smith Sound are almost the same.***"

3. Line 166, "The 6-hourly records of 2 m air temperature, wind speed and humidity in Kane Basin were taken from the ERA-5 global reanalysis database": The 31-km grid of ERA5, comparable to the width of Nares Strait, does not come close to resolving the channel and very its steep surrounding terrain. I suspect that much of the strait's "sea surface" resolved on a 31-km scale will actually be above sea level and therefore "terrestrial". It is very difficult to accept that ERA5's 2-m air temperature values hold much credence for simulating ice growth in the real world. Since sea ice loses the upward flux of latent heat from wintertime freezing by long-wave radiation from its top surface, would you not be better using satellite-derived surface radiation temperature to model freezing? Please strive to persuade readers that my viewpoint is invalid.

I am relieved that the authors acknowledge the serious limitations of the ERA5 meteorological analyses in Nares Strait where variations in substrate (water, sea ice, land, ice cap) and elevation (sea level to 1000m) occur on a scale of a few km. I am pleased that they have added a paragraph that explains their goal in using the ERA5 forcing. They are also correct in stating that local-scale surface air temperature takes second seat to local snow depth in driving variations in ice growth rate.

Unfortunately, as mentioned in my earlier review, the true sea surface off Cape Jackson, which would take the second seat, will likely lie hundreds of metres below where ERA5 "thinks" it is, because sea-level features as narrow as Nares Strait (40 km) probably do not exist within the ERA5 grid. Hundreds of metres in the polar atmospheric boundary layer span tens of degrees in temperature, which is non-trivial for ice growth.

We have to admit that we did not address this particular concern in our previous response. And first of all we need to apologize for not showing from the beginning the location of ERA5 data used in our paper. For our ice growth model, we used ERA5 data taken at 66.75W, 79.625N (the position was added to Fig.1) – at the center of Peabody Bay. The bay is part of the relatively large Kane Basin which is ~130 x ~200 km and able to accommodate a few zonal and meridional ERA5 nodes. We agree that such resolution is not good for resolving orographic effects for example in Kennedy Channel still, but it is thought to be good enough to suggest that ERA5 surface data represents the surface layer characteristics in Kane Basin reasonably well. For example, Fig.10 and 11 in Moore (2021, www.nature.com/articles/s41598-021-92813-9) support the similarity of 30 km ERA5 and 9 km ECOA surface wind data in Peabody Bay.

The following sentence was added in Section 2.4: "*We used data taken from the central part of Peabody Bay (66.75W, 79.625N, the yellow star in Fig.1) where orographic effects are pronounced than in the main channel of the strait (Moore, 2021).*"

Moreover, the authors state in the response (but not in the added paragraph) that "a spatial variation of any meteorological parameter over such a short distance is negligibly small. Therefore, any local anomalies of ice thicknesses here are thought to be controlled by spatial variations of ocean heat flux rather than varying meteorological parameters". This statement may be true on the 1000-km scale of synoptic weather forecasting, but it is not true at the surface within a few tens of km of a change in substrate, such as the edge of a polynya. The transition from fast ice with a surface temperature of perhaps -25C to open water or thin ice is likely at least 20C. Moreover, the atmospheric boundary layer, warmed and moistened as it moves over the polynya, is quickly covered by ice fog and further downwind stratus cloud (the cause of "water sky" over polynyas). Cloud droplets have high emissivity and their presence can reduce or perhaps reverse the upward heat loss by long-wave radiation that drives ice growth. Here huge differences occur on the mesoscale in this environment.

We do not agree with the reviewer here. First of all, our phrase was related to the landfast ice-covered part of Nares Strait only, not to any system with polynya(s). We are aware of all effects associated with open water in winter, but would like to underline that the size of polynya at Cape Jackson is very small and the polynya represents an isolated spot surrounded by hundreds of kilometers of ice. Therefore, its effect is incomparable with the large latent heat flaw polynyas like NOW that may affect a state of the boundary layer over a few hundred kilometers. In addition, the polynya at Cape Jackson is mainly invisible throughout most of winter: an opening barely reaches several hundred meters in early March and it starts to grow in April to reach only about 3-4 km at the end of the month when air temperature is getting higher and air-ocean temperature contrast decreases.

And, with all respect to the reviewer, we suggest that the key point in that phrase was not taken properly. We meant ALL anomalies of ice thicknesses including the coastal anomalies along the western part of the strait and in the northern Peabody Bay. Those anomalies are not associated with open water, just thinner ice that can't change meteorological parameters over these invisible

polynyas much. An effect of open water on the boundary layer near Cape Jackson is doubtless. At least at the end of winter when the polynya turns ice-free. However, even here this effect is a consequence of the polynya, not a cause of its formation.

We added the following text in Section 4.1 to discuss a possible effect of an altered balance over the polynya at Cape Jackson on our ice growth calculations there (Fig. 6c):

"*It was found that keeping the polynya open in early March requires a relatively large (>200 W m-2) additional, presumably sensible, ocean heat flux to compensate the heat loss to the atmosphere. This amount was obtained from the model forced by atmospheric conditions in the central part of Peabody Bay. The polynya at Cape Jackson is usually ice-covered (invisible) during most of winter, starts opening in early March and reaches approximately 3-4 km in size by the end of April. Considering that the model covers the period from December to April, we may suggest that a shift in radiation balance caused by the presence of moister air over open water within the polynya mainly occurs at the end of winter and using ERA-5 data from the central Peabody Bay results in a relatively small overestimation of the sensible heat flux needed to keep this polynya nearly ice-free. From our point of view, the more considerable effect may be the mechanical removal of frazil and nilas ice at the end of winter when the presence of open water results in more intensive formation of ice crystals that may be carried away by advection (e.g. Smith et al., 1990). The full loss of heat to the atmosphere in this case is not entirely compensated by the sensible heat flux from below, but partly explained by the latent heat of ice formation. Although it is difficult to estimate an impact of frazil ice removal and shift of atmosphere boundary layer characteristics on the ice balance without specialized in-situ measurements, we suppose that the reported 200 W m-2 as well as 70 W m-2 heat fluxes may not be entirely associated with the sensible heat and, therefore, overestimated. Note also that another possible mechanism of ice loss through breakage and removal of relatively thick new ice floes associated with strong tidal currents and winds (Topham et al., 1983) is not thought to play a considerable role in the polynya at Cape Jackson. The analysis of daily Sentinel-2 images in April-May 2020 (accessible through Sentinel Playground hub operated by Sinergise Laboratory for geographical information systems, www.sinergise.com/en) revealed that the individual ice floes idle in the center of the polynya most of the time, instead of clustering near edges as they usually do in the Dundas polynya studied by Topham et al. (1983). It implies that tidal currents at Cape Jackson are not thought to be strong enough to initiate breakage and removal of new ice from the polynya here.*"

The authors have also responded with a couple of plots (to me, but not in the revised manuscript) showing probabilities of wintertime air temperatures and wind speeds measured with the Automatic Weather on Hans Island (red line) and from the nearest node of gridded ERA5 product during the same period "… to demonstrate that ERA5 parameters are realistic". Unfortunately, for this comparison to have any value, we need to be absolutely sure that the Hans Island data were NOT USED in the ERA5. If they were, then the re-analysis result would be strongly biased to the data there, and quite possibly nowhere near as good elsewhere. Also, the conditions compared are for a rocky site high up (170 m) in the Polar Inversion. Its relevance to ERA5's capability to predict conditions at marine locations at the bottom of this inversion is questionable.
This is all to say that there is clear potential for atmospheric forcing to vary appreciably on the scale of the polynya because the polynya is capable of creating its own weather.

That example with Hans Island meteorological data was just an attempt to demonstrate the quality of ERA5 based on the only available measurements in the area of research. With a deficit of other available data, we could offer nothing else. We hope that our aforementioned answers to this comment could convince the reviewer that ERA5 data from the central Peabody Bay is the best data available to represent the state of the atmosphere in the surface layer.

This is also to say that it is not good science to make a statement such as the one that I quoted two paragraphs up without being very careful about the distinctions between physics of the atmosphere on a synoptic scale above the planetary boundary layers and that on the mesoscale within the planetary boundary layer. Sweeping generalization can lead one astray.

I don't know what to recommend here to move your paper along, since once again you/we are stymied by the absence of information that you really need, specific to your locations of interest. Rather than torpedoing the paper, I prefer to recommend that you add straightforward discussion to acknowledge and describe the complexity of the analysis that you are undertaking, and the fact that many of your assumptions are ad hoc rather than readily justifiable. Such an approach will enlarge the document, and the result will not be definitive, but a full declaration will be of greater value to future scientific studies than a pretense that everything is just fine.

We added the new text to Section 4.1 that contains discussion of a possible effect of altered boundary layer characteristics over the polynya on the estimated ocean heat flux (admitted to be overestimated) – see our response above. In regarding to using ERA5 data in the model, we added a paragraph in Section 3.3 that describes the sensitivity of the ice growth rate to the uncertainties of the meteorological parameters (air temperature, wind speed, snowfall rate) and ocean forcing (sensible heat flux):

"*To investigate a sensitivity of model results to the uncertainties in forcing parameters and/or possible biases of ERA5 data in Peabody Bay, we conducted several model runs with biased air temperature, wind speed, snowfall rate and ocean heat flux for the experiment presented in Fig. 6b. It was found that increasing wind speeds by 1 m s-1 increased ice thickness at the end of the season by 4.02 cm, whereas increasing air temperature, snow accumulation rate and ocean heat flux by 1° C, 1 cm mo-1 and 1 W m-2 reduced ice thickness by 2.45, 7.41 and 3.36 cm, respectively. Using these numbers, we can estimate a relative input of each parameter to the final results. Using 1 m/s, 2 °C, 2 cm/mo and 5 W m-2 as the possible biases or uncertainties of the model forcing parameters and taking the impact of ocean heat flux as 100%, we may estimate the relative input of other factors as 88% (snow), 29% (air temp) and 24% (wind speed) that underlines a larger contributing effect of snow and sensible heat flux on the ice growth.*"

4. Line 415, "It was found that the ocean heat flux at Cape Jackson needed to exceed 200 W m-2 to open the polynya as early as in March": The message intended here is unclear from the present text. Revision is required.

I don't believe that it is plausible for a polynya to suddenly melt itself into existence – oceanic heat fluxes just aren't large enough. The more likely role of oceanic heat flux is keeping the ice relatively thin (and relatively weak), so that more powerful mechanical (fracturing, rafting, flooding and downstream advection, etc.) and thermodynamic (radiation) processes can do their work. Principal among the former are the stresses exerted by wind, wind-waves and tidal current on already thin ice. Once these open a polynya, new ice (mainly frazil and nilas) created by high rates of heat loss (> 200 W m-2) from the surface will continue to be removed by current, while insolation and downwelling short-wave radiation may

deliver appreciable heat directly to seawater. This "tag-team" approach to creating a polynya in fast ice, involving both dynamics and thermodynamics, has been discussed by Topham et al., (1983. JGR 88). Actually lifting warm water 100 m or more to the surface to open a sensible-heat polynya requires a large input of kinetic energy to the ocean, which cannot occur with a continuous cover of fast ice. It is only possible when strong winds act on mobile ice or open water. There is a paradox because necessity for strong winds is the same requirement for the opening of latent heat polynyas; the distinction between the purported "two types" of polynyas is not clear (see discussion in Melling et al, 2001). Moreover, because the wind must be integrated over a large expanse of mobile pack ice to accumulate enough kinetic energy to drive upwelling, the formation of small polynyas in fast ice, such as that off Cape Jackson, via this mechanism is unlikely.

The authors appear to have misunderstood my response to the authors' original statement "It was found that the ocean heat flux at Cape Jackson needed to exceed 200 W m-2 to open the polynya as early as in March". However, this statement does clearly say, perhaps unintentionally, that the OCEAN HEAT FLUX must exceed 200 W m-2 to open the polynya as early as in March. They have come back in their response with an image of the polynya, showing thin ice and perhaps open areas and a statement that "If water surface is ice-free, 200 W/m2 does not seem to be extraordinarily high". I agree with this statement, with the reminder that if the surface is ice-free, that heat flux is derived from the latent heat of freezing for seawater and not from sensible heat in the ocean. The authors appear to have confounded these two fluxes.

Sorry for misunderstanding this comment first and for not addressing it. We added a paragraph in Section 4.1 that addresses this issue. Please see our response to the previous comment

Continuing their response, the authors write "The mechanical processes maintaining water at Cape Jackson ice-free suggested by reviewer can work to a certain extent only. Occurring within very small, landfast-ice constrained area, none of these processes may prevent the polynya to eventually become covered with ice unless the ocean heat flux is strong enough to melt it". The authors are correct in writing "none of these processes may prevent the polynya to eventually become covered with ice" but a cover of thin and young ice does not terminate the existence of a polynya. Polynyas are not necessarily open-water areas; they are areas with ice cover "appreciably thinner" than adjacent areas of ice on all sides. Many polynyas exist with a thin ice cover through the winter; at times when covered by a new fall of snow, they may be difficult to distinguish from surrounding ice, but they are still there. They only become ice free in the darkness of winter via the action of wind and current in fracturing the thin ice cover and carrying it away – a so called "latent heat polynya". As winter wanes, sometimes in April but more reliably in May and June, insolation will strong enough, at hundreds of W/m^2, to melt the ice.

The polynya at Cape Jackson is ice-covered (invisible) during most part of the winter and usually starts opening only in March. This made us think that mechanical removal of frazil ice or sheets of thin ice does not play a significant role in this polynya. Please see our response to your Overall Assessment and the suggested changes in the Section 4.1 in the response to the previous comment.

The authors last point is "If such process had taken place at Cape Jackson, the continuous accumulation of large amounts of frazil/new ice from polynya below landfast ice sheet would have resulted in much thicker surrounding ice close to edge (at least in certain directions)". The advection of frazil is well known in both oceans and rivers; under level ice, the crystals remain suspended in the

turbulent surface waters and travel far, spreading their influence thinly over an area much larger than the polynya. Remember that tidal currents rotate and typically flow in every direction in the course of 12 hours.

*A possible impact of frazil ice was discussed in the new paragraph added to Section 4.1 (see our response to the previous comment)*

Under sea ice, frazil may settle against ridge keels, or within their porous structure, with a consequent increase in keel consolidation that is undetectable from space.

*We don't think it's a correct statement. The consolidation of frazil ice within interstices of keels would increase a buoyancy of a ridge and result in higher surface elevations.*

Flooding, downfolding and submergence of sheets of new and young ice on the downwind, down-wave or down-current sides of polynyas in fast ice is well documented. It the process responsible Joseph-René Bellot's drowning in 1853 and has provided a close call for many others. Sheets of young ice have low buoyancy and can travel large distances in turbulent flows before the water between them and the overlying ice sheets is squeezed out.

The authors' arbitrary dismissal of earlier research on and knowledge of High Arctic polynyas in fast ice is not acceptable. Their continued insistence that the ocean has the capability of providing an unprecedented 200 W/m2 flux of SENSIBLE heat by some mysterious physical mechanism to an open sea surface continuously throughout the winter is highly unlikely. If they wish to stick with this concept, a great deal of careful work and explanation will be required to make a convincing case, if such can actually be done.

*Although we really did not consider such mechanism from the beginning, we have certain doubts that it is workable for the polynya at Cape Jackson. As we explained earlier, the polynya at Cape Jackson is mainly invisible during winter that considerably reduces a possible effect of thin ice removal as a source of extra (latent) heat to compensate the heat loss to the atmosphere. In the new text added to Section 4.1, we discuss a possible impact of both altered boundary layer characteristics over the polynya and frazil/new ice removal. We agreed that the estimated sensible heat fluxes of 200 W/m2 and 70 W/m2 seem to be overestimated. See the changes presented in our response to the comment #3.*

The revised text in lines 358 and 497 is therefore unacceptable. That in line 230, concerning the appearance of open water in March off Cape Jackson is okay if the phrase "as the daily downwelling shortwave flux increases with the coming of spring" is appended.

*The sentence in Line 358 ("Within the polynya, in order to have open water in May the heat flux would need to reach 70 W m-2, while a heat flux above 200 W m-2 is required to form an ice-free polynya in early March.") just says how much additional heat is needed to keep the polynya open. Although we really suggested initially that this flux is attributed to the sensible heat, there is no mention of it in this paragraph. However, we changed one of the sentence above to avoid any confusion:*

*"Adding in the*  *heat flux **from below** lowers this thickness and also shifts the timing of maximum ice thickness."*

In the new paragraph, added after Line 497 in Section 4.1 (see our response to the previous comment), we discussed the role of other mechanisms (mechanical ice removal and shift of energy balance over polynya) that could result in our overestimating the ocean heat flux at Cape Jackson attributed to sensible heat. In particular, the followed sentence was included to that paragraph:

*"Although it is difficult to estimate an impact of frazil ice removal and shift of atmosphere boundary layer characteristics on the ice balance without specialized in-situ measurements, we suppose that the reported 200 W m-2 as well as 70 W m-2 heat fluxes may not be entirely associated with the sensible heat and, therefore, overestimated."*

We agree that it is a change of surface net balance that turns an invisible polynya to an ice-free area. The requested phrase was appended.

7. Line 484, "We suggest that the observed negative anomalies are attributable to the heat upwelled from the underlying mAW": I believe that this suggestion has merit, but that the details are incorrect.

The flow structure beneath fast ice in Nares Strait depicted by Rabe et al (2012; Fig. 4b), displays a jet of roughly 10-km width against the Ellesmere shore, centered at about 80-100-m depth. The baroclinic adjustment of the ocean to this jet (not shown) involves downwelling below the core of the flow and upwelling above. This leaves no mechanism to raise mAW through the core of the undercurrent to the surface on this side of the strait. Indeed, the cross-strait circulation that compensates for downwelling of mAW on the western side is upwelling on the other side, near Greenland!
However, upwelling does occur above the core of the jet. This would bring Pacific Winter Water as much as 0.2C warmer than the surface-freezing temperature (see Melling et al. 1984 Cont. Shelf Res 3) to the base of the surface mixed layer (see Melling et al. 1984 Cont. Shelf Res 3). This sensible heat in this Pacific Water could provide a heat flux to the underside of the sea ice via entrainment into the turbulent surface mixed layer. The needed turbulence kinetic energy could originate in part from brine-driven convection (ice growth) in the mixed layer and in part from shear between rough immobile ice and the rapid tidal flow. Melling et al. (2015) estimated an oceanic heat flux to the base of ice as 15 W/m2 under similar circumstances in Penny Strait, which would be sufficient to melt about 0.5 cm/d from 3-m ice (with 10-cm snow) at -25C. It should be noted that the submerged jet, and the upwelling above it near the western shore, do not exist when the ice is moving, so that the oceanic flux would be much reduced in years without a fast-ice cover.

The authors response to my suggestion is gratifying. Thank you.

As I have mentioned, the baroclinic adjustment associated with the undercurrent core centred near 100 m only moves water above this depth to the surface; baroclinic adjustment below this depth drives water downward. Münchow et al. (2007) might be a better citation for the fractional presence of Pacific Water at different depths on sections along Nares Strait than Jones (2003) [*Münchow, A., Falkner, K. K., & Melling, H. (2007). Spatial continuity of measured seawater and tracer fluxes through Nares Strait, a dynamically wide channel bordering the Canadian Archipelago. Journal of Marine Research, 65(6), 759-788. figs. 6, 88, 11*]. You can see in their figures how the 50% Pacific Water contour corresponds roughly to the core of the undercurrent, with the remainder being predominately what you call mAW below this depth and meteoric water above it. But you also see that the mAW designation is a bit clumsy, since to be consistent you should be calling the PW, mPW

– no purity so far from the sources. You might be better off abandoning use of mAW (I'm am not sure if this is an accepted designation anyway) and talking about the water above the core being predominately PW and that below as being predominately AW. Incidentally the upper PW, from the Alaskan Coastal Current in summer, near 80m depth in western Nares Strait, is plenty warm enough to melt ice. It doesn't need help from the Atlantic and is much better positioned in the water column to do so.

Adjustments to the text based on my assessment are recommended.

Addressing this concern, we substantially changed the text. Particularly, the following changes were made:

Line 18-20 in the Abstract:

[revised manuscript text omitted]

8. Lines 504-505, "Transformation of these currents over steep topography generates baroclinic semidiurnal tidal wave that may considerably enhance vertical mixing through benthic stresses and shear instabilities": I suggest that the steep cliffs on the western shore, indicative of deep water close to shore, make turbulence and internal waves generated in the benthic boundary layer irrelevant to the ice far above. However, I believe there is a good possibility to generate strong turbulence, mixing and entrainment through the action of the tidal flow (Pite et al., 1995. JPO 25) on the very rough under-ice topography of Nares Strait (Ryan & Munchow, 2017).

I suggest that the authors give some thought to this alternate, and I believe more plausible, explanation for the source of ocean sensible heat.
The modifications in line 607 have created a paragraph that is a little unclear, since it deals with two features in the ice cover that appear traceable to different physical processes: 1) the continuous band of thinner ice within 10 km of shore, most plausibly attributable to warm water upwelling above the upwelling; 2) The early appearance of water near headlands, most plausibly attributable to upward

diffusion of sensible heat driven by turbulent instability of tidal current accelerated around headlines. I suggest that you could be more easily understood by having a separate paragraph for each issue.

*We changed this part to address all these concerns. And we accepted the reviewer's vision about the major mechanism responsible for the anomalies along the western coast:*

"***Similar to Peabody Bay, the oceanic heat flux needed for limiting sea ice growth along the western side of Nares Strait is associated with warm water at depths below 70-80 m (Fig. 9d). The upper thermocline layer in the main channel is mainly comprised of relatively cold PW, whereas the considerably warmer AW prevails at depths below 150 m (Jones et al., 2003; Jones and Eert, 2006; Münchow et al., 2007). The baroclinic adjustment of the ocean to the intensification of the southward current in winter induces upwelling above the core of southward undercurrent that shifts the upper thermocline closer to the surface (Rabe et al., 2012; Shroyer et al., 2017). As a result, water above the freezing point can be found starting from 30-40 m depth near the Ellesmere coast (Fig. 9d) forming favourable conditions for a larger heat transport to the bottom of sea ice here.*** *Unfortunately, the lack of in-situ measurements does not allow us to quantify the vertical oceanic heat fluxes into the western Nares Strait polynyas, but it is likely that the barotropic semidiurnal tide, with magnitudes comparable to the speed of the mean southward flow (Münchow, 2016; Davis et al., 2019), greatly affects their intensity. Transformation of these currents over steep topography generates a baroclinic semidiurnal tidal wave that may considerably enhance vertical mixing* ***in the water column*** *through benthic stresses and shear instabilities (Davis et al., 2019). From this perspective, the fact that most of the western polynyas first appear near prominent headlands (Fig. 11) generally support the idea that the enhanced heat fluxes along the Ellesmere coast are attributed to the topographically controlled instabilities associated with the mean current and reversible tidal flow. Another mechanism that may enhance the heat flux in the area is associated with the sub-ice turbulence generated by interaction of very rough under-ice topography* ***in the channel*** *and tidal flow (Ryan and Münchow, 2017)* ***that is expected to be accelerated around headlands.*** *In combination with upwelling of the upper thermocline water along the western coast in winter (Shroyer et al., 2017), this mechanism may be* ***considered as a key factor*** *resulting in* ***an enhanced vertical heat flux towards the bottom of sea ice*** ***along the Ellesmere coast.***"

Query by the authors: Not for the paper, but we want to point at one thing related to the idea of sub-ice tidal mixing that makes us confused. The semidiurnal tide forms a standing wave pattern in Kane Basin with the lowest tidal velocities in its central part (Davis et al., 2019). If so and if the negative anomalies along the western coast are tidally driven, they should have become smaller around Cape Frazer. However, we don't clearly see it in Fig. 4b-c.

Perhaps you are referring to Davis et al. (2018) fig. 4? This actually depicts the amplitude of the semi-diurnally varying flux of tidal energy, not the speed of tidal current. The flux is $P = \rho \cdot g \cdot h \cdot $, where $u$ is the depth-averaged tidal current and $\eta$ is the tidal elevation; $P$ and $u$ are vectors. Both these properties vary semi-diurnally and can have different phase. Fig. 4 illustrates that differences in phase result in a northward flux of tidal energy on the Greenland side of Kane Basin and a southward flux on the Ellesmere side. Note also that the tidal energy flux increases with water depth if $u$ remains constant. I think that the strength of tidal currents off headlands – a local effect – depends very much on the character of the obstruction that the headland and adjacent submerged terrain pose to tidal

flow. It is also possible that at Cape Frazer – a steep high cliff – drainage of snow melt-water from the heights onto the ice may help to expand the polynya in June.

That query had originated from the analysis of high-resolution tidal simulation data in a few positions along the channel shared with us by Laurie Padman while we were working on our previous Nares Strait paper, not from Davis et al. (2018) paper. That hi-res dataset showed the decrease of the tidal velocities towards the (approximate) center of the Kane Basin.

9. Line 523, "weakens the cohesion of landfast ice against the shoreline in Kane Basin": The tidal cycles in sea level ensure that the ice sheet is always fractured at the coast, not bonded to it. However, the word cohesion implies that the authors consider that bonding of ice to the shoreline is important. This line of thought runs contrary to decades-old discussions of fast ice in deep water, where it is the formation of ice arches across channels which stops the movement of ice behind them, not shear strength at the shoreline. The upwardly convex shape of a masonry arch is the key feature that allows it to resist downward loading; the shape ensures that all the stone in the arch is under compression, the stress state in which it is strongest. Indeed the stress is highest within the wedge-shaded stones of the arch and much less above them. Pack ice also is strongest in compression and much weaker in shear. Although there are likely several arch-shaped load-bearing features distributed in the fast ice along the length of Nares Strait during any winter, much of the fast ice cover will be in a low state of stress; cohesion at the shoreline is probably unnecessary for fast-ice stability, although its confinement by irregularly shaped shorelines may constrain it from moving locally. Conversely, weakening of that confinement by melting at the coast may allow it to shift around in response to wind and tide. It is quite common to see the ice in Kennedy Channel become mobile between arches at its northern and southern ends long before the collapse of the arch in Smith Sound allows the ice in Kane basin to do the same. The same phenomenon is seen annually in Prince Regent Inlet. I don't think that the authors' argument for up-channel polynyas hastening the break-up of fast ice further down-channel has much merit, as presently written. It is possible, of course, that phenomena may be correlated in time because of the influence of a third circumstance not identified.

The authors appear to agree with my comment that cohesion of ice at the coast is not important to the longevity of the ice bridge. However, they continue to use the word 'cohesion' in the revised text at line 628, which I think is misleading. I suggest that they substitute something along the lines of "… we suggest that thinner ice and open leads at the shoreline of Kane Basin, linked to locally enhanced oceanic heat flux, provides weakness that allows patches of fast ice further from the coast to fracture and move in response to wind and current".

They subsequently speculate that collisions of these freed patches with remaining parts of the bridge "may" contribute to an earlier collapse of the bridge. This speculation is okay as an hypothesis, but the comment is of little value to the reader if it is just left hanging there. The reader has no idea whether or not this suggestion is worthy. How about adding a little "back of an envelope" physics to see whether the speculation has merit?

I decided to calculate the shock-loading of the fast-ice arch by the impact of a free-drifting floe for comparison with the sustained loading of an equal segment of the arch by the accumulated drag forces of wind and current on upstream fast ice. I assumed a 1-km width of contact by a square floe, 2.5 m thick, moving at 0.3 m/s and brought to a stop after ridging 30 m of ice along the zone of contact; with assumed constant deceleration, the force of impact was 3.45 million newtons, sustained over 200s. For the comparison, I assumed that wind and current stresses were accumulated over a fast-ice strip of the same width stretching 100 km up-channel from the arch, subject to the same 0.3 m/s current and to 15 m/s wind; the calculated loading was 37.5 million newtons on just this 1 km segment of the arch, sustained for days to months. In engineering terms, the arch has a safety factor of 10.9

for resilience against impacts of the specified type. The colliding floe would have to stretch more than 10.9 km up-drift for the safety factor to drop below 1, but this vulnerability would last for only a brief 200-s interval.

In my mind, this calculation suggests that the authors' idea that collisions with freed patches may contribute to an earlier collapse of the bridge is a non-starter … but perhaps the authors can devise a different scenario and simple physical model wherein the odds are not heavily stacked against their idea. If they cannot, I suggest that it be dropped from the text.

With all respect to the reviewer and understanding all difficulties of testing our hypothesis, we don't want to drop it from the text. The numbers used by the reviewer are somehow arbitrary chosen from our point of view, but even if they are right, the impact of freed ice floes does not replace but adds stress to the arch. Taking into account that the sea ice in the strait is already deteriorated by the middle of summer and therefore weaker, the arch might not be able to bear this top-up stress.

However, we have to admit our other mistake. We forgot to mention another factor associated with the mobile ice areas that seems to play the key role in facilitating the ice bridge collapse – an inward destruction of the bridge. This process is perfectly seen in the MODIS images between 23-26 June, 2020. The partial collapse and closing the polynya at Cape Jackson on June 23 led to a situation when more and more fractures appeared in the surrounding ice and the ice bridge breaks "inward". Such collapse can be observed in our Fig. 12 and 13, but is seen more clearly when switching between the sequential MODIS (for example in the WorldView) or Sentinel-2 images.

One more argument will be referencing Plante et al. (2020) who investigated the behaviour and collapse of the ice bridge in Nares Strait with an idealized ice bridge model. They described a collapse of the bridge initiated by upstream fractures: "The propagation of damage from these locations is composed of two separate fractures. First, a shear fracture progresses downstream along the channel walls, resulting in the decohesion of the landfast ice in the channel from the channel walls. The decohesion of the ice bridge increases the load on the downstream ice arch and on the landfast ice upstream of the channel…".

Addressing this concern, we changed Fig.13 (to make a better focus on fracturing by replacing MODIS images with Sentinel-2 data) and modified this part of the text as follows:

"*We can further speculate that such weakening may facilitate an earlier ice bridge break-up (comparing to a supposed no-polynyas situation) as it leads to formation of patches of mobile ice in the middle of the ice bridge in Kane Basin. While shifting around, this ice may gain some kinetic energy from wind and tide and eventually result in additional dynamical load on the parts of the bridge that still remain in place **but are thermally deteriorated by this time of year. More important, however, is that the edge of these patches does not provide a necessary structural support (as the arch in Smith Sound does) and that mechanical stresses cause an inward collapse of the ice bridge. This process can be seen as a propagation of new fractures outwards from the polynya area. Similar fracturing, which results in a decohesion of the landfast ice in the channel from the coast, was reported for simulations of the ice bridge collapse in Plante et al. (2020). For example, in 2020,** the polynya was generally well-constrained during March-May **(Fig. 12)**. The polynya started to expand in late May, reached its maximum size in mid-June, and **the surrounding ice cover** broke up around 22 June (Fig. 12). This **partial** break-up **of the landfast ice***

*surrounding the polynya* appeared to initiate further fracturing of the ice cover in the main channel **(Fig. 13)."**

And the beginning of the next paragraph:

*"**Although the satellite imagery presented Fig. 13 and results of the numerical modelling (Plante et al., 2020) generally support** our **suggestion**  that visible and invisible (i.e., thin ice areas) polynyas **may** facilitate ice bridge break-up in Nares Strait**, this suggestion remains speculative without more detailed research. However,** we would like to emphasize the observations that the first movements of the immobilized ice cover occurred in areas with negative ice thickness anomalies during winter and where polynyas are observed."*

10. Line 528-529, "This break-up appeared to release internal stresses in the ice bridge and led to concomitant ice cover break-ups in the main channel": This statement appears to rely upon a knowledge of the dynamical state of the ice cover. Nothing is known about stresses. In reality all you have access to is evidence of deformation (in the form of cracks) and of motion. Also see comment #9.

The authors have voiced agreement with my comment. They have changed the sentence to read "This break-up appeared to initiate the further fracturing of ice cover in the main channel". Since break-up and fracturing of the ice cover are essentially the same thing, this sentence seems circular and of little import. I may have missed the point, but either way, the sentence needs to be re-worked for improved clarity.

Under a breaking of the polynya we meant a partial collapse of the surrounding ice and filling the polynya area with mobile ice. We see the reviewer's point, but we could not find a better wording than:

*"This **partial** break-up **of the landfast ice surrounding the polynya** appeared to initiate further fracturing of the ice cover in the main channel on 23-26 June **(Fig. 13)**."*

Also see our more detailed response to the previous comment.

11. Line 531: This paragraph gives the impression that the polynya has played a role in the breakup, but really all that you demonstrate is that the breakup was correlated with expansion of the polynya. Perhaps the expansion of the polynya is just one event in the process. A more robust discussion, with a more useful take-away, would review the other factors in play, as listed in Line 520. If such completeness is thought to be beyond the scope of the paper, perhaps it should be covered in a separate paper. See comment #9.

The revised manuscript has new text in line 646: "although our hypothesis that polynyas facilitate ice bridge break-up in Nares Strait is speculative, we would like to emphasize the observations that the first movements of the immobilized ice cover occurred in areas with negative ice thickness anomalies during winter and where polynyas are observed." Hypotheses are typically the second stage of the scientific method, following collection and examination of data relevant to the phenomenon to be studied. They remain as hypotheses until a mechanism of interaction has been identified and the direction of causation established.
Since both these subsequent steps remain undone, the authors' science is incomplete, meaning in strict terms that this paper is premature. However, if what appears in the conclusions is a limited list

of plausible hypotheses to guide future research, with mechanisms tentatively identified) and with the list clearly flagged as such conjecture, with nothing really "proven", this could be sufficient for acceptance of this paper.

We hope that the changes suggested in the text and a new version of Fig. 13 were found convincing to address this concern (see our response to the comment #9). Particularly, the problematic sentence was rewritten as follows:

*"Although the satellite imagery presented Fig. 13 and results of the numerical modelling (Plante et al., 2020) generally support our suggestion  that visible and invisible (i.e., thin ice areas) polynyas may facilitate ice bridge break-up in Nares Strait, this suggestion remains speculative without more detailed research. However, we would like to emphasize the observations that the first movements of the immobilized ice cover occurred in areas with negative ice thickness anomalies during winter and where polynyas are observed."*

12. Line 539-540, "The only oceanic heat source available to maintain such a polynya through winter is the modified Atlantic Water": This is not true. It may be the warmest source, but it is not the one closest to the ice. My comment on Line 484 raises the possibility that the less conspicuous warmth of the Pacific Water might be more influential than you give credit for. I recommend that you re-think the paper with this in mind.

The authors response to this comment appears entangled with the definitions of the water masses themselves; see my concluding comments under item (7) above. All water masses are only pure at their sources, so every water mass is a modified something. Moreover, the relationship is reciprocal, if PW mixes with AW, is the result mAW or mPW? The distinction between AW and mAW seems irrelevant. Admittedly, some sensible heat from the AW has reached the overlain Pacific Winter Water by the time the mixture enters Nares Strait, but less dense varieties of PW higher in the water column, namely Bering Sea Summer Water and Alaskan Coastal Current (summer) Water are less likely to have been influenced from below and more likely to have been influenced from above, by summertime Meteoric and ice-melt waters and wintertime freezing-associated brines. As stated under (7), I consider it less confusing to label a water mass PW if it is more than 50% virgin Pacific, and similarly AW if more than 50% virgin Atlantic. Then the hair-splitting about which water mass is most influential can be abandoned.

We did the best to solve this problem in the text and focus on consequences of the presence of warm subsurface water in the region rather than on its origin. The suggested changes in the text can be found in our response to the comment #7 above.

Definitions aside, I still maintain that if AW is to be lifted somehow to the base of the ice, the overlying PW will by necessity have to have gotten there first, and with a smaller expenditure of energy. So I judge the hypothesis that sensible heat reaches the ice from PW and perhaps also from AW to be more plausible than the converse, namely that sensible heat reaches the ice from AW and perhaps also from PW. Only if the authors could come up with an energetically viable mechanism to explain the latter version would I be more believing of it.

We agreed on this and placed PW on the first place wherever we discussed the impact of the uptake of heat from below in the main channel. However, we suggest that the warm subsurface

water in Peabody Bay mainly consists of AW originated from Baffin Bay. But a possible recirculation of PW in Kane Basin has also been considered and mentioned in the text.

I recommend that the revised text provided by the authors and quoted under this item be re-worked either in terms of a simpler stack of water masses (i.e., ASW, PW, AW) which perforce are blended across their interfaces, or instead referenced to a partition of source waters in terms of T & S thresholds. Either approach would make this discussion much less enigmatic.

We got rid of mAW term and changed the text accordingly in terms of water mass terminology.

With reference to "This flow occupies the entire water column and consists of 3 distinctive layers; i) cold brackish polar mixed water within the upper 50-60 m, ii) the upper thermocline coinciding with halocline that is observed at 70-110 m …" in the revised version of the text, I caution the authors to look carefully at the Rabe papers which show appreciable differences between T & S profiles in winter and those in summer, which are quoted from the Jones papers. Since the present paper concerns winter, the former reference may be a more relevant source.

We did not understand completely what differences the reviewer meant particularly. But we would like to say that we used FESOM-2 data obtained during winter period (Fig. 9b and Fig. 9d) while describing a possible impact of warm subsurface water on ocean heat flux in the main channel and in Peabody Bay. Probably, this comment is not relevant anymore since the text was modified.

**Comments (minor)**

Line 38, "… peak in the fraction of sea ice with a draft between 2.6-2.8 m": It is important to note here, as was in the cited paper, that this range in draft was computed on the assumption of no snow cover, which may bias values appreciably high. Also, a referenced estimate of the empirical accuracy in draft estimates from CryoSat freeboard should be included here.

In their response, the authors do not draw a distinction between random errors, which do shrink as the number of values averaged increases, and bias errors that do not. Since all values in a 1.5x1.5 km square share the same bias, traceable to errors in estimating propagation delays from space, in the geoid and in the (unseen under fast ice) sea level, the average is biased by the same amount, whatever it is. The authors still need to provide information on the magnitude and character of bias in CryoSat as part of the necessary error analysis.

We understand the difference between random errors and possible biases. However, the changes suggested in our first response were related to the accuracy of the ICESat-2 readings and calculated anomalies, i.e. random errors. The possible biases related to other factors were discussed further in the text and also in Section 3.2. And because we are working with regional anomalies, we limited our discussion to the biases that vary regionally (sea level, steric heights). The biases that seem to not vary much over Kane Basin (like propagation delays from space or the geoid) are not thought to affect our findings and the observed strong local elevation gradients.

Otherwise, we don't understand what kind of additional information on the magnitude and character of bias in ICESat-2 data is requested by the reviewer.

Lines 46-47, "That bridgeless years only occurred during last 15 years underscore a general shortening of bridge existence period and point to changes …": It would be appropriate to clarify that this statement refers to the absence of an ice bridge at Smith Sound (think) and not to the much smaller number of years when there was no bridge anywhere between Baffin Bay and the Arctic Ocean.
In this clarified context, it should then be noted that there was one winter (1995) in the 1990s with no arch at Smith Sound – in 1995 the arch formed at Hans Island – and one (1993) essentially like 2007 with no arch anywhere; "essentially" because an arch in Smith Sound that year lasted only 10 days (Vincent 2019). With a 30-year perspective, the record looks less amenable to interpretation via trend: there is a cluster of 2 of 3 years with no arch at Smith Sound in the mid-1990s, then an 11-y period with annual arches, then a cluster of 3 of 4 years with no arch in the 2nd half of the 2000s, then a 6-y period with annual arches, then a cluster of 2 of 3 years with no arch in the second half of the 2010s. Disregarding clustering and estimating the probability of no bridge in any year from the data as 7/31, one uses the Poisson Distribution to estimate the likelihoods of the observed gaps between no-bridge winter – that is having 2 no-bridge years in 2 years, 2 in 3 y, 2 in 7 y and 2 in 12 y. These are 6.4%, 11.7%, 25.7%, 24.4%. The low values for the small gaps suggest there is clustering in play; the relatively high values for the large gaps suggest that such wide gaps are not unexpected, so that bridging despite weak clustering, looks like a Poisson process. On these grounds I suggest a re-examination the statistical confidence of the statement in lines 46-47, which is based on such a short time series.

I took pains in my first review to demonstrate that the small number of bridgeless events is insufficient to draw with high confidence a distinction in their occurrence between the first two and the last two decades.
I stress that it remains important to provide limits of confidence on 2 as the mean for the first two decades and on 6 as the mean for the last two, and to calculate the confidence with which the data allows these two numbers to be considered different. Without error analysis, this is just handwaving.

We apologise for not including the results of statistical analysis made by the reviewer, but we don't really think it is relevant to the Introduction. We just share general information and known facts about the bridge in Nares Strait. In the concerned sentence, we presented **the fact** that "the ice bridge failed to form only two times during the first two decades of observational records (in 1993 and 1995; Vincent, 2019) and six times during last two decades (in 2007, 2009, 2010, 2017, 2019 and 2022)". If the main concern is about attributing this information to "changes in environmental conditions", we can partly agree with the reviewer. Neither the fact that 6 out of 8 bridgeless years happened during last two decades nor a general shortening of bridge existence period reported by Vincent (2019) have very high statistical significance.

We slightly changed this sentence as follows:

"*Based on AVHRR satellite data from 1979 to 2019, Vincent (2019) reported on a recent trend**, though not confident,** towards later formation and earlier breakup of the ice bridge. **Additionally, Vincent (2019) found** that the ice bridge failed to form only two times during the first two decades of observational records (in 1993 and 1995; Vincent, 2019) and six times during last two decades (in 2007, 2009, 2010, 2017, 2019 and 2022), underscoring a general shortening **in the duration** of the bridge and pointing to changes in the environmental conditions.*"

Line 54-56, "it is the sensible heat polynyas … that are more common in the Canadian Arctic (Hannah et al., 2009)": The authors appear to mis-quote Hannah et al. (2009), who state "… are widely distributed across the Canadian Arctic Archipelago"; Hannah at al. are clear that these sensible heat polynyas are features within fast ice in this region. Their map (Fig. 1) shows that the latent heat flaw-leads and polynyas that form along the perimeter of the fast ice are actually more widespread across the Canadian Arctic waters and occupy much more area.

Okay, except that Hannah's paper actually did not consider heat sources. It simply correlated existence of polynyas in fast ice with h/u^3. Areas with small values of this parameter typically have strong current, maintaining the likelihood that both latent and sensible heat processes are in play. The Topham reference documented just this situation at the Dundas Island polynya. Please adjust your text accordingly.

The criticism is reasonable. The sentences were changed as follows:

"*Beyond the NOW and other latent heat polynyas, there are several polynyas that form within the landfast ice cover of the Canadian Arctic **and are at least partly attributed to the sensible heat flux from the ocean (Topham et al., 1983; Smith et al., 1990)**. Most of these polynyas are formed in the narrows where strong tidal and mean currents **cause vigorous vertical mixing and** facilitate upward heat transfer from **the warm subsurface water at depth (Topham et al., 1983)**.*"

Line 132, "Although ATL07 data are manifested to be adjusted for geoidal/tidal variations and inverted barometer effects": The correction for the inverted barometer effect is probably only accurate in wide deep ocean basins where the long ocean wave which is the ocean's response to changing atmospheric pressure can move as fast as, and in the same direction as, the SLP anomalies moving at 20-25 m/s. I suspect that the correction will not work well in a long (550 km) narrow (35 km) strait. I urge the authors to find and reference research that provides a discussion of the accuracy of the inverted barometer correction in confined coastal waters.

Notwithstanding the authors' hopeful "don't worry" comment in their response, this issue remains unresolved in my mind without reassuring observations. End-to-end SLP difference of 25 mb are quite common in Nares Strait (Samelson papers); who knows how much of the 25 cm of sea-level adjustment is actually distributed along the strait?
I therefore urge the authors to at least acknowledge the possibility that bias from inaccurate inverse-barometer correction could be a source of error, unless they can find data do better.

We don't really like such interpretation of our initial comment. In our response we wrote that "the width of the strait is too small for spatial (cross-channel) SLP variations having a large effect on the CROSS-CHANNEL sea level difference". Regarding the along-channel variations of the SLP we can roughly estimate the mean gradient of the sea level that corresponds to the mentioned difference: 0.45 cm per 10 km. The ICESat-2 along-track elevation changes (within coastal anomaly areas) exceed this gradient more than by an order of magnitude. In addition, our Fig. 4 represents the time-averaged anomalies, whereas 25 mb difference is considerably larger than a typical 4-5 mb along-channel pressure difference (e.g. Samelson and Barber, 2008).

In our initial response we also presented another argument about a long-wave speed in Nares Strait. We found it large enough to adjust the sea level in the strait to moving SLP anomalies reasonably fast.

All aforesaid allows us to reasonably suggest that SLP in general and the accuracy of the inverse-barometer correction in particular have negligible impact on the observed elevation anomalies discussed in our paper. With a respect to the reviewer's opinion, we don't see any necessity of the additional discussion around inverse-barometer correction in the context of our study.

Line 139-140, ">0.3 m mean snow depth in Kane Basin. However, as 140 we will show later, this height seems to be overestimated". Reference to Samelson and Barbour (2010) is again appropriate, since the extremely strong winds common in Kennedy Channel and the vicinity of Cape Jackson (see also Melling, Oceanography Mag, 2011) may indeed provide a strong disincentive for the accumulation of snow.

The authors must not have looked carefully at the suggested reference before making their comment: "we are talking about the central Kane Basin and Peabody Bay. We don't think the orographic effect plays the same role in snow accumulation rates as it does in Kennedy Channel". In fact, as clear in Samelson's paper, fig. 2, 6 and particularly 9, the areas of strongest gap-flow winds are not in Kennedy Channel but in north-central Kane Basin, off Cape Jackson and south of Smith Sound. Also, I suspect that there is a strong likelihood of katabatic winds off the Humboldt Glacier into Peabody Bay in winter, but the authors seem not to explored this possibility. A renewes look at these issues is highly recommended.

We can give a contra argument by referencing Moore (2021) and his Fig. 10c published in the Scientific Reports. From that figure it's clearly seen that Cape Jackson is located slightly aside of the area where strong orographic or gap-flow winds are present. In addition, we would like to emphasize again that we used no-snow assumption in simulating the ice growth within polynya area at Cape Jackson. The different snow accumulation rates were used for simulating of ice growth in an abstract position located few dozen kilometers south-east of the polynya within landfast ice – much farther from the channel and from the gap-flow region. Also see our response to the comment #3.

However, we don't see a need of referencing Samelson and Barbour (2008) because the gap-flow effect is attributed to the exit from the Kennedy Channel. The mean snow depth of >0.3 m (Landy et al., 2017) characterises snow cover approximately in the centre of Kane Basin. We have specified in the text that Landy's data is related to the **central** Kane Basin.

In terms of a katabatic wind we can't do much. The only relevant publication we could find was a field report of Günther Heinemann who presented the results of a POLAR-5 flight over Humboldt glacier and Peabody Bay in June 2010 ("Investigation of Katabatic winds and Polynyas during Summer" in Reports on Polar and Marine Research, 633). They reported a weak katabatic flow over the glacier and even more weaker wind over landfast ice, but their results were obtained during summer. Without more robust information about this flow, we don't want to speculate about it in the paper.

Lines 231-242 & Fig. 4, "In the main channel, the anomalies are highly irregular and form a speckled pattern, whereas the anomalies in Peabody Bay form a consistent pattern with positive anomalies in the

southeast and negative anomalies to the northwest": It is unclear, with the continually moving ice of 2019, why the elevation anomalies are not smoothed out via averaging over time. The small scale of the speckle in elevation in 2019, not so different from that in the years with immobile ice is difficult to understand. Please explain.

Perhaps the authors could include in the manuscript, for readers' benefit, a truncated version of the explanation provided in their response to me?

Done as requested. The followed sentence was added to Section 2.2

"…and then averaged onto a 1.5x1.5 km grid. **This resolution was chosen to ensure that neighboring strong beams, which are separated by ~3 km, were not projected into the same grid cell. Although such resolution increases the noise in the spatial distribution of h̃ (especially in the areas with mobile ice), it highlights the areas with strong elevation gradients that are spatially consistent throughout winter.**"

Lines 288-289, "For having ice-free water in May, the heat flux should reach 70 W m$_{-2}$ and be above 200 W m$_{-2}$ to let polynya form in early March": These estimates presume that there is no advection of newly formed ice downstream and beneath thicker pre-existing level ice and, I believe, that there is no insolation.

See my comment under (4)

A few changes were made in the text to address this reviewer's concern. Please see our responses to the comments #3 and #4

Lines 543-544, "Münchow (2011) reported a very similar warming in the southward branch of mAW of 0.23 °C/decade": Actually Münchow et al. (2011). This paper provides very weak evidence of long-term warming because the period of observation was only 6 years. The present authors have taken the liberty of extrapolating this to 10 years, and then referring to a supposed "as further warming of mAW progresses" – all this without having made a bullet-proof case for an influence of mAW on the sea ice of Nares Strait. It is one thing to have mAW affect glacial ice at the same depth, quite another to postulate an influence on sea ice at the surface hundreds of meters above. I suggest to the authors that the present evidence to make this projection is not statistically robust.

I am okay with the first part of the authors' response, but not with the second, namely "We … have reduced the emphasis on the future projections. However, we can't simply reject it because, if our suggestion about the impact of mAW on landfast ice is right, the warming of this water is important to mention".
The issue in contention is the difference between scientific publication and news publication. In the latter, the primary goal is to sell newspapers. In science we strive for robust results with carefully estimated bounds of uncertainty, numbers that we can provide with confidence to other scientists to help in their work. We can all speculate that something may or may not happen. However, the key factors are how confident we are in what we know has happened, how confident we are that it will continue to happen and what will be the value some years hence, plus or minus what. Since these key factors have not been addressed, the statement about warming, "if it is right" as the authors say, is

of little value to science and can be omitted. If the authors can address these key factors, then the statement certainly belongs in the paper.

Even if our response to the reviewer sounded scientifically unacceptable for some reason, the speculation about the warming ocean impact on the ice bridge is written in the text as a pure suggestion. We rewrote this part as follows

*"**In terms of Peabody Bay,** AW in front of **the nearby** Humboldt glacier **has warmed** by 0.9 °C since 1961 (Rignot et al., 2021**), although this trend is based on fairly sparse summer data in the area and is therefore highly uncertain**. However, based on a more consistent, **albeit** much shorter time series of mooring data obtained in Kennedy Channel in 2007-2009, Münchow et al. (2011) reported a statistically significant warming in the southward branch of AW of 0.027 °C/year. **There is also evidence that AW temperatures increased in the Lincoln Sea between 2003-2011 and the 1990s (de Steur et al., 2013). Through the mechanisms we have described here, warming AW in Nares Strait exerts a greater influence on ice growth and therefore the stability of the ice bridge. In fact, coincident to the warming of AW in and around Nares Strait has been a recent tendency for the duration of the ice bridge to become shorter, or for the bridge to not form at all, an event that has now happened during** six of the last 15 years (i.e., 2007, 2009, 2010, 2017, 2019 and 2022). **Continued warming of AW as a result of climate change will continue to affect the formation and stability of the ice bridge, and may lead to even more years when an ice bridge fails to form in Nares Strait."***

which sets a task for future studies:

*"However, a quantitative estimate of the **changing** ocean thermal impact on bridge stability can be only accomplished with coupled ice-ocean models incorporating a comprehensive ice rheology or sea ice dynamics model (e.g. Plante et al., 2020; West et al., 2021), which is beyond the scope of this study."*

that will address factors that were not addressed in the current research.

In the following, the comments by the reviewer are in black characters and our responses to the comments are made in blue and indented. The comments where the reviewer was satisfied with our initial answers and/or changes were left out of this document for simplicity. Modifications to the text are shown in quotation marks with bold characters indicating newly added text, and normal characters indicating text that was already present in the previous version.

General comments:

This manuscript presents a scientifically interesting body of work, with great use of a variety of remote-sensing datasets and a modeling exercise together to make novel inferences about Nares Strait ice bridge structure and breakup. The authors have done a good job addressing reviewers' questions and concerns, and have produced a much clearer and polished revised manuscript. Assumptions and limitations of the results are much improved, and the storyline is a lot tighter. I am very impressed with the creativity in how these datasets were used and am looking forward to referencing it once published. I have a few comments and points of clarification (stated in the attached document) that should be addressed, but otherwise I recommend acceptance with minor revisions.

Specific comments:

121-125– Great addition so far. However, MODIS is thermal infrared not microwave, and brightness temperature is also impacted by cloud and atmospheric constituents/temperature.

> Thank you for finding this. We changed the sentence as followed:
> "*Note neither AMSR2 nor MODIS brightness temperatures are indicative of surface temperature alone, but measure the radiance of microwave **and mid-infrared** radiation that is expressed in units of temperature (K) of an equivalent blackbody.*"
>
> The fact that MODIS band 31 channel is influenced by cloud was mentioned earlier in Section 2.1.

444-450/Figure 10 – Apologies for the not-quite-complete comment on this during the first round of review. Your edits here improved this part of the discussion a lot despite a lack of complete info on what I was meaning. What I had meant to say is that when I looked at the timeseries of MODIS thermal images in WorldView, the image you show from 2019 seems like it may be a high wind event pushing ice away from the coast (maybe a latent heat polynya of sorts, which would not necessarily be indicative of high ocean temperatures and could cause high sea ice production) and not the best representation of the Tb for that month.
It's an extreme and ephemeral event for the month. A few days earlier and later in Dec 2019 looked very different. A similarly "warm-looking" event occurred in the 2018/2019 season for just a day or two. However, looking at all of the images from Dec 2019, I would agree that Tb was generally higher. A monthly mean Tb or timeseries of clear-sky days for the area would be much more useful in supporting your statements here for this reason. I don't think that a change is required for publication here, although it would make for a stronger statement and it could give you more insight about the relationship between sea ice height and Tb, as well as understanding mechanisms impacting bridge formation and breakup. I also use this as a caution against using extreme synoptic event days (Dec 18, 2019) to represent a monthly/interannual difference. That said, in general this section of text is oversimplifying a lot about Tb and not accounting for the complex sea ice-ocean-atmosphere interactions that are taking place. These

statements are not quite accurate to what you can say from the figures you show without further explanation.

We generally agree with the reviewer that monthly mean MODIS $T_b$ would be more beneficial for showing a "significant interannual variability". However, in addition to a clear-sky limitation (there are certain difficulties of determining clear sky over the entire Kane Basin during polar night), the presence of mobile ice in the channel make this task impossible. Because what we wanted to demonstrate was a difference of temperatures within individual leads or openings (not the mean temperatures over a large area) in the different parts of the region. If higher $T_b$ are associated with open water at freezing point within leads opened by winds or currents, we would have seen approximately the same temperatures in the leads everywhere.
To address this concern, we suggested the following changes in this paragraph:

"*The MODIS brightness temperatures, Tb, shown in Fig. 10 generally support the idea that the thermal state of the surface* **water** *in Nares Strait* **may vary considerably**. *In December 2019 (Fig. 10b)*, **for example**, *the high Tb conditioned the ice-free (or covered with thin ice) area in the northwestern part of Peabody Bay and at the eastern side of Kennedy Channel.* **The pattern of higher temperatures in this area resembles a plume that extends from the landfast ice edge towards the entrance of Kennedy Channel.** *Although the signatures* **of surface water with different temperatures** *in Kennedy Channel can also be traced in the leads within the mobile sea ice in December 2018 and 2020, Tb was observed to generally be lower and may indicate reduced ocean heat transport towards the surface from below* **during those years**. *Of course, the* **spatial** *difference in temperatures* **(within leads)** *presented in these early winter snapshot images cannot explain the seasonally averaged elevation anomalies shown in Fig. 4. However, these differences demonstrate that there may be significant variability in the* **sensible** *heat flux* **at the ocean surface around the time when sea ice begins to form**."

444-446 – "The high Tb conditioned the ice-free area in Peabody Bay", "signatures of warmer water in Kennedy Channel can also be traced through leads within the mobile sea ice" – this ends up being a little misleading for the reader. Since the temperatures you are showing aren't sea surface temperature and sit below saltwater freezing temperatures, it isn't clear what is causing the high Tb in 2019 or in the leads. The high Tb may merely be arising from a lack of sea ice at the surface and cold ocean surface temperatures (ocean is warmer than sea ice). The high Tb and lower sea ice concentrations could arise from low sea ice transport out of the Arctic and/or a windstorm blowing sea ice out of this area (which also could drive upwelling of warm water but doesn't necessarily need to do so to produce the Tb pattern you see). It may or may not also be from warmer ocean water coming to the surface, but I doubt that is happening in isolation from the other phenomena I mentioned. These comments also play into line 488-489.

As we explained above it is a contrast of temperatures within individual leads in different parts of the research area that made us thinking that surface water temperatures vary spatially and "interannually". The leads formed by wind or currents alone would not have resulted in such differences because the temperatures would have been at freezing point everywhere. In December, when the sea ice has already started to form in the strait, it is the sensible heat flux from below that could explain higher surface water temperatures in some leads compared to others. Please see our suggested changes above.

447 - "may indicate reduced ocean heat transport towards the surface from below" – this is technically true with the use of "may", but it would be helpful to present the other alternatives so that the argument is balanced for the reader.

Please see our response to the comments to Lines 444-450 and 444-446.

447-449 - "the difference in temperatures presented in these early winter snapshot images cannot explain the seasonally averaged elevation anomalies shown in Fig. 4" - The notion of high winds/low sea ice transport into the strait causing the high Tb can more easily explain why sea ice thicknesses aren't consistent with the high Tb in 2019 than higher ocean heat flux to the surface can. For instance, more sea ice cover on the surface early in the season could insulate and prevent ocean heat escape that impacts sea ice formation afterwards. This could be discussed a little more in the text.

Please see our response to the comments to Lines 444-450 and 444-446.

605-608 – Since most of the iceberg discussion was removed from the paper, this no longer applies, correct?

No. The removed part was about a possible mechanism that causes the icebergs to form well-separated chains in the Peabody Bay. The idea that grounded icebergs affect the vertical mixing is self-evident, but without real measurements we put it here in a form of suggestion.

Line comments:

29 – ", which" makes the sentence a little clearer here

Changed as suggested

50 – not sure it makes sense to include "the last bridgeless winter" here since it is the most recent winter. I would cut this.

Removed as suggested

51 – points

Corrected

67 – spots based on evidence from nearby…?

Changed as follows:

*"On the western side of Nares Strait, sensible heat polynyas form near the Bache Peninsula along the eastern side of Ellesmere Island (Schledermann, 1980; Hannah et al., 2009) and seem to be highly biologically productive spots given the presence of prehistoric settlements in the area extending back 2500 to 3000 years (Schledermann, 1978)."*

85 – Cape

Corrected

103 – Spell out FESTOM-2 acronym here for first use, rather than on L170

Changed as suggested

125 - "did not"

Changed

130-131 – the segment lengths are in reference to ATL07 specifically and not all ICESat-2 data, so I recommend moving this sentence down one line to come after ATL07 is introduced.

It's a good point. Thank you for finding this out. The suggested changes were made.

134-137 – Have "although" and "however" in same sentence so recommend removing "Although"

Changed as suggested

140 – into "a" 1.5….

Corrected

145 – Compared to

Corrected

150 – Spell out for the first use of acronym (DMSP/SSM/I-ISSMIS), slashes are in the wrong spot at the end also.

We corrected the acronyms accordingly. However, we decided not to spell these acronyms out because they are related to the data used in the referenced studies, not in our paper.

161, 162, 166 – Spell out acronyms for first use?

We decided not to spell all these acronyms for the same reason.

166 – "a" finer

Corrected

178 – the model performs well in reproducing the shift…

Thank you for this suggestion. Changed as suggested.

181 – depth, which is in good…

Changed as suggested

196 – lower than ideal for resolving

Changed accordingly

198 – calculated using

Corrected

199-201 – what is your evidence for this statement?

To address the similar concern from the other reviewer, we performed a few 1D ice growth model experiments to show a sensitivity of ice thickness at the end of winter to the key forcing parameters (wind speed, air temperature, snowfall rate and ocean heat flux). The following text was added to Section 3.3:

"*To investigate a sensitivity of model results to the uncertainties in forcing parameters and/or possible biases of ERA5 data in Peabody Bay, we conducted several model runs with biased air temperature, wind speed, snowfall rate and ocean heat flux for the experiment presented in Fig. 6b. It was found that increasing wind speeds by 1 m s-1 increased ice thickness at the end of the season by 4.02 cm, whereas increasing air temperature, snow accumulation rate and ocean heat flux by 1° C, 1 cm mo-1 and 1 W m-2 reduced ice thickness by 2.45, 7.41 and 3.36 cm, respectively. Using these numbers, we can estimate a relative input of each parameter to the final results. Using 1 m/s, 2 °C, 2 cm/mo and 5 W m-2 as the possible biases or uncertainties of the model forcing parameters and taking the impact of ocean heat flux as 100%, we may estimate the relative input of other factors as 88% (snow), 29% (air temp) and 24% (wind speed) that underlines a larger contributing effect of snow and sensible heat flux on the ice growth.*"

217 – presence of a sensible

Corrected

218 – remove 'sea surface'

Thanks for finding this remain. Removed.

220 – I think you don't want "the latter" here

"…both the later" stands for two last (out of 3) reasons mentioned in the sentence.

238 – central main seems redundant so maybe this should just be central

The "main" was removed

271 – corresponds

Corrected

278 – compared to, also if you have it, it would be good to give a quantitative order of magnitude estimate for each of the gradients

> Unfortunately, there is no data on sea level under the ice bridge that would allow us to do so. This is the reason why we wrote *"are thought to be small".* We could probably use the data on the mean winter sea level difference between Alert and Thule (1.59 m, about 680 km apart) to demonstrate the mean gradient of about 2.3 cm per 10 km. This number is considerably smaller than 10-20 cm cross-shore changes of the elevations over approximately the same distance. However, the real distribution of sea level gradients below the bridge is unknown for such speculations.

292 – spell out SAR

> Done

303-304 – we will roughly estimate … or even "will estimate"

> Changed as suggested

331 – to the 19….

> Corrected

336 – would need to reach 70

> Changed as suggested

338 – at some distance from the polynya

> Corrected

339-40 – is large enough () to keep it ice-free…

> Changed as suggested

341 – to the 0.26 m mode

> Corrected

341-343 – A set of model experiments showed that the maximum…corresponds to … and a snow accumulation rate…

> Changed as suggested

From L343 to the end – stopped noting grammatical errors, but quite a few still exist beyond this point

> We have done a thorough review of the manuscript for grammatical errors.

365 – spell out first use of AO – but since it is the only use could remove the acronym

We don't know why AO appeared there. Changed to the Arctic Ocean

383 – how much shorter? Could give ranges or the average for each

The sentence was changed as follows:

*"Although the average duration of the observed ice bridges in Nares Strait (**about 5 months, between 20 January and 28 June; Vincent, 2019**) is shorter than that predicted with the model (>6 months),…"*

Figure 10 – What is the vertical-ish line in (b) and (c)?

These lines correspond to the approximate positions of the landfast ice at the moment of each imagery. The figure caption was modified accordingly.

459 – what does "contrast of the latest" mean?

"The latest" was replaced by " $\tilde{h}$ "

462 – away from Cape Jackson

Changed, thank you.

491 – the "only" available source – this isn't accurate based on your second sentence

Following recommendations of the other reviewer, this part was considerably changed.

527 – "ice elevation anomalies"

Changed as suggested

530-532 – "by the fact of forming the chain of polynyas" to "by the fact that chains of polynyas for…"

Changed as suggested

532 – Fig 11 should probably be referenced the sentence before.

Changed as suggested